# Masked, Regularized Fidelity With Diffusion Models For Highly Ill-posed Inverse Problems

## Abstract

Diffusion models have been well-investigated for solving ill-posed inverse problems to yield excellent performance. However, their application to highly ill-posed inverse problems remains challenging. In this work, we propose zero-shot diffusion model for large and complex kernels, dubbed Dilack, incorporating novel data fidelity terms. Based on our analyses on the ill-posedness for challenging inverse problems, we propose *regularized fidelity* called pseudo-inverse anchor for constraining (PiAC) fidelity loss. Inspired by locally acting classical regularizers, we also propose to incorporate *masked fidelity* within PiAC loss that can interact with globally acting diffusion models, which adaptively enforces spatially and step-wisely local fidelity via masks. Our proposed scheme effectively reduces erratic behavior and inherent artifacts in diffusion models, thereby improving restoration quality including perceptual aspects and outperforming prior arts on both synthetic and real-world datasets for modern lensless imaging and large motion deblurring.

## 1 Introduction

Image restoration (IR) is a fundamental low-level vision task that solves ill-posed inverse problems (Lim et al., 2017; Kupyn et al., 2019; Larsson et al., 2016; Song et al., 2021; Sriram et al., 2020). IR attempts to reconstruct the original image $\mathbf{x} \in \mathbb{R}^n$ from the degraded and/or undersampled measurement $\mathbf{y} \in \mathbb{R}^m$ by exploiting the forward model for imaging:

$$\mathbf{y} = A\mathbf{x} + \mathbf{n} \tag{1}$$

where $\mathbf{n}$ denotes a measurement noise and $A$ is an $m \times n$ measurement matrix. This operator $A$, often a kernel matrix, encapsulates the effects of forward imaging processes in various low-level vision tasks (Liang et al., 2021b; Luo et al., 2022; Xu et al., 2017; Quan et al., 2021). A typical formulation to solve the inverse problem based on Eq. (1) or to recover $\mathbf{x}$ from $\mathbf{y}$ involves minimizing a cost function expressed as:

$$\hat{\mathbf{x}} = \underset{\mathbf{x}}{\arg\min}\ \mathcal{L}(\mathbf{x}; \mathbf{y}) + \lambda \mathcal{R}(\mathbf{x}) \tag{2}$$

where $\mathcal{L}(\mathbf{x}; \mathbf{y})$ represents a data-fidelity term, $\mathcal{R}(\mathbf{x})$ denotes a prior term for regularization, $\lambda$ is a parameter balancing regularization with fidelity, and $\hat{\mathbf{x}}$ is the estimated image.

There are a number of approaches to formulate and solve Eq. (2). A classical approach uses regularizers such as sparsity or low-rankness for $\mathcal{R}(\mathbf{x})$ (Krishnan & Fergus, 2009; Dong et al., 2018; Tirer & Giryes, 2020). This approach works well with diverse imaging problems by modifying $A$ and $\mathbf{n}$ while using the same $\mathcal{R}(\mathbf{x})$, but it lacks perceptual priors (Boyd et al., 2011; Beck & Teboulle, 2009). Eq. (2) usually yields an iterative algorithm, but often leads to a closed form solution for some cases. Deep learning-based approach solves Eq. (2) by training deep neural networks (DNNs) to directly map from the measurement to the estimated image $\hat{\mathbf{x}} = \mathcal{D}(\mathbf{y})$ (Lee et al., 2023; Zeng & Lam, 2021; Khan et al., 2019; Wan et al., 2023; Li et al., 2023b; Zhong et al., 2023). While this approach has yielded excellent image quality, it often struggles due to their reliance on scarce, high-quality datasets and limited generalizability to new data variations (Song et al., 2021).

One promising approach to formulate and solve Eq. (2) is a hybrid form to use both the data fidelity term in the classical approach and the regularization term $\mathcal{R}(\mathbf{x})$ by exploiting DNNs so that DNNs can be decoupled with the forward model. This zero-shot (ZS) approach for IR can utilize pre-trained models such as generic denoisers (Metzler et al., 2017; Zhang et al., 2019; Ryu et al., 2019; Zhang

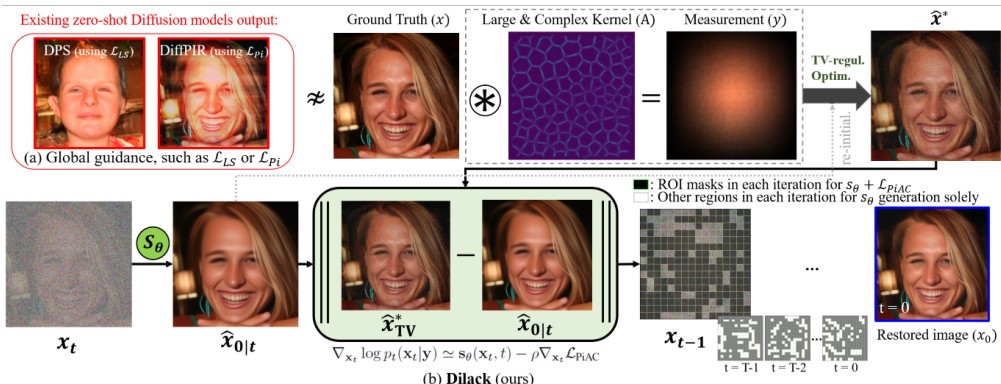

Figure 1: **Dilack** leverages total variation (TV)-regularized optimization to identify local inconsistencies and mask-guide sparse iterations, ensuring global consistency and quality. Unlike (a) existing zero-shot diffusion methods using $\mathcal{L}_{LS}$ (Eq. 9) or $\mathcal{L}_{Pi}$ (Eq. 10) guidance, (b) Dilack effectively tackles *severe* ill-posed problems, such as those found in lensless camera raw and large motion blur.

et al., 2021) or generative models (Bora et al., 2017). Recently, there have been significant advances in diffusion models (DMs), yielding state-of-the-art performance in diverse IR tasks (Dhariwal & Nichol, 2021; Song et al., 2020), enhancing reconstruction performance (Tang et al., 2024; Song et al., 2022; Mardani et al., 2023; Garber & Tirer, 2024) and boosting sampling efficiency (Chung et al., 2022b; Avrahami et al., 2023; Wang et al., 2024; Yang et al., 2024). Latent diffusion models (LDMs) have been proposed that operate in a lower-dimensional latent space instead of high-dimensional pixel space, satisfying both performance and computational cost (Rout et al., 2024; Song et al., 2024).

While existing zero-shot diffusion models (ZS DMs) effectively handle various standard degradation scenarios (Kawar et al., 2021; 2022; Wang et al., 2022) by leveraging the fidelity-prior framework in Eq. (2), these models have not been well-adopted to highly ill-posed inverse problems since they struggle with them in real-world applications such as modern lensless imaging and large motion deblurring unlike classical methods. Here we argue that generative models such as DMs interact with data fidelity term globally so that they may not be as effective as locally acting classical regularizers for highly ill-posed inverse problems. In fact, Bora et al. (2017) showed that using both generative model and classical regularizer helped to improve performance for less measurements (*i.e.*, more ill-posed) while classical regularizer did not help much for sufficient measurements.

Here we investigate the behavior of DMs interacting with data fidelity and then we propose ZS DM with masked fidelity that adaptively enforces spatially and step-wisely local fidelity for highly ill-posed restoration problems such as modern lensless imaging and large motion deblurring. Surprisingly, for these challenging problems, we demonstrate that ZS DMs often yielded poor results over classical methods (Boyd et al., 2011; Beck & Teboulle, 2009) even with model mismatch and calibration errors (Monakhova et al., 2019; Rego et al., 2021; Poudel & Nakarmi, 2024). Our masked data fidelity term enforces spatially and step-wisely local interactions with DMs by minimizing outlier generations in DMs for stability while maintaining enough global interaction.

**Our contributions:** (1) Analyzing the behaviors of current ZS DMs for solving highly ill-posed inverse problems with large and complex kernels. (2) Proposing a novel zero-shot **Di**ffusion model for **l**arge **a**nd **c**omplex **k**ernels, dubbed **Dilack**, with our novel fidelity that utilizes region-of-interest (ROI)-based spatially masked fidelity to dynamically toggle for emphasizing local consistency as well as step-wisely masked fidelity over iterations. Moreover, our regularized data fidelity term, PiAC (Pseudo-inverse Anchor for Constraining) loss, replaces the pseudo-inverse (Pi) with a more robust approximation tailored for highly ill-posed tasks. (3) Demonstrating that Dilack outperforms existing ZS DMs and classical methods in highly ill-posed IR tasks encountered in modern imaging systems with large and complex kernel degradations, across both synthetic and real-world datasets.

## 2 BACKGROUND

**Diffusion models.** We utilize a task-agnostic diffusion model capable of generating images, trained on numerous images. Diffusion models (Sohl-Dickstein et al., 2015; Song & Ermon, 2019), particularly

those under the variance-preserving framework such as DDPM (Ho et al., 2020), transform data distributions into Gaussian through a linear stochastic differential equation (SDE) (Song et al., 2020). This forward process incrementally adds noise over finite steps:

$$dx = -\frac{\beta(t)}{2}\mathbf{x}dt + \sqrt{\beta(t)}d\mathbf{w} \tag{3}$$

where $\beta(t)$ denotes the variance schedule, and $\mathbf{w}$ is standard Brownian motion. As the forward process evolves, the data distribution, initially $\mathbf{x}(0) \sim p_{\text{data}}$, gradually reaches a Gaussian state at $\mathbf{x}(1)$. The reverse process restores the original data distribution by reversing the noising SDE via applying denoising score matching techniques to reduce noise levels progressively:

$$d\mathbf{x} = \left[-\frac{\beta(t)}{2}\mathbf{x} - \beta(t)\nabla_{\mathbf{x}_t}\log p_t(\mathbf{x}_t)\right]dt + \sqrt{\beta(t)}d\bar{\mathbf{w}} \tag{4}$$

where $t$ is the DM sampling iteration that goes from $T$ to 0. In Eq. (4), $\nabla_{\mathbf{x}_t}\log p_t(\mathbf{x}_t)$ can be approximated by the pre-trained score function $\mathbf{s}_\theta(\mathbf{x}_t, t)$.

**Zero-shot diffusion models for IR.** In ZS DMs for solving low-level vision tasks where singular value decomposition (SVD)-based methods (Kawar et al., 2021; 2022; Wang et al., 2022; Cao et al., 2024) are impractical, recent studies (Chung et al., 2023b;a; Tang et al., 2024; Yang et al., 2024; Mardani et al., 2023; Zhu et al., 2023; Song et al., 2022; Garber & Tirer, 2024; Rout et al., 2024) have proposed to replace the score function in Eq. (4) with Bayesian framework. In this framework, $p(\mathbf{x})$ acts as the *prior*, with updates from the *posterior* $p(\mathbf{x}|\mathbf{y})$ computed using:

$$\nabla_{\mathbf{x}_t}\log p_t(\mathbf{x}_t|\mathbf{y}) = \nabla_{\mathbf{x}_t}\log p_t(\mathbf{x}_t) + \nabla_{\mathbf{x}_t}\log p_t(\mathbf{y}|\mathbf{x}_t). \tag{5}$$

In Eq. 5, after replacing $\nabla_{\mathbf{x}_t}\log p_t(\mathbf{x}_t)$ with the score estimate $\mathbf{s}_{\theta^*}(\mathbf{x}_t, t)$, the posterior mean from $p(\mathbf{x}_0|\mathbf{x}_t)$ can be approximated by factorizing $p_t(\mathbf{y}|\mathbf{x}_t)$ using Tweedie's formula (Efron, 2011):

$$\hat{\mathbf{x}}_{0|t} \simeq \frac{1}{\sqrt{\bar{\alpha}(t)}}(\mathbf{x}_t + (1 - \bar{\alpha}(t))\mathbf{s}_{\theta^*}(\mathbf{x}_t, t)). \tag{6}$$

This leads to the following equation for solving Eq. (1), approximated errors using the Jensen gap (Gao et al., 2017; Chung et al., 2023b):

$$\nabla_{\mathbf{x}_t}\log p(\mathbf{y}|\mathbf{x}_t) \simeq \nabla_{\mathbf{x}_t}\log p(\mathbf{y}|\hat{\mathbf{x}}_{0|t}) \tag{7}$$

where $\hat{\mathbf{x}}_{0|t}$ is the posterior mean of $p(\mathbf{x}_0|\mathbf{x}_t)$ obtained during the DDPM reverse diffusion sampling process starting from time step 0. By differentiating $p(\mathbf{y}|\mathbf{x}_t)$ with respect to $\mathbf{x}$, we can obtain the final sampling process for IR (Chung et al., 2022a; 2023b):

$$\nabla_{\mathbf{x}_t}\log p_t(\mathbf{x}_t|\mathbf{y}) \simeq \mathbf{s}_\theta(\mathbf{x}_t, t) - \rho\nabla_{\mathbf{x}_t}\mathcal{L}(\mathbf{x}; \mathbf{y}), \quad \rho \triangleq 1/\sigma^2 = \text{step size}. \tag{8}$$

Here, $\mathbf{s}_\theta(\mathbf{x}_t, t)$ serves as the regularization term $\mathcal{R}(\mathbf{x})$, while $\mathcal{L}(\mathbf{x}; \mathbf{y})$ is the fidelity term in Eq. 2.

The existing ZS DMs guide the diffusion process using the approximation of $\mathcal{L}(\mathbf{x}; \mathbf{y})$. Most ZS DM methods (Chung et al., 2023b;a; Tang et al., 2024; Yang et al., 2024; Mardani et al., 2023; Rout et al., 2024; Song et al., 2024) utilizing least-squares (LS) fidelity in Eq. (8) solve general image inverse problems:

$$\mathcal{L}(\hat{\mathbf{x}}_{0|t}; \mathbf{y}) := \mathcal{L}_{LS} = \|\mathbf{y} - \boldsymbol{A}\hat{\mathbf{x}}_{0|t}\|_2^2. \tag{9}$$

Some of recent methods (Zhu et al., 2023; Song et al., 2022; Garber & Tirer, 2024) integrate a Wiener deconvolution (Wiener, 1949) as a pseudo-inverse operator $\boldsymbol{A}^\dagger$ to enhance performance of Eq. (8):

$$\mathcal{L}(\hat{\mathbf{x}}_{0|t}; \mathbf{y}) := \mathcal{L}_{Pi} = \|\boldsymbol{A}^\dagger\mathbf{y} - \boldsymbol{A}^\dagger\boldsymbol{A}\hat{\mathbf{x}}_{0|t}\|_2^2. \tag{10}$$

DiffPIR (Zhu et al., 2023) enhances performance in deblurring by using $\mathcal{L}_{Pi}$ and $\Pi$GDM (Song et al., 2022) integrates $\mathcal{L}_{Pi}$ with the vector-Jacobian product to enhance consistency between measurements and results. DDPG (Garber & Tirer, 2024) utilizes both $\mathcal{L}_{Pi}$ (10) and $\mathcal{L}_{LS}$ (9), achieving significant improvements in balancing fidelity and perceptual quality. Note that the prevalent text-conditioned ZS DMs (Radford et al., 2021; Couairon et al., 2023; Luo et al., 2023; Yu et al., 2024) are not suitable for our tasks as they require decipherable measurements for reliable text extraction.

**Highly ill-posed real-world inverse problems with large and complex kernel degradations.** *(i) Lensless imaging*: Mask-based lensless cameras utilize phase or amplitude masks close to the

sensor instead of lenses in conventional cameras, achieving low-cost and compact designs based on compressive sensing techniques (Antipa et al., 2018; Asif et al., 2016; Boominathan et al., 2022). To accomplish miniaturization and multiplexed imaging, these cameras feature large and complex patterned point spread functions (PSFs) with a large aperture that match the size of the measurements, as shown in Fig. 2 and 3. Moreover, these measurements are limited (*i.e.*, cropped) due to the physical constraints of the sensor size (Poudel & Nakarmi, 2024). Also, they are affected by an overly idealized shift-invariance assumption, shooting environment, and sensor intrinsic noise, resulting in significant ill-posed problems that require a thorough approach for raw measurement reconstruction (Boominathan et al., 2022). Note that in this paper, the term "lensless imaging" is restricted to mask-based lensless camera raw reconstruction. Appendix E provides a detailed explanation of mask-based cameras, their measurements, and the need for zero-shot lensless imaging.

*(ii) Large motion deblurring*: As modern cameras aim for ultra-high resolution and image quality, managing motion blur becomes more difficult. The finer details captured by these cameras magnify the effects of motion blur, particularly when zoomed in. Higher pixel density makes images more sensitive to even slight camera shake or movement. Moreover, capturing high-resolution images often requires more light, and in low-light conditions, slower shutter speeds increase motion blur, further exacerbated by higher ISO settings. The scenario with large motion blur kernels, such as Fig. 2 and 3, highlights these issues, emphasizing the need for innovative solutions.

# 3 ANALYSES ON DIFFUSION MODELS FOR HIGHLY ILL-POSED RESTORATION

We investigated two diffusion methods for image restoration, DPS (Chung et al., 2023b) and Diff-PIR (Zhu et al., 2023), with the different existing data fidelity terms $\mathcal{L}_{LS}$ in Eq. (9) and $\mathcal{L}_{Pi}$ in Eq. (10), respectively, for highly ill-posed inverse problems of large motion deblurring and modern lensless imaging to observe the behaviors of them with challenging large and complex kernels.

**Challenge 1: Large motion deblurring.** We assess the influence of large and complex kernel by simulating a motion blur kernel, focusing on its size and complexity as shown in Figure 2(a). In the first row of Figure 2, when utilizing a $64^2$ kernel with an intensity of 0.5 that are equivalent settings in DPS (Chung et al., 2023b) and DiffPIR (Zhu et al., 2023), both methods achieved excellent results that are close to the ground truth (GT). However, increasing the kernel size to $256^2$ results in notable performance drop for DPS while much less performance drop for DiffPIR. In the second row of Figure 2, the kernel size is set to $256^2$, and the intensity is increased to 1.0, simulating a larger kernel with increased nonlinearity, DPS exhibited a notable loss in consistency with the GT, while DiffPIR, guided by $A^\dagger$ (*i.e.*, Wiener deconvolution in Appendix B), achieved much better restoration, even though it lost some fidelity as compared to the cases with smaller motion. To analyze these empirical results, we examine the sensitivity and stability issues caused by high condition numbers

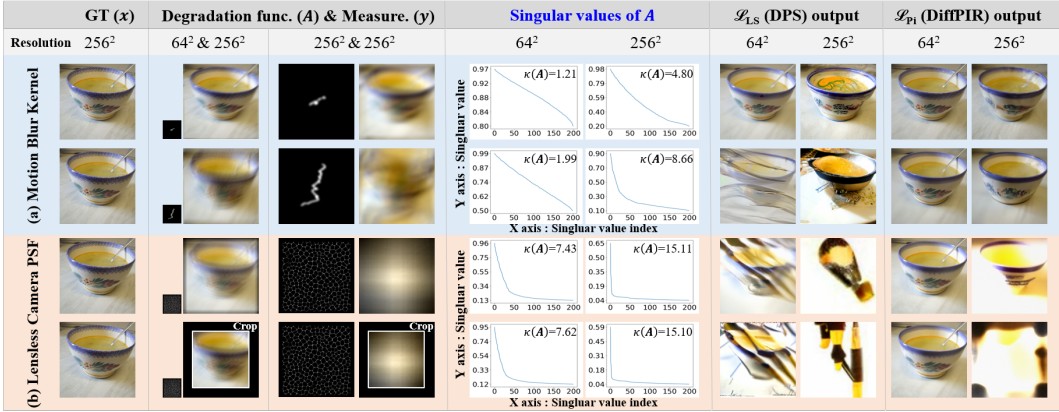

Figure 2: Analysis on (a) motion blur kernels and (b) lensless camera PSFs of varying sizes and complexity. As the kernel size increases, the non-uniformity of $A$'s singular values increases, leading to deteriorated image reconstruction performance. Using $\mathcal{L}_{LS}$ show a notable decline in performance. In more complex kernel patterns (b) with kernel size $256^2$, using $\mathcal{L}_{Pi}$ also severely underperforms.

and non-uniform singular values (Golub & Kahan, 1965; Brand, 2006; Karras et al., 2019), as shown in the 'Singular values of $\boldsymbol{A}$' column in Figure 2.

**Challenge 2: Modern lensless imaging.** We now analyze the effectiveness of $\boldsymbol{A}^\dagger$ on the point spread functions (PSFs) from real lensless cameras, which have much more complex features compared to large motion blur. Our analysis compares ideal measurements obtained by convolution with the kernel (*third row*) to more realistic simulations involving a 12.1% crop in each dimension (*fourth row*). As shown in Figure 2(b), DPS with $\mathcal{L}_{LS}$ fidelity faces significant challenges with the lensless kernel $\boldsymbol{A}$, resulting in poor performance under both ideal and realistic conditions, worsening as the PSF size increases. In contrast, while DiffPIR using $\boldsymbol{A}^\dagger$ performs better than DPS with normal PSF sizes, it still produces completely incorrect reconstruction outputs for large PSF sizes. This performance drop is due to the poor quality of the pseudo-inverse in high condition number scenarios.

**Analysis 1: LS fidelity vs. Pi fidelity.** The condition number of $\boldsymbol{A} \in \mathbb{R}^{m \times n}$ is defined as $\kappa(\boldsymbol{A}) = \frac{\sigma_{\max}}{\sigma_{\min}}$, where $\sigma_{\max}$ and $\sigma_{\min}$ are the largest and smallest singular values, respectively. When large kernels cause significant non-uniformity in the singular values as shown in the graphs of Fig 2, the condition number increases, reducing the effective rank, $\mathrm{rank}_\eta(\boldsymbol{A})$, defined as the number of singular values above a threshold $\eta$. For the (a) motion blur kernel, the condition number increases from 1.21 to 4.80 as the first row kernel size increases from 64 to 256. Similarly, for the (b) lensless camera PSF, it increases from 7.43 to 15.11, demonstrating that larger and more complex kernels result in higher condition numbers. A high condition number implies greater sensitivity to perturbations, as expressed in the inequality below (see Appendix A for details):

$$\|\Delta \mathbf{x}\| \le \|\boldsymbol{A}^{-1}\| \|\mathbf{y}\| (\delta + \epsilon \kappa(\boldsymbol{A})). \tag{11}$$

Therefore, the fidelity $\mathcal{L}_{LS}$ in Eq. (9) yields unstable data fidelity at each iteration, affecting quality. A suitable pseudo-inverse $\boldsymbol{A}^\dagger$ (*e.g.*, Wiener deconvolution) significantly aids in solving image inverse problems (Tirer & Giryes, 2020; Garber & Tirer, 2024). If $\boldsymbol{A}$ is imperfect with the error $\delta \boldsymbol{A}$, the solution with the fidelity $\mathcal{L}_{Pi}$ in Eq. (10) will be perturbed as $\mathbf{x} + \delta \mathbf{x} \approx (\boldsymbol{A}^\dagger - \boldsymbol{A}^\dagger \delta \boldsymbol{A} \boldsymbol{A}^\dagger + O(\|\delta \boldsymbol{A}\|^2))\mathbf{y}$. Focusing on first-order terms, the change in $\mathbf{x}$ due to $\delta \boldsymbol{A}$ is $\delta \mathbf{x} \approx -\boldsymbol{A}^\dagger \delta \boldsymbol{A} \boldsymbol{A}^\dagger \mathbf{y}$. This illustrates how the pseudo-inverse $\boldsymbol{A}^\dagger$ mediates the impact of $\boldsymbol{A}$ with high conditional number by removing the direct influence of the conditional number in the upper bound. While the Pi fidelity with Eq. (10) seems to work well even for the forward models with high conditional number, it is important to ensure if we can obtain a reliable pseudo-inverse $\boldsymbol{A}^\dagger$.

These comparisons seem consistent with the prior work on two fidelity terms of LS and Pi guidance in the setting of image restoration with classical regularizers. For the forward model $\boldsymbol{A}$ with all singular values less than 1, (Tirer & Giryes, 2020; Garber & Tirer, 2024) reported that $\mathcal{L}_{Pi}$ tends to exhibit smaller bias but higher variance compared to $\mathcal{L}_{LS}$ in both noiseless and moderately noisy cases. Consequently, $\mathcal{L}_{Pi}$ generally achieves a smaller mean squared error (MSE) than $\mathcal{L}_{LS}$. Therefore, as shown in Figure 2, regardless of different kernel sizes and non-uniform singular values, $\mathcal{L}_{Pi}$ generally exhibits better restoration performance than $\mathcal{L}_{LS}$, as shown in DiffPIR original paper.

**Remark 1.** *$\mathcal{L}_{Pi}$ fidelity in Eq. (10), which leverages the pseudo-inverse, generally outperforms $\mathcal{L}_{LS}$ fidelity in Eq. (9). The latter is effective in noiseless or moderately noisy cases under reasonable conditions reasonable conditions regarding the non-uniformity of $\boldsymbol{A}$'s singular values of the forward model.*

**Analysis 2: Fidelity vs. global regularizer.** Here we have tackled highly ill-posed inverse problems with very high condition numbers, so it is reasonable to start our investigation from the Pi fidelity $\mathcal{L}_{Pi}$ rather than the LS fidelity $\mathcal{L}_{LS}$. Here we focus on the relationship between the fidelity term and the regularization term in Eq. (2). For classical regularization such as total variation with the Pi fidelity, the loss function becomes

$$\|\boldsymbol{A}^\dagger \mathbf{y} - \boldsymbol{A}^\dagger \boldsymbol{A}\mathbf{x}\|_2^2 + \lambda \|\nabla \mathbf{x}\|_2^2 = \sum_j \{[\boldsymbol{A}^\dagger \mathbf{y} - \boldsymbol{A}^\dagger \boldsymbol{A}\mathbf{x}]_j^2 + \lambda [\nabla \mathbf{x}]_j^2\} \tag{12}$$

where $[\cdot]_j$ is the $j$th element of an input vector. Then, the trade-off between the fidelity and the regularizer will be controlled *locally*. In other words, for some $j$ indices, the fidelity term is minimized while for other $j$ indices, the regularizer term will be minimized simultaneously. In the meanwhile, for diffusion model prior $\mathcal{R}(\mathbf{x})$ with the Pi fidelity such as

$$\sum_j \{[\boldsymbol{A}^\dagger \mathbf{y} - \boldsymbol{A}^\dagger \boldsymbol{A}\mathbf{x}]_j^2\} + \lambda \mathcal{R}(\mathbf{x}), \tag{13}$$

the trade-off between the fidelity and the regularizer is not controlled locally, but controlled *globally*. This particular relationship may not be favorable for highly ill-posed inverse problems where there

may be usually very high errors for some indices $j$ since these errors can be considered as a structure in an image for a generative model like DMs, leading to completely different images as illustrated in Figure 2, especially for large and complex kernels. This argument seems consistent with the results in (Bora et al., 2017) where using a hybrid regularizer with (global) generative prior and (local) classical regularizer was advantageous in performance for the cases with challenging small measurements.

# 4 PROPOSED METHOD: DILACK

Based on the analysis in Sec. 3, we introduce **Dilack**, a zero-shot diffusion model for large and complex kernel degradations, providing solutions beyond the reach of existing methods.

## 4.1 REGULARIZED FIDELITY: PSEUDO-INVERSE ANCHOR FOR CONSTRAINING (PiAC)

Consider $\mathcal{L}_{Pi}$ in Eq. 10, $\|\boldsymbol{A}^\dagger \mathbf{y} - \boldsymbol{A}^\dagger \boldsymbol{A} \hat{\mathbf{x}}_{0|t}\|_2^2$, where $\boldsymbol{A}^\dagger \mathbf{y}$ represents an analytic solution like Wiener deconvolution, and $\boldsymbol{A}^\dagger \boldsymbol{A}$ projects onto the subspace spanned by $\boldsymbol{A}^\dagger \mathbf{y}$. For well-posed problems, $\boldsymbol{A}^\dagger$ is equivalent to the exact inverse $\boldsymbol{A}^{-1}$, making $\boldsymbol{A}^\dagger \mathbf{y}$ the true estimate $\mathbf{x}$ and $\boldsymbol{A}^\dagger \boldsymbol{A}$ the identity matrix. However, for highly ill-posed problems, these terms fail to preserve fidelity effectively. To address this, we propose a new fidelity term to approximate $\mathcal{L}_{Pi}$ under such conditions.

**Pseudo-inverse anchor.** The first term, $\boldsymbol{A}^\dagger \mathbf{y}$, serves as an anchor to enforce fidelity to the measurement $\mathbf{y}$. However, for highly ill-posed problems, the pseudo-inverse solution $\boldsymbol{A}^\dagger \mathbf{y}$ is insufficient. We replace it with a regularized solution $\tilde{\mathbf{x}}^*$, selecting a total variation (TV)-regularized solution (Rudin et al., 1992) obtained via Alternating Direction Method of Multipliers (ADMM) (Beck & Teboulle, 2009; Boyd et al., 2011; Yang et al., 2013) (Appendix C). This leads to the approximate fidelity term $\|\tilde{\mathbf{x}}^* - \boldsymbol{A}^\dagger \boldsymbol{A} \hat{\mathbf{x}}_{0|t}\|_2^2$. The optimization starts with an initial point of 0 for the first $\tilde{\mathbf{x}}^*$, updated $G - 1$ times using normalized intermediate sampling outputs, where $G$ is the number of initializations. The TV regularizer weight $\lambda_t$ decreases over re-initialization steps, starting at $\lambda_{T-1} = 10^{-7}$ (Appendix G.6). If $G = 1$, the initial $\tilde{\mathbf{x}}^*$ is reused without updates.

For well-posed problems, $\boldsymbol{A}^\dagger \boldsymbol{A}$ acts as an exact identity, and for mildly ill-posed problems, it approximates the identity. In highly ill-posed cases, however, $\boldsymbol{A}^\dagger \boldsymbol{A}$ loses significant information, making it unsuitable as a fidelity term. To address this, we approximate $\boldsymbol{A}^\dagger \boldsymbol{A}$ as the identity, leading to the simplified fidelity term $\|\tilde{\mathbf{x}}^* - \hat{\mathbf{x}}\|_2^2$ in posterior sampling, where $x$ becomes $x_t$. This enables applying DPS Theorem 1, resulting in $\|\tilde{\mathbf{x}}^* - \hat{\mathbf{x}}_{0|t}\|^2$.

The difference between $\mathcal{L}_{Pi}$ and $\mathcal{L}_{PiAC}$ lies in two aspects: $\mathcal{L}_{Pi}$ uses $\boldsymbol{A}^\dagger \mathbf{y}$ as a minimum norm solution, while $\mathcal{L}_{PiAC}$ employs a TV-regularized solution. Additionally, $\mathcal{L}_{Pi}$ directly applies $\boldsymbol{A}^\dagger \boldsymbol{A}$, whereas $\mathcal{L}_{PiAC}$ approximates it as the identity. As a result, $\mathcal{L}_{Pi}$ evaluates fidelity within the projected space, while $\mathcal{L}_{PiAC}$ accounts for the entire space, including the null space filled by TV regularization. Our Dilack sacrifices the aspect of measuring in the projected spaces for highly ill-posed inverse problems, but instead we approximate it by comparing the values in the null space that was filled by TV regularization.

**Pseudo-inverse anchor for constraining fidelity.** Therefore, the proposed Pseudo-inverse Anchor for Constraining (PiAC) fidelity loss is:

$$\mathcal{L}_{\text{PiAC}} = \|\tilde{\mathbf{x}}^* - \hat{\mathbf{x}}_{0|t}\|_2^2 \tag{14}$$

where $\hat{\mathbf{x}}_{0|t}$ is the posterior mean $\mathbb{E}_{p(\mathbf{x}_0|\mathbf{x}_t)}[\mathbf{x}_0]$ by Eq. (6) as defined in Eq (7). Then, we reformulate the gradient of the log likelihood from Eq. (5) as follows:

$$\nabla_{\mathbf{x}_t} \log p_t(\mathbf{x}_t|\mathbf{y}) \simeq \mathbf{s}_{\theta^*}(\mathbf{x}_t, t) - \rho \nabla_{\mathbf{x}_t} \mathcal{L}_{\text{PiAC}} \tag{15}$$

where $\rho \triangleq 1/\sigma^2$ functions as guidance weight and $\mathcal{L}_{\text{PiAC}}$ acts as an approximate $\nabla_{\mathbf{x}_t} \log p_t(\mathbf{y}|\mathbf{x}_t)$.

$\mathcal{L}_{\text{PiAC}}$ addresses the shortcomings of traditional fidelity measures by incorporating them into the log likelihood gradient, ensuring adherence to both observed measurements and the non-linear estimate, effectively filling gaps in data fidelity. Particularly useful when Wiener deconvolution fails with large and complex kernel degradations, it guarantees consistency in reconstruction performance. Note that while other ZS DMs (Chung et al., 2023b; Zhu et al., 2023; Garber & Tirer, 2024) adjust $\rho$ for each dataset and task, we standardize $\rho = 1$ across all cases, making our approach more generalizable.

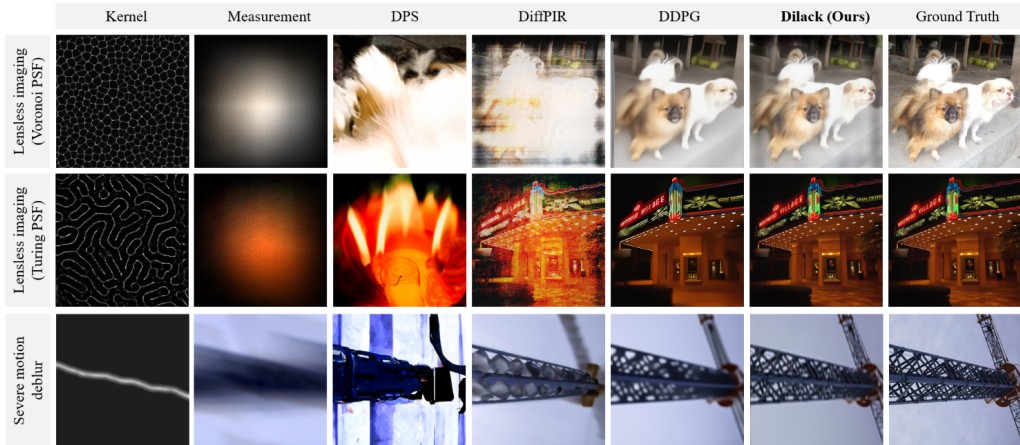

Figure 3: Qualitative results of our zero-shot diffusion model, Dilack, on ImageNet.

## 4.2 MASKED FIDELITY: REGION-OF-INTEREST (ROI) MASK FOR LOCALIZED PiAC

**Dynamic ROI mask design.** Model-based optimization approaches in PiAC fidelity (Boyd et al., 2011; Beck & Teboulle, 2009) enhance or reduce fidelity point-wise across regions. Integrating diffusion priors adds complexity, as their global application during sampling may overfit noise and artifacts in $\tilde{\mathbf{x}}^*$, degrading realism. This occurs because $\tilde{\mathbf{x}}^*$ is sparsely updated $G-1$ times or remains static when $G=1$. To address this, we introduced ROI masking to selectively control these effects, balancing fidelity and perceptual quality while leveraging the generative strengths of the diffusion model. To compute the ROI and adaptively mitigate the local effects of $\mathcal{L}_{PiAC}$, we construct a 2D binary mask $\mathcal{M}_t$, which is dynamically updated at each iteration $t$. Let the TV-regularized solution $\tilde{\mathbf{x}}^*$ and the current sampled image $\hat{\mathbf{x}}_{0|t}$, each with dimensions $H \times W$, be divided into $\psi \times \psi$ non-overlapping patches. In our implementation, we set $\psi = 32$, resulting in $N = H \times W/\psi^2 = 64$ patches for $H = W = 256$. Define the indices of the patches on the image grid as $i, j$, where $i, j \in \{1, \ldots, H/\psi\}$. Let $P^{(i,j)}$ denote the patch located at the $(i, j)$-th position of the image grid. The binary mask for each patch at iteration $t$, denoted as $\mathcal{M}_t^{(i,j)}$, is determined based on the differences between the corresponding patches in $\tilde{\mathbf{x}}^*$ and $\hat{\mathbf{x}}_{0|t}$. Specifically, $\mathcal{M}_t^{(i,j)}$ is defined as:

$$\mathcal{M}_t^{(i,j)} = \begin{cases} 1, & \text{if } \mathcal{D}_{\text{sum}}^{(i,j)}(\tilde{\mathbf{x}}^*, \hat{\mathbf{x}}_{0|t}) \geq \Omega_\nu \left[ \mathcal{D}_{\text{sum}}(\tilde{\mathbf{x}}^*, \hat{\mathbf{x}}_{0|t}) \right], \\ 0, & \text{otherwise}, \end{cases} \tag{16}$$

where $\mathcal{D}_{\text{sum}}^{(i,j)}(\tilde{\mathbf{x}}^*, \hat{\mathbf{x}}_{0|t})$ computes the sum of $\mathcal{L}_1$-norm differences for all pixel values within the patch $P^{(i,j)}$, and $\Omega_\nu [\cdot]$ represents the $\nu$-th percentile of all $\mathcal{D}_{\text{sum}}^{(i,j)}$ values across the entire image grid.

The complete binary mask for iteration $t$, $\mathcal{M}_t$, is then obtained by combining all patch-wise binary masks $\mathcal{M}_t = \bigcup_{i=1}^{H/\psi} \bigcup_{j=1}^{W/\psi} \mathcal{M}_t^{(i,j)}$, where $\mathcal{M}_t^{(i,j)} \in \{0, 1\}$ for all $i, j$. We set a patch size of $32 \times 32$ with an 80th percentile threshold for lensless imaging and a 30th percentile threshold for large motion deblurring, which yields optimal performance (Appendix F.1). For regions where the mask value is 1, $\mathcal{L}_{\text{PiAC}}$ effectively maintains spatial consistency. This prevents the generation of random eccentric images from the pre-trained diffusion model. Conversely, in regions where the mask value is 0, it exclusively modulates beneficial local elements for IR. It utilizes the diffusion prior $\mathbf{s}_{\theta^*}(\mathbf{x}_t, t)$ without $\mathcal{L}_{\text{PiAC}}$.

**Algorithm 1** Dilack

**Require:** $A, \mathbf{y}, \lambda_{T-1}, \rho, T, \mathcal{C}, G, s_\theta(\cdot, t)$, and $\{\tilde{\sigma}_t\}_{t=1}^T$
1: $\mathbf{x}_T \sim \mathcal{N}(\mathbf{0}, \mathbf{I})$
2: **for** $t = T - 1$ **to** $0$ **do**
3:     $\hat{s} \leftarrow s_\theta(\mathbf{x}_t, t)$
4:     $\hat{\mathbf{x}}_0 \leftarrow \frac{1}{\sqrt{\bar{\alpha}_t}}(\mathbf{x}_t + (1 - \bar{\alpha}_t)\hat{s})$
5:     $z \sim \mathcal{N}(\mathbf{0}, \mathbf{I})$
6:     $\mathbf{x}_{t-1}' \leftarrow \frac{\sqrt{\alpha_t}(1-\bar{\alpha}_{t-1})}{1-\bar{\alpha}_t}\mathbf{x}_t + \frac{\sqrt{\bar{\alpha}_{t-1}}\beta_t}{1-\bar{\alpha}_t}\hat{\mathbf{x}}_{0|t} + \tilde{\sigma}_t z$
7:     **if** $t \in C$ **then**
8:         $\tilde{\mathbf{x}}^* \in \arg\min_\mathbf{x} \|\mathbf{y} - A\mathbf{x}\|_2^2 + \lambda_t \text{TV}(\mathbf{x})$ *// Classical*
        *TV-regularized optimization starts with initial values of 0 and is re-initialized $G - 1$ times using intermediate sampling outputs.*
9:         $\mathbf{x}_{t-1} \leftarrow \mathbf{x}_{t-1}' - \rho\mathcal{M}_t \cdot \nabla_{\mathbf{x}_t}\|\tilde{\mathbf{x}}^* - \hat{\mathbf{x}}_{0|t}\|_2^2$
10:     **else**
11:         $\mathbf{x}_{t-1} \leftarrow \mathbf{x}_{t-1}'$
12:     **end if**
13: **end for**
14: **return** $\hat{\mathbf{x}}_{0|t}$

Table 1: Quantitative results of the zero-shot IR methods on **ImageNet**(*top*) and **FFHQ**(*bottom*), including lensless imaging and large motion deblurring. Note that large motion blur uses *relatively* simple kernel features, allowing existing methods using $A^\dagger$ to perform well, but Dilack shows comparable results and our mask-guided approach outperforms pure ADMM$_{TV}$ in all aspects.

| **ImageNet** | Lensless Imaging (w/ Voronoi) | Lensless Imaging (w/ Turing) | Large Motion Deblurring |
|---|---|---|---|
| Method | PSNR↑/SSIM↑/FID↓/LPIPS↓ | PSNR↑/SSIM↑/FID↓/LPIPS↓ | PSNR↑/SSIM↑/FID↓/LPIPS↓ |
| $A^\dagger \mathbf{y}$ (Wiener, 1949) | 13.17 / 0.274 / 241.33 / 0.606 | 13.12 / 0.288 / 252.38 / 0.589 | 15.80 / 0.421 / 170.4 / 0.616 |
| ADMM$_{TV}$ (Boyd et al., 2011) | 19.74 / 0.574 / **36.45** / 0.299 | 20.57 / 0.575 / **34.84** / 0.293 | 19.90 / 0.528 / 120.07 / 0.492 |
| DPS (Chung et al., 2023b) | 8.13 / 0.268 / 130.77 / 0.666 | 8.15 / 0.265 / 128.12 / 0.666 | 17.46 / 0.488 / **38.17** / **0.364** |
| DiffPIR (Zhu et al., 2023) | 11.22 / 0.448 / 153.97 / 0.479 | 10.66 / 0.248 / 152.84 / 0.576 | 21.04 / 0.511 / 61.97 / 0.414 |
| DDPG (Garber & Tirer, 2024) | 19.55 / 0.658 / 91.33 / 0.385 | 19.54 / 0.653 / 91.90 / 0.391 | 22.30 / 0.593 / 92.64 / 0.449 |
| **Dilack(ours)** | **22.88** / **0.773** / 41.54 / **0.250** | **24.94** / **0.798** / 35.61 / **0.225** | 20.99 / **0.612** / 77.46 / 0.420 |
| **FFHQ** | Lenssless Imaging (w/ Voronoi) | Lensless Imaging (w/ Turing) | Large Motion Deblurring |
| Method | PSNR↑/SSIM↑/FID↓/LPIPS↓ | PSNR↑/SSIM↑/FID↓/LPIPS↓ | PSNR↑/SSIM↑/FID↓/LPIPS↓ |
| $A^\dagger \mathbf{y}$ (Wiener, 1949) | 12.89 / 0.228 / 345.41 / 0.679 | 12.98 / 0.241 / 398.2 / 0.662 | 16.76 / 0.547 / 183.58 / 0.565 |
| ADMM$_{TV}$ (Boyd et al., 2011) | 19.63 / 0.491 / 54.89 / 0.367 | 20.28 / 0.488 / 54.98 / 0.362 | 21.32 / 0.620 / 125.02 / 0.459 |
| DPS (Chung et al., 2023b) | 9.98 / 0.362 / 76.55 / 0.564 | 9.96 / 0.361 / 71.52 / 0.561 | 17.46 / 0.488 / 38.17 / 0.364 |
| DiffPIR (Zhu et al., 2023) | 12.78 / 0.534 / 132.76 / 0.453 | 13.68 / 0.559 / 112.97 / 0.429 | 23.85 / 0.664 / **32.90** / **0.271** |
| DDPG (Garber & Tirer, 2024) | 13.68 / 0.535 / 135.72 / 0.441 | 13.85 / 0.539 / 130.12 / 0.440 | **26.15** / **0.763** / 69.36 / 0.288 |
| **Dilack(ours)** | **23.83** / **0.836** / **34.55** / **0.179** | **26.24** / **0.860** / **28.69** / **0.156** | 23.15 / 0.745 / 59.60 / 0.313 |

PSF (kernel)   Measure   $A^\dagger y$   ADMM-TV   DPS   DiffPIR   DDPG   Dilack(ours)   GT

Figure 4: Qualitative results of the synthetic lensless imaging with real Voronoi PSF.

**Total loss design.** By utilizing ROI masks to selectively activate the PiAC fidelity term locally, Dilack enhances consistency by emphasizing areas with large differences. It also generates realistic global content using the generative capabilities of the pre-trained diffusion model.

In conclusion, the loss design of Dilack strategically integrates multiple components to balance fidelity and perceptual quality in the reconstructed images. By integrating the dynamic ROI mask strategy with Eq. (15), the final posterior $p(x|y)$ is expressed in Eq. (17), with the detailed process outlined in **Algorithm 1**:

$$\nabla_{\mathbf{x}_t} \log p_t(\mathbf{x}_t|\mathbf{y}) \simeq \mathbf{s}_{\theta^*}(\mathbf{x}_t, t) - \rho \mathcal{M}_t \left[ \nabla_{\mathbf{x}_t} \|\tilde{\mathbf{x}}^* - \hat{\mathbf{x}}_{0|t}\|_2^2 \right]. \tag{17}$$

**Additional considerations.** Firstly, the fixed size of $\mathcal{M}_t$ can cause artifacts at patch boundaries in the reconstruction output due to its limited ability to fully capture the entire local area. To address this, we adopted the shifted window partition strategy from the Swin Transformer (Liu et al., 2021; Liang et al., 2021a), which overcomes the limitations of non-overlapping patch partitioning and reduces overall artifacts (Appendix D). Secondly, we applied skip step guidance $\mathcal{C}$ (Ding et al., 2023; Song et al., 2024) during the initial sampling phase to loosely align with Dilack fidelity, alleviating the large scalar disparity between $\mathcal{L}_{\text{PiAC}}$ and $\mathbf{s}_{\theta^*}(\mathbf{x}_t, t)$, which caused local artifacts in some restored images (Appendix D).

## 5 EXPERIMENTS

### 5.1 EXPERIMENTAL SETUP

In our study, we evaluated the performances of Dilack on two datasets, ImageNet (Deng et al., 2009) and FFHQ (Karras et al., 2019), both at $256^2$ resolution, with a validation set of 1,000 images from each dataset. For ImageNet, we used a task-agnostic pre-trained diffusion model from (Dhariwal & Nichol, 2021), and for FFHQ, the pre-trained model from (Chung et al., 2023b).

Degradation models included: (i) Synthetic lensless camera measurements are simulated by convolving zero-padded ground truth images ($512^2$) with real lensless camera PSFs of size $512^2$ in Fourier space. This is followed by cropping to $450^2$, achieving a cropping rate of 12.1% in each dimension, and re-padding to $512^2$ to mimic a real lensless camera system. We then utilize the central area of $256^2$ as a diffusion sampling input. (ii) Large motion blur degradation was simulated using $256^2$

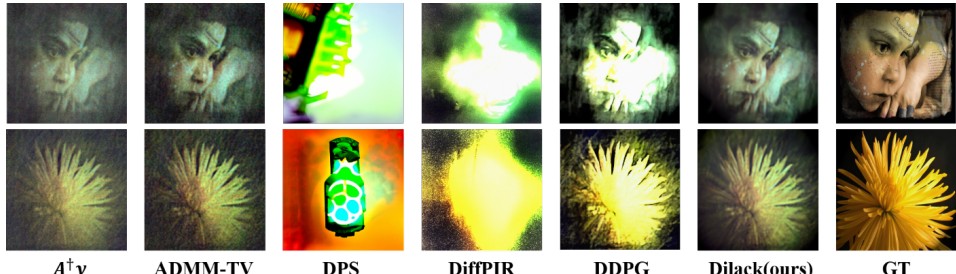

| $A^{\dagger}y$ | ADMM-TV | DPS | DiffPIR | DDPG | Dilack(ours) | GT |

Figure 6: Qualitative results of *real-world* mask-based lensless camera raw restoration.

kernels with an intensity of 1.0, sourced from an open-source repository[1], scaling up from prior studies' $61^2$ sized kernels and doubling the intensity, thereby intensifying the challenge in severe motion blur scenarios. A small amount of Gaussian noise is added to the measurement in each task.

We benchmarked our method against classical methods including the Wiener deconvolution ($A^{\dagger}\mathbf{y}$) (Wiener, 1949) and ADMM$_{TV}$ (Boyd et al., 2011), alongside state-of-the-art ZS DMs for IR, such as DPS (Chung et al., 2023b), DiffPIR (Zhu et al., 2023), and DDPG (Garber & Tirer, 2024). To evaluate image fidelity, we computed metrics PSNR and SSIM, and for perceptual quality, we utilized FID and LPIPS. Details of experimental setting are in Appendix F.

## 5.2 EXPERIMENTAL RESULTS

**Lensless imaging.** In mask-based camera raw reconstruction, Dilack outperforms both conventional model-based methods and leading zero-shot diffusion approaches in lensless imaging, as shown in Tab. 1 and Figs. 3, 4, and 5. Dilack excels by preserving fidelity and mitigating severe artifacts common in classical methods through its generative capabilities. This success is due to integrating diffusion priors with skip-step guided $\mathcal{L}_{\text{PiAC}}$ and our mask-guided locality, which enhances spatial consistency, reduces erratic outputs from $s_{\theta}$, and improves local details vital for effective image reconstruction. This demonstrates Dilack's strong performance in severely ill-posed problems.

**Large motion deblurring.** In the large motion deblurring task, Dilack shows comparable performance, as shown in Tab. 1 and Fig. 3. It is noteworthy that motion blur kernels, compared to lensless camera PSFs, have relatively simpler kernel structures in their features. This results in fewer instances of the exploding phenomena observed with existing DM methods in lensless imaging. Consequently, the Wiener filter-based pseudo-inverse $A^{\dagger}$ proves to be somewhat effective. This underscores the significant impact of kernel complexity, not merely size, on the performance of image inverse problems, as discussed in Sec. 3. Nonetheless, in the large motion deblurring task, Dilack shows comparable results, and our mask-guided approach outperforms ADMM$_{TV}$ in all aspects.

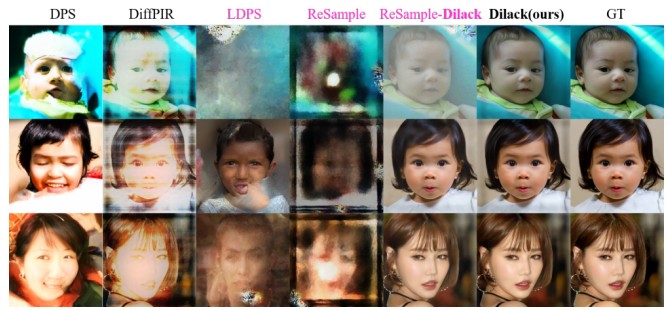

| DPS | DiffPIR | LDPS | ReSample | ReSample-Dilack | Dilack(ours) | GT |

Figure 5: Results of lensless imaging on FFHQ.

To sum up the synthetic experimental results in lensless imaging and large motion deblurring, our mask-guided approach performs better as the ill-posedness becomes more severe. More comparative qualitative outputs of two tasks are in Appendix G.18 and G.16.

## 5.3 FURTHER EXPERIMENTS

***Real-world* lensless imaging.** We utilized a custom-built mask-based lensless camera with a Voronoi pattern

Table 2: Quantitative results and ablation studies on the *real* lensless camera dataset.

| **MirFlickr-lensless** Method | *Real* Lensless Imaging PSNR↑/SSIM↑/FID↓/LPIPS↓ |
|---|---|
| $A^{\dagger}\mathbf{y}$ (Wiener, 1949) | 12.29 / 0.175 / 322.79 / 0.666 |
| ADMM$_{TV}$ (Boyd et al., 2011) | 13.05 / 0.225 / 312.09 / 0.629 |
| DPS (Chung et al., 2023b) | 6.97 / 0.182 / **273.02** / 0.739 |
| DiffPIR (Zhu et al., 2023) | 8.30 / 0.276 / 302.02 / 0.716 |
| DDPG (Garber & Tirer, 2024) | 10.45 / 0.221 / 293.78 / 0.655 |
| PiAC | 12.80 / 0.304 / 299.56 / 0.603 |
| PiAC w/ random mask | 12.87 / 0.323 / 295.04 / 0.593 |
| PiAC w/ ROI mask(**Dilack**) | **13.47** / **0.326** / 290.54 / **0.584** |

[1]https://github.com/LeviBorodenko/motionblur

hardware mask to capture real lensless measurements, demonstrating our approach's robustness and generalization capabilities. We displayed images from the MirFlickr (Huiskes & Lew, 2008) dataset on a screen and captured them, fitting the results to the ground truth for evaluation. Experimental results show that Dilack outperforms other methods in *real* lensless imaging, as presented in Tab. 2 and Fig. 6, marking the first application of ZS DMs to real lensless imaging. We will soon release this lensless raw dataset, dubbed 'MirFlickr-lensless' publicly. Note that real motion blur kernels do not naturally exist, making experiments with actual data impractical. Details on the lensless camera and real measurements are in Appendix E, with more qualitative outputs in Appendix G.18.

**Additional studies on latent diffusion models.** We extended our experiments to evaluate the effectiveness of Dilack in highly ill-posed kernel degradation settings, using SOTA LDM methods LDPS (Rout et al., 2024) and ReSample (Song et al., 2024) on lensless imaging (Turing PSF). Due to the absence of pre-trained unconditional LDMs on ImageNet, the experiments were conducted on the FFHQ dataset. In Tab. 3, ReSample-Dilack, which applies Dilack to ReSample, replaces the original gradient descent optimization of $\hat{\mathbf{z}}_0$ using $\mathcal{L}_{LS}$ with optimization based on $\mathcal{L}_{\text{PiAC}}$ and $\mathcal{M}_t$.

As shown in Tab. 3 and Fig. 5 (*pink-colored methods*), LDPS (Rout et al., 2024) and ReSample (Song et al., 2024) produce irregular results in the lensless imaging task, consistent with our expectations due to use of the $\mathcal{L}_{LS}$ in loss functions, as discussed in Sec. 3. In Tab. 3, ReSample-Dilack mitigates this issue by

Table 3: Additional results of *latent* diffusion models with Dilack on the FFHQ dataset.

| FFHQ | Lensless Imaging | Large Motion Deblurring |
|---|---|---|
| Method | PSNR↑/SSIM↑/FID↓/LPIPS↓ | PSNR↑/SSIM↑/FID↓/LPIPS↓ |
| LDPS (Rout et al., 2024) | 12.80 / 0.341 / 319.75 / 0.610 | 18.72 / 0.469 / 152.29 / 0.456 |
| ReSample (Song et al., 2024) | 12.22 / 0.336 / 408.17 / 0.620 | **24.09** / 0.686 / 98.38 / 0.319 |
| (**Dilack** in Tab. 1) | **26.24** / **0.860** / **28.69** / **0.156** | 23.15 / **0.745** / **59.60** / 0.313 |
| ReSample-**Dilack** | 22.91 / 0.770 / 77.49 / 0.239 | 21.69 / 0.699 / 140.09 / 0.366 |

using $\mathcal{L}_{\text{PiAC}}$ and $\mathcal{M}_t$, but it still performs worse across all metrics compared to the original Dilack (third row of Tab.3). This is due to the decoder's nonlinearity and nonconvexity in LDMs, complicating pixel-space solvers (Song et al., 2024). Similarly, in the large motion deblurring task, the original Dilack outperformed ReSample-Dilack on all metrics except PSNR. The algorithm for ReSample-Dilack is detailed in Appendix G.7. Note that additional experiments on LDMs, including ReSample-Dilack, under the *lighter* (normal) degradation settings are provided in Appendix G.2.

## 5.4 DISCUSSION

**Ablation studies.** *i*) The bottom three rows of Tab. 2 demonstrate the effectiveness of our ROI mask design by comparing PiAC fidelity *without* the ROI mask and *with* a random mask. Detailed results on synthetic datasets and qualitative comparisons of the ROI mask effectiveness are provided in Appendix G.1. Further studies on *ii*) masking ratio settings (Appendix G.3), *iii*) guidance scale adjustments (Appendix G.4), *iv*) skip step guidance settings (Appendix G.5), *v*) effect of re-initialization of $\tilde{\mathbf{x}}^*$ (Appendix G.6), *vi*) comparisons in *lighter* (normal) kernel cases (Appendix G.2), *vii*) effect of replacing PiAC with other denoisers (Appendix G.8), *viii*) comparisons with other PnP methods (Appendix G.9), *ix*) experiments on various optimization methods for $\tilde{\mathbf{x}}^*$ (Appendix G.11, G.12), and *x*) effect of the number of iterations in optimization (Appendix G.14) are in Appendix.

**Limitation.** Dilack employs a model-based algorithm iteration and ROI mask calculation, resulting in slightly slower processing times (DPS: 340 seconds vs. Dilack: 390 seconds), despite its demonstrated effectiveness (Appendix F.1). Nonetheless, we challenge the assumption that the latest diffusion models are universally optimal for most image inverse problems. In the context of modern imaging systems, their limitations are evident. As the first study to explore zero-shot diffusion models in this domain, our work offers a fresh perspective. Addressing efficiency concerns will be a focus of future work. Next-generation ultra-high-definition camera technology, capable of operating without a lens or under large motion blur effects, remains largely unexplored. Our research serves as a foundational study, laying the groundwork for future advancements in this field.

## 6 CONCLUSION

Existing zero-shot diffusion models for IR struggle with large and complex kernel degradations. To address these challenges, we propose Dilack, which revisits classical optimization and introduces a novel masked data fidelity with skip step-guided, ROI masked PiAC loss. This approach ensures localized regularization, improves local fidelity in each iteration, and thus delivers robust, realistic image restoration results in modern lensless imaging and large motion deblurring.

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

# Appendix

## A  PROOF OF SENSITIVITY TO PERTURBATIONS IN HIGH CONDITION NUMBER SYSTEMS

We begin by recalling fundamental linear algebra concepts for systems of equations. Consider the solution $\mathbf{x}$ to the linear system:

$$\boldsymbol{A}\mathbf{x} = \mathbf{y},$$

where $\boldsymbol{A}$ is invertible. This solution can be expressed as:

$$\mathbf{x} = \boldsymbol{A}^{-1}\mathbf{y}.$$

Now, introduce perturbations $\Delta\boldsymbol{A}$ to the matrix $\boldsymbol{A}$ and $\Delta\mathbf{y}$ to the vector $\mathbf{y}$, leading to the perturbed system:

$$(\boldsymbol{A} + \Delta\boldsymbol{A})(\mathbf{x} + \Delta\mathbf{x}) = \mathbf{y} + \Delta\mathbf{y}.$$

where $\Delta\boldsymbol{A}$ is a small perturbation to $\boldsymbol{A}$, and $\Delta\mathbf{y}$ is a small perturbation to $\mathbf{y}$. Expanding and rearranging, and noting that $\boldsymbol{A}\mathbf{x} = \mathbf{y}$, we get:

$$\boldsymbol{A}\Delta\mathbf{x} + \Delta\boldsymbol{A}\mathbf{x} + \Delta\boldsymbol{A}\Delta\mathbf{x} \simeq \Delta\mathbf{y}.$$

Neglecting the higher-order term $\Delta\boldsymbol{A}\Delta\mathbf{x}$, we simplify to:

$$\boldsymbol{A}\Delta\mathbf{x} + \Delta\boldsymbol{A}\mathbf{x} \simeq \Delta\mathbf{y}.$$

Assuming $\boldsymbol{A}$ is non-singular (invertible), we solve for $\Delta\mathbf{x}$:

$$\Delta\mathbf{x} \simeq \boldsymbol{A}^{-1}(\Delta\mathbf{y} - \Delta\boldsymbol{A}\mathbf{x}).$$

Taking norms on both sides and applying the triangle inequality, we obtain:

$$\|\Delta\mathbf{x}\| \leq \|\boldsymbol{A}^{-1}\|(\|\Delta\mathbf{y}\| + \|\Delta\boldsymbol{A}\mathbf{x}\|).$$

Using the sub-multiplicative property of norms, the term $\|\Delta\boldsymbol{A}\mathbf{x}\|$ can be bounded as:

$$\|\Delta\boldsymbol{A}\mathbf{x}\| \leq \|\Delta\boldsymbol{A}\|\|\mathbf{x}\|.$$

Thus, we have:

$$\|\Delta\mathbf{x}\| \leq \|\boldsymbol{A}^{-1}\|(\|\Delta\mathbf{y}\| + \|\Delta\boldsymbol{A}\|\|\mathbf{x}\|).$$

Substituting $\|\Delta\mathbf{y}\| \leq \delta\|\mathbf{y}\|$ and $\|\Delta\boldsymbol{A}\| \leq \epsilon\|\boldsymbol{A}\|$, where $\Delta$ and $\epsilon$ are small constants representing the relative perturbation magnitudes, gives:

$$\|\Delta\mathbf{x}\| \leq \|\boldsymbol{A}^{-1}\|(\delta\|\mathbf{y}\| + \epsilon\|\boldsymbol{A}\|\|\mathbf{x}\|).$$

Recalling that $\mathbf{x} = \boldsymbol{A}^{-1}\mathbf{y}$, this simplifies to:

$$\|\Delta\mathbf{x}\| \leq \|\boldsymbol{A}^{-1}\|\|\mathbf{y}\|(\delta + \epsilon\|\boldsymbol{A}\|\|\boldsymbol{A}^{-1}\|).$$

Recognizing that $\|\boldsymbol{A}\|\|\boldsymbol{A}^{-1}\| = \kappa(\boldsymbol{A})$, where $\kappa(\boldsymbol{A})$ is the condition number, we conclude:

$$\|\Delta\mathbf{x}\| \leq \|\boldsymbol{A}^{-1}\|\|\mathbf{y}\|(\delta + \epsilon\kappa(\boldsymbol{A})).$$

Finally, normalizing by $\|\mathbf{x}\|$ gives:

$$\frac{\|\Delta\mathbf{x}\|}{\|\mathbf{x}\|} \leq \kappa(\boldsymbol{A})\left(\epsilon + \delta\frac{\|\boldsymbol{A}\|}{\|\mathbf{y}\|}\right).$$

Thus, the relative change in the solution $\mathbf{x}$ is proportional to the condition number $\kappa(\boldsymbol{A})$, demonstrating the sensitivity of the solution to perturbations in systems with a high condition number.

## B  FORMULATION OF THE WIENER DECONVOLUTION

In image inverse problems, the forward model can be formulated as (1). The goal of Wiener deconvolution is to estimate the original image $\mathbf{x}$ by minimizing the overall mean square error in the presence of noise. Wiener deconvolution accomplishes this by applying the Wiener filter, which is designed to minimize the mean square error between the estimated image and the original image.

The Wiener filter is calculated as:

$$W(f) = \frac{|\mathcal{H}(f)|^2}{|\mathcal{H}(f)|^2 + \frac{S_n(f)}{S_x(f)}},$$

where $\mathcal{H}(f)$ is the Fourier transform of $\boldsymbol{A}$, $S_n(f)$ is the power spectral density of the noise, and $S_x(f)$ is the power spectral density of the original image.

The Wiener filter in the context of linear algebra can be represented as:

$$\boldsymbol{A}^\dagger = (\boldsymbol{A}^T\boldsymbol{A} + \lambda\mathbf{I})^{-1}\boldsymbol{A}^T,$$

where $\lambda$ is a regularization parameter.

Using the singular value decomposition (SVD) of $\boldsymbol{A}^T\boldsymbol{A}$, we have:

$$\boldsymbol{A}^T\boldsymbol{A} = V\Sigma^2 V^T,$$

where $V$ and $U$ are orthogonal matrices, and $\Sigma$ is a diagonal matrix containing the singular values $\sigma_i$.

Thus,

$$(\boldsymbol{A}^T\boldsymbol{A} + \lambda\mathbf{I})^{-1} = V(\Sigma^2 + \lambda\mathbf{I})^{-1}V^T.$$

The Wiener filter can then be expressed as:

$$\boldsymbol{A}^\dagger = V(\Sigma^2 + \lambda\mathbf{I})^{-1}\Sigma U^T.$$

Applying this filter to the observed image vector $\mathbf{y}$, we get the estimate of the original image:

$$\mathbf{x}^* = \boldsymbol{A}^\dagger\mathbf{y} = V(\Sigma^2 + \lambda\mathbf{I})^{-1}\Sigma U^T\mathbf{y}.$$

It is important to note the effect of the singular values $\sigma_i$ on the filter. When the singular values $\sigma_i$ are very small, the term $(\Sigma^2 + \lambda\mathbf{I})^{-1}$ becomes very large. This indicates that the filter is highly sensitive to noise for small singular values, which can amplify the noise in the recovered image. The regularization parameter $\lambda$ helps to mitigate this effect by preventing the amplification of noise, thus stabilizing the inversion process.

In summary, the Wiener deconvolution leverages the Wiener filter to recover the original image from a blurred and noisy observation by optimally balancing the noise reduction and image deblurring in the frequency domain. This process, grounded in minimizing the mean square error, is fundamental in restoring degraded images effectively.

## C  ADMM WITH TOTAL VARIATION (TV) REGULARIZATION

The Alternating Direction Method of Multipliers (ADMM) breaks down complex optimization problems into simpler subproblems, speeding up solutions and enhancing flexibility. By incorporating Total Variation (TV) regularization, ADMM becomes adept at tasks like image deblurring and denoising, where preserving edges and reducing noise are critical. With TV regularization, ADMM promotes sparsity in image gradients, thus maintaining sharp edges by penalizing total image variation.

### C.1  ALGORITHMIC FRAMEWORK

The algorithmic framework for ADMM with TV regularization involves the following steps:

**Variable splitting.** To separate the fidelity term from TV regularization, an auxiliary variable is introduced, allowing independent updates.

**Updates.** Iterative updates proceed through three main steps:

- **x-update.** Minimization of the fidelity term with respect to the original variable, often employing linear inversion or gradient descent.
- **z-update.** Application of TV regularization to the auxiliary variable, typically solved using a proximal operator enforcing the TV constraint.
- **Dual update.** Adjustment of the dual variable to align solutions of decomposed subproblems, ensuring consistency across splits.

## C.2 Mathematical Formulation

We revisit a generalized cost function in a classical image restoration approach in Eq. (2). For ADMM with TV regularization, minimizing the cost function in Eq. (2) is expressed as:

$$\tilde{\mathbf{x}} = \arg\min_{\mathbf{x}} \frac{1}{2}\|\boldsymbol{A}\mathbf{x} - \mathbf{y}\|_2^2 + \lambda \text{TV}(\mathbf{x}) \tag{S1}$$

where $\|\boldsymbol{A}\mathbf{x} - \mathbf{y}\|_2^2$ represents a data-fidelity term, and $\text{TV}(\mathbf{x})$ represents a regularization prior promoting sparsity in image gradients.

The optimized estimate at each step for Eq. (S1) is:

$$\tilde{\mathbf{x}}^{k+1} = \mathcal{D}_\sigma\left(\mathbf{T}(\tilde{\mathbf{x}}^k, \mathbf{y}, \boldsymbol{A})\right) \tag{S2}$$

where $\mathcal{D}_\sigma$ is the proximal operator, and $\mathbf{T}$ is the update function specific to the non-linear optimization estimation.

The detailed proximal operator formulation of ADMM with TV regularization is:

$$\mathbf{x}^{k+1} = \arg\min_{\mathbf{x}}\left(\|\boldsymbol{A}\mathbf{x} - \mathbf{y}\|_2^2 + \rho\|\mathbf{x} - \mathbf{z}^k + \mathbf{u}^k\|_2^2\right), \tag{S3}$$

$$\mathbf{z}^{k+1} = \arg\min_{z}\left(\lambda \text{TV}(\mathbf{z}) + \rho\|\mathbf{x}^{k+1} - \mathbf{z} + \mathbf{u}^k\|_2^2\right), \tag{S4}$$

$$\mathbf{u}^{k+1} = \mathbf{u}^k + \mathbf{x}^{k+1} - \mathbf{z}^{k+1}. \tag{S5}$$

Here, $\lambda$ is the regularization parameter controlling the strength of the TV term, and $\rho$ is the penalty parameter for constraint violations.

TV regularization allows precise control over the smoothness and sparsity in image gradients, significantly boosting edge preservation crucial for high-quality visual applications. Additionally, it improves noise reduction without compromising image structure, offering a clear advantage over traditional methods. Furthermore, integrating TV regularization extends ADMM's scalability and flexibility, making it well-suited for large-scale problems across various imaging modalities.

# D Additional Considerations in Method

**Shifted window partition.** Due to the initial fixed location of $\mathcal{M}_t^{(i,j)}$, there is a slight limitation in that the local fidelity attention proceeds only within the patch boundary. Therefore, we adopt a shifted window partition to reduce discontinuities at patch edges and overall artifacts. Detailed algorithm for shifted window partition setting follows as:

$$\mathcal{M}_t^{(i,j)}(r) = \begin{cases} 1 & \text{if } \mathcal{D}_{\text{sum.}}^{(i+r,j+r)}\left(\tilde{\mathbf{x}}^*, \hat{\mathbf{x}}_{0|t}\right) \geq \Omega_\nu\left[\mathcal{D}_{\text{sum.}}^{(i+r,j+r)}\left(\tilde{\mathbf{x}}^*, \hat{\mathbf{x}}_{0|t}\right)\right] \\ 0 & \text{otherwise} \end{cases}$$

$$\text{where } r = t - 16\left\lfloor\frac{t}{16}\right\rfloor$$

Here, $\mathcal{D}_{\text{sum.}}^{(i,j)}(\cdot)$ represents the sum of differences between the pixel values within each patch located at $(i,j)$, and $P_\nu$ is the top percentage threshold. The shift amount $r$ is the pixel-wise index, empirically calculated as $t \mod 16$.

**Skip step guidance.** As mentioned in the main paper, we implement skip step guidance $\mathcal{C}$ during the initial sampling phase to loosely follow PiAC fidelity. The large scalar disparity between $\mathcal{L}_{\text{PiAC}}$ and $\mathbf{s}_{\theta*}(\mathbf{x}_t, t)$ caused local artifacts in some degraded images during the initial sampling. We empirically set the skip step guidance to $\mathcal{C} = 2$ for the first half of the initial reverse sampling steps and $\mathcal{C} = 1$ for the second half, fully applying our mask guidance (Appendix G.5). In other words, steps for unconditional generation and our PiAC with $\mathcal{M}_t$ proceed alternately in the half of initial phase. Detailed algorithm for skip step guidance setting follows as:

$$\mathbf{x}_{t-1} \simeq \begin{cases} \mathbf{x}_t + \mathbf{s}_{\theta*}(\mathbf{x}_t, t), & \text{if } t \geq 500 \text{ and } t \text{ is odd}, \\ \mathbf{x}_t + \mathbf{s}_{\theta*}(\mathbf{x}_t, t) - \rho \mathcal{M}_t \left[ \nabla_{\mathbf{x}_t} \|\tilde{\mathbf{x}}^* - \hat{\mathbf{x}}_{0|t}\|_2^2 \right], & \text{if } t \geq 500 \text{ and } t \text{ is even}, \\ \mathbf{x}_t + \mathbf{s}_{\theta*}(\mathbf{x}_t, t) - \rho \mathcal{M}_t \left[ \nabla_{\mathbf{x}_t} \|\tilde{\mathbf{x}}^* - \hat{\mathbf{x}}_{0|t}\|_2^2 \right], & \text{if } t < 500. \end{cases} \quad \text{(S6)}$$

# E  REAL-WORLD LENSLESS CAMERA AND ITS MEASUREMENTS

**Mask-based lensless camera.** A lensless camera is a new class of compact and low-cost imaging devices based on computational image reconstruction. Instead of using a lens, a lensless camera uses a phase mask placed in front of an image sensor, achieving ultra-thin designs by reducing the thickness of the lens and the focal length of the imaging system. Because the mask randomly modulates the incident light from the scene, the encoded intensity information of the scene should be recovered through computational processing of the raw measurement that is otherwise unidentifiable. Additionally, lensless imaging with 2D PSFs decouples the one-to-one mapping between each position in the scene and the sensor pixels, enabling single-shot multiplexed measurements without using superpixels. Along with their miniaturization and multiplexing capabilities, various applications based on lensless imaging have been widely explored recently, including depth imaging(Antipa et al., 2018; Adams et al., 2022), hyperspectral imaging(Monakhova et al., 2020; Kim et al., 2023), high-speed imaging(Chan et al., 2023), polarization imaging (Baek et al., 2022), and wavefront sensing(Wu et al., 2024).

**Advantages: lens camera vs. lensless camera.** Lensless cameras are typically smaller and lighter, making them suitable for applications where space and weight are critical. Without the need for expensive lens assemblies, lensless cameras can be more cost-effective to produce. They can naturally have a very wide field of view without the distortion issues often associated with wide-angle lenses in traditional cameras. With fewer moving parts and no glass lenses, lensless cameras are more robust and less susceptible to damage. Furthermore, lensless cameras can utilize advanced computational algorithms to reconstruct images, potentially leading to new imaging capabilities and applications. They are crucial in the development of next-generation technologies, such as ultra-thin sensors and integration into compact electronic devices (Zeng & Lam, 2021; Boominathan et al., 2022; Lee et al., 2023).

**Design and fabrication of custom-built lensless camera.** Here, we utilize a designed phase mask that exhibits a sharper and higher contrast point spread function (PSF) to validate our approach with real measurements. We employ a 2D random pattern generation algorithm such as Voronoi constellation to generate a target PSF, which exhibits uniform directionality and high contrast with a given specific density. Then, the height profile ($h$) of the phase mask with a given target pattern is designed with the following optimization problem, where the smoothness constraints are additionally utilized to compensate for the fabrication resolution. Revisiting Eq. (2) in the main paper:

$$\arg\min_h ||\hat{v} - v_0||_2^2 + \lambda ||\Delta h||_2^2, \quad \text{(S7)}$$

where $\hat{v}$ and $v_0$ denote the estimated and measured PSFs, respectively, $\Delta$ is a Laplacian operator, and $\lambda$ is the weight parameter for smoothness constraint. Following the fabrication protocol in (Lee et al., 2023), we fabricated customized phase masks for a lensless camera, sized 1.5 mm x 1.5 mm with a 1.4 mm focal length, exhibiting a Voronoi-patterned PSF (Fig. S2 (a)). We then built customized lensless cameras by combining the fabricated phase masks with a 3D-printed aperture (Fig. S1 (b) and (c)).

**Camera setting for data generation.** The dataset is captured by displaying scenes on an OLED screen 20 cm in front of the lensless camera with a color sensor (IMX 219), using auto exposure to maximize the raw images' SNR (Fig. S1 (d)). The target FOV is 70° in both horizontal and vertical directions, with a camera resolution of 0.42° per pixel.

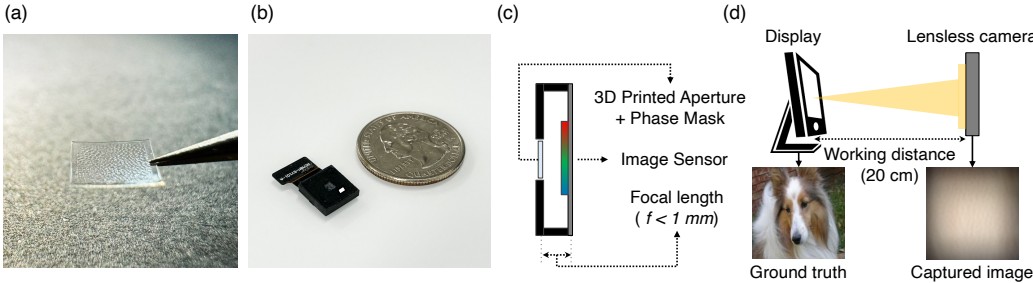

Figure S1: Schematic of lensless camera and setup for dataset generation. (a) Example of custom designed phase mask with Voronoi patterned PSF. (b) Photograph of the lensless camera for real experiments and (c) its side-view schematic. (d) Real dataset generation setup with display.

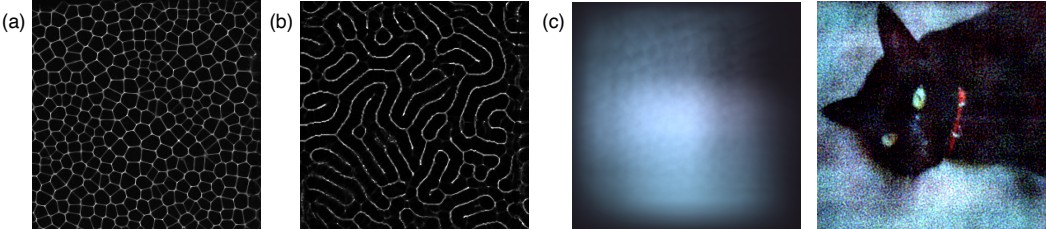

Figure S2: Example of PSF of (a) Voronoi pattern and (b) Turing pattern. (c) Example of raw measurement captured with our lensless camera and its reconstruction results.

**Dataset description for experiment with real measurements.** A total of 100 MirFlickr (Huiskes & Lew, 2008) dataset images is captured with the lensless camera with a Voronoi-shaped PSF (Fig. S2 (a)). After the raw measurement is cropped to $2400 \times 2400$ pixels for preprocessing, the raw measurement is reconstructed by solving the optimization problem formulated in Sec. B and C, obtaining reconstructed images (Fig. S2 (c)).

**Dataset pre-processing for diffusion sampling inputs in our experiments.** Real measurement and real point spread function (PSF), initially sized at $2464 \times 3280$, are processed by performing a center cropping to a dimension of $2400 \times 2400$. Subsequently, these cropped images are downscaled to a resolution of $512 \times 512$ through bicubic interpolation. The resulting measurement and PSF are then utilized to construct PiAC fidelity. Note that for fair evaluation for the restoration outputs, we cropped approximately 5 percent of the outputs to measure performance metrics, aiming to mitigate the mismatch between the ground truth (GT) and the restoration outputs caused by sensor hardware limitations and shooting conditions, as we discussed before.

**Need for zero-shot lensless imaging models.** Data-driven lensless camera raw reconstruction enhances perceptual quality using paired datasets from specific cameras (Lee et al., 2023; Poudel & Nakarmi, 2024; Li et al., 2023a; Rego et al., 2021; Zeng & Lam, 2021), but our zero-shot learning approach requires no additional training. Lensless cameras, designed for various applications such as privacy-preserving imaging, need reconstruction algorithms that generalize well across out-of-distribution datasets, and our method meets this requirement. This zero-shot model accelerates product development for various applications without the need for sensitive training data. While ADMM-TV offers reasonable reconstructions, it suffers from quality degradation that depends on scene and FOV, especially with increased crop factors. Le-ADMM (Monakhova et al., 2019) uses U-net architectures for enhancement but tends to overfit specific hardware datasets. In contrast, our Dilack method manages locally varying degradation using masked fidelity without additional training, making it adaptable to various lensless cameras.

# F  EXPERIMENTAL DETAILS

## F.1  IMPLEMENTATION DETAILS.

$\rho$ **for** $\mathcal{L}_{\textbf{PiAC}}$ **settings.** We observe that setting $\rho$, the guidance scale for $\mathcal{L}_{\text{PiAC}}$, to a constant value yields stable results. It is important to note that other zero-shot diffusion methods (Chung et al., 2023b; Zhu et al., 2023; Garber & Tirer, 2024) set $\rho$ differently for each dataset and task. However, we standardize $\rho$ to 1 for all cases, making our approach more generalizable: $\rho = 1/\|\tilde{\mathbf{x}}^* - \hat{\mathbf{x}}_{0|t}\|$.

$\nu$ **for** $\mathcal{M}_{\textbf{ROI}}$ **settings.** We also observe that taking $\nu$, which is percentage of guided by $\mathcal{L}_{\text{PiAC}}$ for $\mathcal{M}_t$, set to constant, yields stable results. We list the $\nu$ values used in our Dilack algorithm for each problem setting as defined in Eq. 16.

- FFHQ
    - lensless imaging: $\nu = 80^{th}$ percentile
    - Large motion deblurring: $\nu = 80^{th}$ percentile
- ImageNet
    - lensless imaging: $\nu = 80^{th}$ percentile
    - Large motion deblurring: $\nu = 30^{th}$ percentile
- *Real* mask-based camera raw dataset (*i.e.*, MirFlickr-lensless)
    - *Real* lensless imaging: $\nu = 70^{th}$ percentile

**Compute time.** All experiments were conducted on an RTX 3090 GPU. On a single GPU, processing each image takes about 390 seconds, including the ADMM algorithm iterations and mask calculation process. Note that DPS (also based on DDPM) takes 340 seconds, indicating that the Dilack process does not significantly increase processing time as much as one might expect.

**Code availability.** Dilack is based on DPS (Chung et al., 2023b), and the sampling code along with sample data are submitted as supplementary materials. This code is an experiment for a real-world mask-based lensless camera task.

## F.2  INVERSE PROBLEM SETUP.

Our two tasks: 1) Lensless imaging (*i.e.*, mask-based camera raw reconstruction) and 2) Large motion deblurring both involve deblurring kernels, so they share the same forward model. The measurement operator $\boldsymbol{A} \in \mathbb{R}^{n \times n}$ (with $m = n$) is a convolution with some blur kernel $\mathbf{k}$, i.e., $\boldsymbol{A}\mathbf{x} = \mathbf{x} * \mathbf{k}$. The only difference is the source of the measured kernel: one is from a mask-based camera (PSF), and the other is generated from open-source data as mentioned in the main text.

Assuming $\boldsymbol{A}$ is a circulant matrix, it can be diagonalized by the discrete Fourier transform (DFT). Thus, convolution can be computed as element-wise multiplication in the discrete Fourier domain, efficiently implemented via Fast Fourier Transform (FFT). Specifically, for $\mathbf{z} \in \mathbb{R}^n$, the convolution is $\boldsymbol{A}\mathbf{z} = \mathcal{F}^{-1}(\mathcal{F}(\mathbf{k})\mathcal{F}(\mathbf{z}))$, where $\mathcal{F}$ denotes the FFT. Similarly, convolution with the flipped $\mathbf{k}$, represented by $\boldsymbol{A}^T$, is applied as $\boldsymbol{A}^T\mathbf{z} = \mathcal{F}^{-1}\left(\overline{\mathcal{F}(\mathbf{k})}\mathcal{F}(\mathbf{z})\right)$. Finally, the operation $\boldsymbol{A}^T\left(\boldsymbol{A}\boldsymbol{A}^T + \eta\mathbf{I}_n\right)^{-1}\mathbf{z}$ can be computed efficiently as:

$$\boldsymbol{A}^T\left(\boldsymbol{A}\boldsymbol{A}^T + \eta\mathbf{I}_n\right)^{-1}\mathbf{z} = \mathcal{F}^{-1}\left(\frac{\overline{\mathcal{F}(\mathbf{k})}\mathcal{F}(\mathbf{z})}{|\mathcal{F}(\mathbf{k})|^2 + \eta}\right). \tag{S8}$$

## F.3  COMPARISON METHODS SETUP.

We utilize Wiener deconvolution and ADMM-TV with specific hyperparameters for each method. For DPS, DiffPIR, DDPG, and our method Dilack, we use the same weights from the pre-trained diffusion model.

$\boldsymbol{A}^\dagger\mathbf{y}$ **(Wiener deconvolution).** We set the parameter $\alpha$ of the Wiener filter to 0.0 for the lensless imaging task and to 0.01 for the motion deblurring task.

**ADMM$_{TV}$.** We configured the parameters of ADMM-TV differently for our tasks. The number of optimization iterations was empirically fixed at 1,000. In Eq. (S1), we set initial $\lambda_{T-1}$ to $10^{-7}$ for all the synthetic and *real* tasks in both lensless imaging and motion blur tasks. Note that the remaining parameters are the same as those used for PiAC fidelity in each task. We assumed zero-gradient boundary conditions (Neumann conditions) at the solution space boundaries.

**DPS.** We set the step size of DPS as $\zeta_i = 1/\|y - A(\hat{x}_0(x_i))\|$ for all tasks. In previous research, the step size was optimized for each specific task. However, for the new tasks proposed in our study, optimal parameters have not yet been established, leading us to configure the same parameters across all tasks.

**DiffPIR.** We set the same parameter settings as those proposed for the 100 NFEs in DifPIR. The performance varies significantly depending on the given image noise level. In the case of the (large) motion deblur task, injecting the actual noise level of 0.0005 tends to produce artifacts. Therefore, the noise level set in the algorithm was adjusted to 0.025.

**DDPG.** We used the same settings as those used in the motion deblurring task in DDPG. Specifically, using the hyper-parameters for the motion deblur task with $\sigma_e = 0.05$, we applied $\gamma = 6$, $\zeta = 0.6$, and $\eta = 0.7$ across all tasks.

While SVD-based methods like DDRM (Kawar et al., 2022), DDNM (Wang et al., 2022), and DeqIR (Cao et al., 2024) are effective for separable kernels, they struggle with highly ill-posed kernels, making them impractical for 2D image experiments involving complex non-linearity or asymmetry. For instance, DDRM works for Gaussian deblurring but fails with more complex tasks like motion deblurring. This is why the DPS authors did not test DDRM for motion blur restoration. In contrast, our method overcomes these limitations, offering a more flexible solution for complex kernel degradations. As noted in Sec. 2, we propose replacing the score function in Eq. 4 with a Bayesian framework to tackle more challenging tasks.

# G  ADDITIONAL EXPERIMENTAL RESULTS

## G.1  EFFECT OF ROI MASK.

Table S1: Ablation study of ROI mask on the synthetic datasets. For *real* lensless dataset, see Tab. 2 in the main paper. Note that PiAC fidelity is also one of our proposed methods.

| Ablation studies Guidance | Lensless Imaging (ImageNet) PSNR↑/SSIM↑/FID↓/LPIPS↓ | Lensless Imaging (FFHQ) PSNR↑/SSIM↑/FID↓/LPIPS↓ | Large Motion Deblur (ImageNet) PSNR↑/SSIM↑/FID↓/LPIPS↓ | Large Motion Delbur (FFHQ) PSNR↑/SSIM↑/FID↓/LPIPS↓ |
|---|---|---|---|---|
| ADMM$_{TV}$ | 20.57 / 0.575 / 34.84 / 0.293 | 20.28 / 0.488 / 54.98 / 0.362 | 19.90 / 0.528 / 120.07 / 0.492 | 21.32 / 0.625 / 125.02 / 0.459 |
| $s_\theta + \mathcal{L}_{PiAC}$ | **25.26 / 0.818 / 33.77 / 0.205** | **26.55 / 0.872 / 27.92 / 0.143** | 20.61 / 0.593 / 109.88 / 0.448 | 22.63 / 0.725 / 104.27 / 0.366 |
| $s_\theta + \mathcal{M}_{random}[\mathcal{L}_{PiAC}]$ | 23.74 / 0.727 / 49.29 / 0.295 | 25.62 / 0.828 / 35.83 / 0.191 | **20.99 / 0.616** / 84.38 / 0.421 | 23.04 / 0.738 / **55.86** / 0.320 |
| $s_\theta + \mathcal{M}_{ROI}[\mathcal{L}_{PiAC}]$ | 24.94 / 0.798 / 35.61 / 0.225 | 26.24 / 0.860 / 28.69 / 0.156 | **20.99** / 0.612 / **77.46** / 0.420 | **23.15 / 0.745** / 59.6 / **0.313** |

**Without ROI mask.** From the ablation studies indicated in Tab. 2 and S1, without the ROI mask, using pure $\mathcal{L}_{PiAC}$ leads to poor outputs due to $\tilde{x}^*$'s inherent noise and artifacts. Additionally, alone also struggles to capture local details. The ROI mask helps capture details such as small text, fingers, textures, patterns, and facial features (see Fig. S3 and S5). These issues are more pronounced in real lensless measurements, which are noisier and blurrier with less accurate local details.

**With random mask.** From Tab. 2 and S1, using an ROI mask leads to better performance than a random mask. This improvement is due to the ROI mask's ability to calculate local ROI differences between iterations, maintaining continuity, focusing more on local details, and enhancing restoration. This validates our ROI mask design. Qualitative results are in Fig. S4.

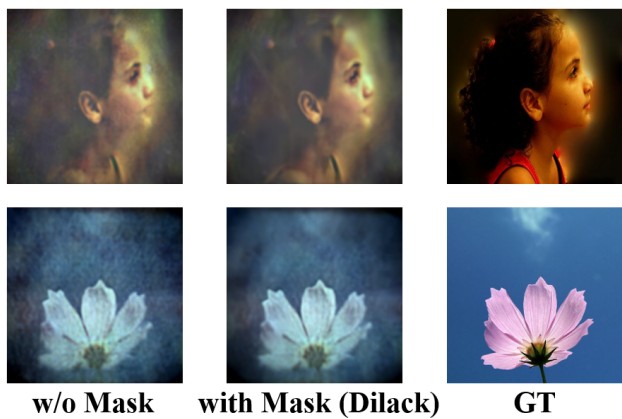

**w/o Mask**  **with Mask (Dilack)**  **GT**

Figure S3: The qualitative comparisons for the effectiveness of the ROI mask in 'real' lensless camera dataset. 'w/o Mask' refers to $\mathbf{s}_\theta + \mathcal{L}_{\text{PiAC}}$, while 'with Mask', which is Dilack, refers to $\mathbf{s}_\theta + \mathcal{M}_{\text{ROI}}[\mathcal{L}_{\text{PiAC}}]$.

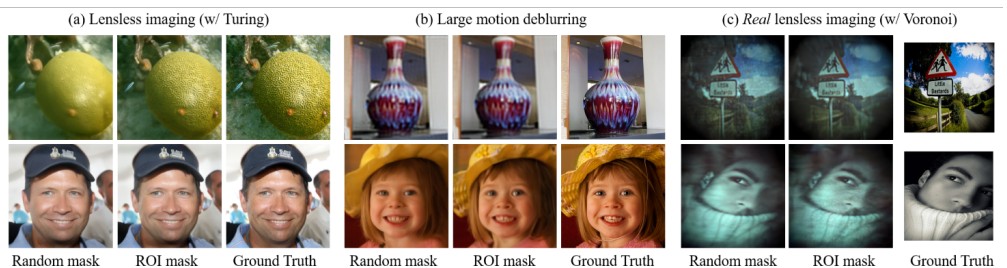

Figure S4: Qualitative comparisons for PiAC with Random mask or ROI mask.

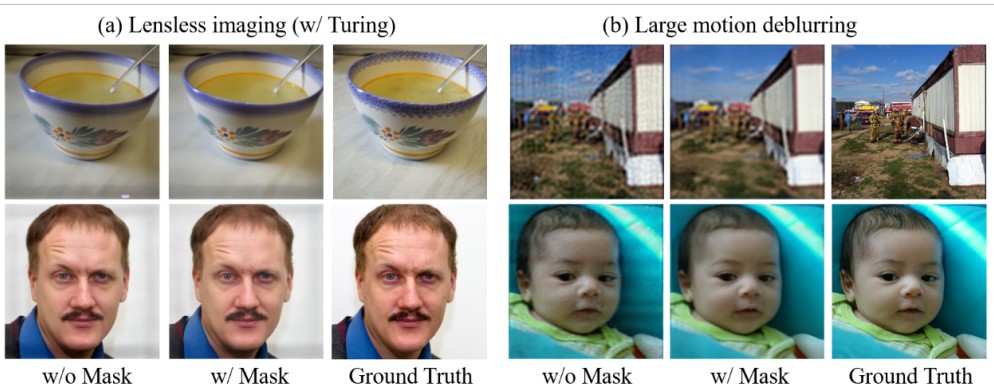

Figure S5: Qualitative comparisons for PiAC with or without Mask. Note that real lensless comparisons are in Fig. S3 above.

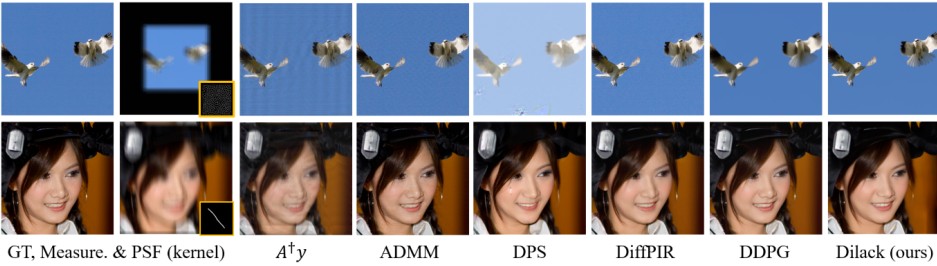

Figure S6: Qualitative comparison in *lighter* (normal) kernel degradation cases. First row: lensless imaging, kernel size = $64^2$; second row: motion blur, kernel size = $61^2$, intensity = 0.5.

## G.2 COMPARISONS IN *lighter* (NORMAL) KERNEL DEGRADATION CASES

Table S2: Additional experiments: the comparisons in *lighter* (normal) kernel degradation cases.

| Lighter cases | Lensless Img. w/ kernel 64 (ImageNet) | Lensless Img. w/ kernel 64 (FFHQ) | Motion Deblur (ImageNet) | Motion Deblur (FFHQ) |
|---|---|---|---|---|
| Method | PSNR↑/SSIM↑/FID↓/LPIPS↓ | PSNR↑/SSIM↑/FID↓/LPIPS↓ | PSNR↑/SSIM↑/FID↓/LPIPS↓ | PSNR↑/SSIM↑/FID↓/LPIPS↓ |
| $ADMM_{TV}$ | 31.76 / 0.897 / 18.13 / 0.092 | 31.55 / 0.879 / 33.46 / 0.111 | 27.89 / 0.894 / 41.20 / 0.135 | 29.96 / 0.914 / 46.34 / 0.123 |
| DPS | 18.82 / 0.548 / 142.94 / 0.409 | 13.39 / 0.691 / 79.60 / 0.261 | 18.03 / 0.413 / 172.14 / 0.442 | 23.32 / 0.728 / 65.55 / 0.218 |
| DiffPIR | 29.20 / 0.875 / 13.78 / 0.087 | 25.76 / 0.711 / 54.95 / 0.227 | 32.83 / 0.922 / 16.65 / 0.068 | 28.89 / 0.813 / 48.61 / 0.163 |
| DDPG | 26.75 / 0.748 / 144.16 / 0.317 | 31.50 / 0.903 / 27.50 / 0.114 | 35.42 / 0.947 / 30.63 / 0.065 | 34.40 / 0.934 / 33.29 / 0.080 |
| **Dilack(ours)** | 31.94 / 0.877 / 49.74 / 0.158 | 35.17 / 0.943 / 34.90 / 0.081 | 24.78 / 0.703 / 94.82 / 0.317 | 28.63 / 0.858 / 62.78 / 0.178 |
| LDPS | — | 17.76 / 0.482 / 264.39 / 0.512 | — | 22.07 / 0.608 / 119.15 / 0.352 |
| ReSample | — | 25.44 / 0.763 / 123.12 / 0.281 | — | 27.16 / 0.800 / 57.36 / 0.179 |
| ReSample-Dilack | — | 27.37 / 0.867 / 49.67 / 0.130 | — | 23.94 / 0.770 / 101.37 / 0.277 |

As shown in Tab. S2 and Fig. S6, Dilack performs well in lighter cases. However, our main focus is on addressing the limitations of existing methods in challenging scenarios like lensless imaging and large motion deblurring, using the first 100 images (indexes 0–99) from the ImageNet and FFHQ datasets. While we respect zero-shot methods for standard tasks, our work targets real-world challenges like lensless imaging and severe motion blur, where current methods fail. Dilack addresses these gaps. In lensless imaging, reducing the PSF size from 256 to 64 improves performance, and for motion blur, decreasing the kernel size (256 to 61) and intensity (1.0 to 0.5) proves effective. The lighter motion blur setting aligns with that used in the original DPS paper. Note that the pre-trained weights of latent diffusion models are only available for those trained on FFHQ, so experiments on ImageNet could not be conducted.

## G.3 EFFECT OF MASKING RATIO

As shown in Fig. S7, there is a trade-off between structural consistency and perceptual quality of the restored image depending on the ROI masking ratio. Empirically, we set the synthetic ROI masking ratio to 0.8 for synthetic tasks and 0.7 for real lensless imaging, as detailed in Sec. F.1.

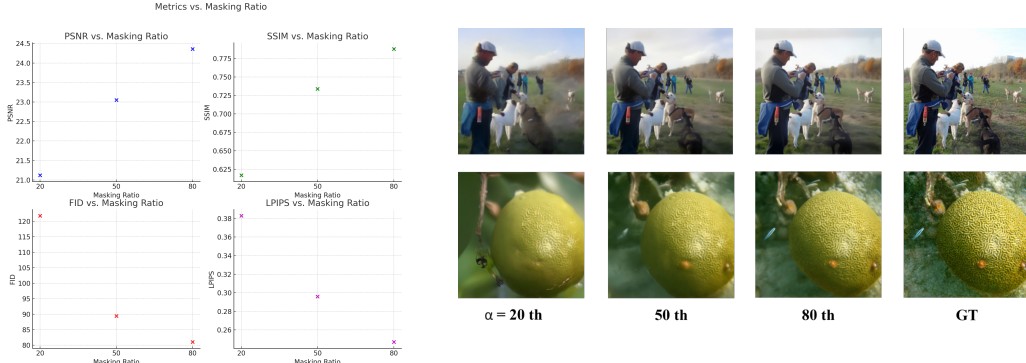

Figure S7: Effectiveness of masking ratio ($\nu^{th}$ percentile) on the ImageNet dataset with synthetic lensless task: (a) quantitative results and (b) qualitative results.

## G.4 EFFECT OF GUIDANCE SCALE

The guidance scale for $\mathcal{L}_{PiAC}$ is important hyper-parameter since it is given to approximation of likelihood (i.e. data consistency) of the inverse problem. In Fig. S8, we show the tendency of consistency control according to the intensity of guidance $\rho$. The lower the guidance scale $\rho$ is, we get results that are not consistent with the original image and get blurry. We empirically set the $\rho$ value 1 for best results in consistency.

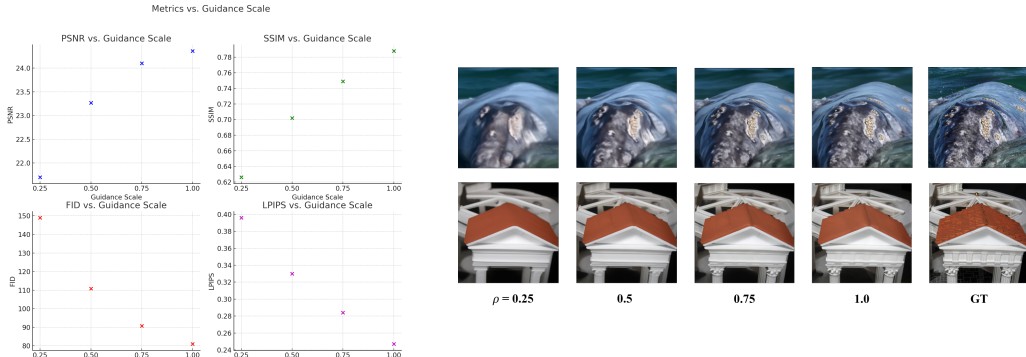

Figure S8: Effectiveness of guidance scale $\rho$ for $\mathcal{L}_{\text{PiAC}}$ on the ImageNet dataset with synthetic lensless task: (a) quantitative results and (b) qualitative results.

### G.5 EFFECT OF SKIP STEP GUIDANCE

As mentioned in the main paper Sec. 4.2 **Additional considerations** and detailed in Appendix D, we conducted an ablation study on the skip step guidance $\mathcal{C}$, which defines the steps in the sampling iteration where Dilack guidance (PiAC guidance with ROI mask) are applied. To quickly assess the trend, we tested on 30 images (indices 0–29) from ImageNet in the lensless imaging task with a Turing PSF. Setting $\mathcal{C} = 2$ during the initial diffusion sampling iterations and $\mathcal{C} = 1$ in the latter half resulted in the best time-performance efficiency.

Table S3: Inference time and performance results for Dilack with various skip step guidance.

| Method | Skip Step Guidance | Inference Time | PSNR↑/SSIM↑/FID↓/LPIPS↓ |
|---|---|---|---|
| Dilack(ours) | 1 | 6:58 | 24.48 / 0.791 / 79.90 / 0.245 |
| | **2 (first half) + 1 (latter half)** | 6:14 | 24.66 / 0.788 / 80.99 / 0.247 |
| | 1 (first half) + 2 (latter half) | 6:19 | 23.05 / 0.712 / 119.79 / 0.320 |
| | 2 | 6:07 | 22.72 / 0.702 / 125.59 / 0.330 |
| | 5 | 5:32 | 21.36 / 0.604 / 150.63 / 0.411 |
| | 10 | 5:11 | 19.56 / 0.529 / 175.15 / 0.478 |

### G.6 EFFECT OF TOTAL NUMBERS FOR UPDATING TV-REGULARIZED OPTIMIZATION SOLUTION $\tilde{\mathbf{x}}^*$

As shown in **Algorithm 1** of the main paper, classical TV-regularized optimization starts with initial values of zero and is re-initialized $G - 1$ times using intermediate sampling outputs $\hat{\mathbf{x}}_{0|t}$. By using an appropriate initialization point $\hat{\mathbf{x}}_{0|t}$, the optimization converges faster and helps prevent the model from getting stuck in local minima.

We conducted an ablation study on $G$, which specifies the total number of updates for the TV-regularized optimization solution $\tilde{\mathbf{x}}^*$. The study was performed on 30 images (indices 0–29) from the synthetic lensless imaging dataset, using the same implementation settings as in Sec. G.5. Additionally, we iteratively updated $\tilde{\mathbf{x}}^*$ and reduced the regularizer's hyperparameter $\tau$ during optimization, allowing traditional regularizers like TV to interact locally with the fidelity term.

The initial $\lambda_t$ value, $\lambda_{T-1}$, was initially set to $10^{-7}$ and then decreased based on the total initialization count (G). Specifically, $\lambda_t$ decreases as:

$$\lambda_t = 10^{-7} \times \{1 - (\text{current ADMM update iteration/total ADMM updates})\} \tag{S9}$$

at every $t = 1,000/G$ iteration step when the sampling iteration $t \in T$ is at an initialization point.

Increasing $G$ resulted in a modest increase in PSNR, a pixel-based quality metric, but only slight improvements in perceptual quality metrics like LPIPS and FID, while significantly increasing inference time. Therefore, to balance inference time and performance metrics, we set $G = 1$ in our experiments, meaning no re-initialization was performed. This approach has the added advantage of

performing optimization once before entering the **for**-loop in **Algorithm 1**, allowing for continuous use of the initial $\tilde{\mathbf{x}}^*$, simplifying the implementation.

Table S4: Inference time and performance results for Dilack with various total numbers of optimization re-initializations.

| Method | Num. of Updating of $\tilde{\mathbf{x}}^*$ | Inference Time | PSNR↑/SSIM↑/FID↓/LPIPS↓ |
|---|---|---|---|
| Dilack(ours) | **1** (no further updates after the first optim.) | 6:14 | 24.66 / 0.788 / 80.99 / 0.247 |
| | 5 (re-init. every 200 iter.) | 6:58 | 24.97 / 0.791 / 78.80 / 0.243 |
| | 10 (re-init. every 100 iter.) | 7:24 | 25.08 / 0.792 / 78.81 / 0.242 |
| | 50 (re-init. every 20 iter.) | 10:06 | 25.55 / 0.793 / 78.87 / 0.240 |

### G.7 EXPERIMENTAL SETUP FOR APPLYING DILACK TO *Latent* DIFFUSION MODELS

As discussed in Sec. 5.3 of the main paper, diffusion models in pixel space perform better under our highly ill-posed kernel degradation setting. To investigate whether our Dilack fidelity can produce similar results in latent diffusion models, we conducted additional experiments. **Algorithm 2** presents the application of our proposed Dilack approach to the original state-of-the-art latent diffusion model ReSample (Song et al., 2024).

The main differences between the original Re-Sample and ReSample-Dilack are: *i*) the use of latent diffusion models leveraging $\mathbf{z}$, *ii*) Re-Sample employs least square guidance $\mathcal{L}_{LS}$, and *iii*) ReSample optimizes $\hat{\mathbf{z}}_0$ using a gradient descent method.

While all other settings remain unchanged from the original ReSample paper, the optimization step (highlighted in *purple*) varies, leading to significantly different results under highly ill-posed kernel degradation settings, as shown in Fig. 5. This demonstrates the effectiveness of our Dilack fidelity in challenging scenarios, even when applied to existing latent diffusion methods.

---

**Algorithm 2** ReSample-Dilack

**Require:** $\boldsymbol{A}$, $\mathbf{y}$, $\lambda_{T-1}$, $\rho$, $T$, $\mathcal{C}$, $G$, $s_\theta(\cdot, t)$, $\{\tilde{\sigma}_t\}_{t=1}^T$, Encoder $\mathcal{E}(\cdot)$, Decoder $\mathcal{D}(\cdot)$, Pretrained LDM Parameters $\beta_t$, $\alpha_t$, $\eta$, $\delta$, and Hyperparameter $\gamma$ to control $\sigma_t^2$

1: $\mathbf{z}_T \sim \mathcal{N}(\mathbf{0}, \mathbf{I})$
2: **for** $t = T - 1, \ldots, 0$ **do**
3:     $\epsilon_1 \sim \mathcal{N}(\mathbf{0}, \mathbf{I})$
4:     $\hat{\epsilon}_{t+1} = s_\theta(\mathbf{z}_{t+1}, t+1)$
5:     $\hat{\mathbf{z}}_0(\mathbf{z}_{t+1}) = \frac{1}{\sqrt{\bar{\alpha}_{t+1}}}(\mathbf{z}_{t+1} - \sqrt{1 - \bar{\alpha}_{t+1}}\hat{\epsilon}_{t+1})$
6:     $\mathbf{z}_t' = \sqrt{\bar{\alpha}_t}\hat{\mathbf{z}}_0 + \sqrt{1 - \bar{\alpha}_t - \eta\delta^2}\hat{\epsilon}_{t+1} + \eta\delta\epsilon_1$
7:     **if** $t \in \mathcal{C}$ **then**
8:       $\tilde{\mathbf{x}}^* \in \arg\min_{\mathbf{x}} \|\mathbf{y} - \boldsymbol{A}\mathbf{x}\|_2^2 + \lambda_t \text{TV}(\mathbf{x})$ *// Classical TV-regularized optimization starts with initial values of 0 and is re-initialized $G - 1$ times using intermediate sampling outputs.*
9:       $\hat{\mathbf{z}}_0(\tilde{\mathbf{x}}^*) \in \arg\min_{\mathbf{z}} \mathcal{M}_{\text{ROI}} \cdot \|\tilde{\mathbf{x}}^* - \mathcal{D}(\mathbf{z})\|_2^2$
10:       $\mathbf{z}_t = \text{StochasticResample}(\hat{\mathbf{z}}_0(\tilde{\mathbf{x}}^*), \mathbf{z}_t', \gamma)$
11:     **else**
12:       $\mathbf{z}_t = \mathbf{z}_t'$ *// Unconditional sampling.*
13:     **end if**
14: **end for**
15: **return** $\mathcal{D}(\mathbf{z}_0)$

---

### G.8 REPLACING PIAC (PSEUDO-INVERSE ANCHOR FOR CONTRAINING) WITH OTHER DENOISING TECHNIQUES

BM3D (Dabov et al., 2007) is one of the denoising methods and can be applied to pseudo-inverse anchor. To evaluate the effectiveness of our PiAC guidance, we replaced PiAC with BM3D while preserving all other components of the framework. To quickly assess the trend, we tested on 3 images (indices 0–2) from FFHQ in the lensless imaging task with a Turing PSF and large motion deblurring task. As demonstrated in Fig. S9, BM3D fails to provide effective guidance as it performs denoising without incorporating the kernel, making it unsuitable for highly ill-posed problems involving large and complex kernels.

### G.9 COMPARISON WITH OTHER PLUG-AND-PLAY METHODS

Since our Dilack algorithm can be regarded as a Plug-and-Play (PnP) method that combines a total variation (TV)-regularized solution and a diffusion prior, we conducted additional experiments comparing its results with DPIR (Zhang et al., 2021), a representative PnP approach utilizing a CNN-based pre-trained denoiser. Because the pre-trained weights of DPIR are designed for $128 \times 128$ resolution, we resized our inputs accordingly and evaluated our lensless turing kernel based imaging task on the ImageNet dataset. We employed a pre-trained diffusion model prior on $128 \times 128$ inputs to match the resolution used in these experiments. As demonstrated in Fig. S10 and Tab. S5, due to the limitations of the Wiener process and the use of a CNN-based pre-trained denoiser as a prior instead of a diffusion prior, DPIR's performance was inferior to that of Dilack.

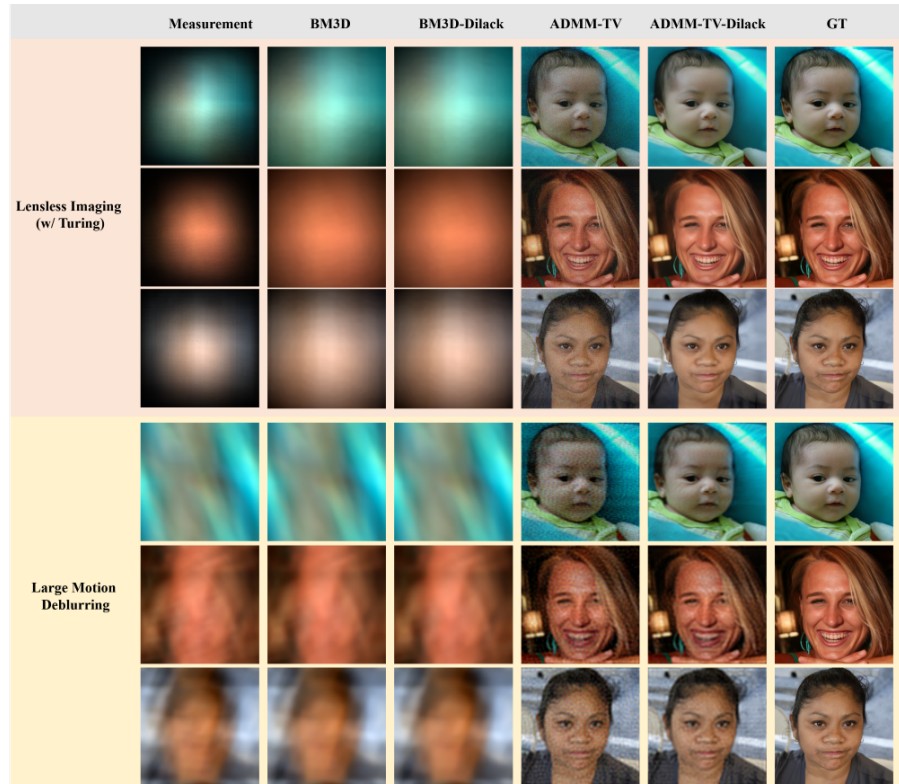

Figure S9: Qualitative results for the synthetic lensless imaging task showing the impact of pseudo-inverse anchor variations in our method. BM3D-Dilack indicates that our method uses BM3D as the pseudo-inverse anchor instead of $\text{ADMM}_{TV}$.

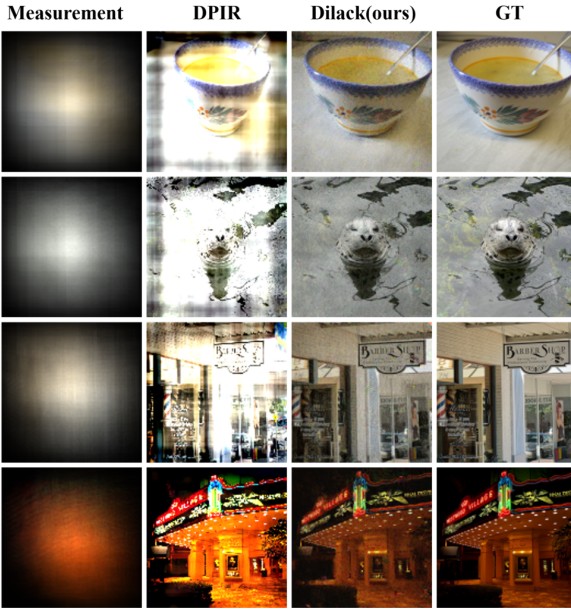

Figure S10: Additional qualitative results of the synthetic lensless imaging task using the Turing kernel on the ImageNet dataset under Plug-and-Play algorithms.

Table S5: Performance comparison of Dilack and DPIR for lensless turing kernel imaging task on the ImageNet dataset.

| Task | Dataset | Method | PSNR↑ / SSIM↑ / FID↓ / LPIPS↓ |
|------|---------|--------|-------------------------------|
| Lensless Imaging (w/ Turing) | ImageNet | DPIR | 9.28 / 0.265 / 236.07 / 0.552 |
| | | **Dilack(ours)** | **24.88 / 0.796 / 92.89 / 0.208** |

### G.10 DPS PERFORMANCE ACROSS VARIOUS STEP-SIZE ADJUSTMENTS

DPS (Chung et al., 2023b) is a zero-shot diffusion model designed to address restoration tasks, including super-resolution and motion deblurring. However, as shown in Fig. 2, DPS exhibited degraded image reconstruction performance, as $\mathcal{L}_{LS}$ provides insufficient guidance for highly ill-posed problems involving large and complex kernels. To evaluate whether step-size optimization could improve DPS's performance, we conducted additional experiments using our two tasks on 100 sample images (indices 0–99) from the FFHQ dataset. We found that even when optimized, it fails to resolve the challenges posed by lensless imaging and large motion deblurring, as shown in Tab. S6.

Table S6: Performance comparison of DPS across different step-sizes for lensless imaging and large motion deblurring tasks. Regardless of the step-size setting, DPS consistently demonstrates poor performance on our tasks, as reflected in the evaluation metrics.

| Task | Dataset | Step-size | PSNR↑ / SSIM↑ / FID↓ / LPIPS↓ |
|------|---------|-----------|-------------------------------|
| Lensless Imaging (w/ Turing) | FFHQ | 0.25 | 11.20 / 0.385 / 120.83 / 0.514 |
| | | 0.50 | 10.71 / 0.382 / 134.62 / 0.528 |
| | | 0.75 | 10.34 / 0.376 / 136.65 / 0.541 |
| | | 1.00 | 9.95 / 0.369 / 150.88 / 0.558 |
| Large Motion Deblurring | FFHQ | 0.25 | 17.70 / 0.502 / 94.48 / 0.364 |
| | | 0.50 | 17.99 / 0.514 / 91.80 / 0.348 |
| | | 0.75 | 17.87 / 0.512 / 96.66 / 0.347 |
| | | 1.00 | 17.85 / 0.511 / 99.10 / 0.349 |

### G.11 COMPARISON WITH OTHER SPLITTING-BASED ITERATIVE OPTIMIZATION METHODS

As discussed in Sec. C, we use $ADMM_{TV}$, as an anchor to enhance the performance of Dilack. To validate the effectiveness of using $ADMM_{TV}$ as the primary anchor in our Dilack algorithm, we conducted additional experiments. We compared our method with $ADMM\text{-}L_1$, $ADMM\text{-}L_2$, FISTA (Fast Iterative Soft-Thresholding Algorithm, as an alternative optimization method), and HQS (a proximal splitting algorithm similar to $ADMM_{TV}$). FISTA employed $\mathcal{L}_1$ regularization, while HQS used Total Variation (TV), like $ADMM_{TV}$. Each approach was applied to $\mathcal{L}_{PiAC}$ guidance and integrated with the diffusion prior in our Dilack framework. These experiments were conducted on the lensless turing kernel imaging task using the FFHQ dataset, with a sample size of 100 images (indices 0–99). The quantitative results are summarized below. As shown in Fig. S11 and Tab. S7, $ADMM_{TV}$ consistently outperformed others, demonstrating its effectiveness and suitability for integration into our Dilack algorithm.

Table S7: Quantitative comparison of Dilack and its variants with different regularizers and optimization algorithms.

| Task | Dataset | Method | PSNR↑ / SSIM↑ / FID↓ / LPIPS↓ |
|------|---------|--------|-------------------------------|
| Lensless Imaging (w/ Turing) | FFHQ | FISTA-Dilack | 19.30 / 0.745 / 98.06 / 0.249 |
| | | HQS-Dilack | 12.51 / 0.341 / 363.52 / 0.637 |
| | | $ADMM_{L_1}$-Dilack | 16.11 / 0.672 / 126.69 / 0.301 |
| | | $ADMM_{L_2}$-Dilack | 22.69 / 0.817 / 64.25 / 0.183 |
| | | **Dilack(ours)** | **26.23 / 0.863 / 54.84 / 0.149** |

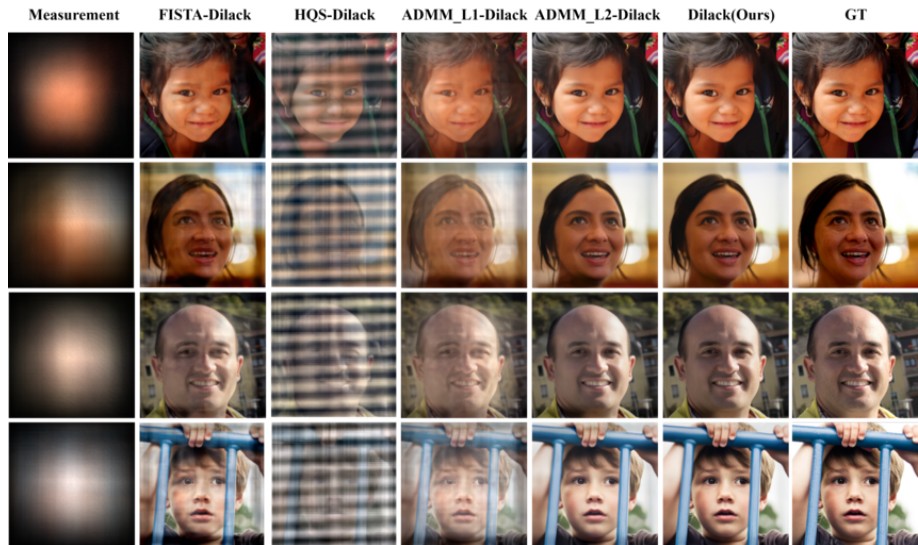

Figure S11: Additional qualitative analysis of the synthetic lensless imaging task on the FFHQ dataset comparing Dilack and its variants with different regularizers and optimization algorithms.

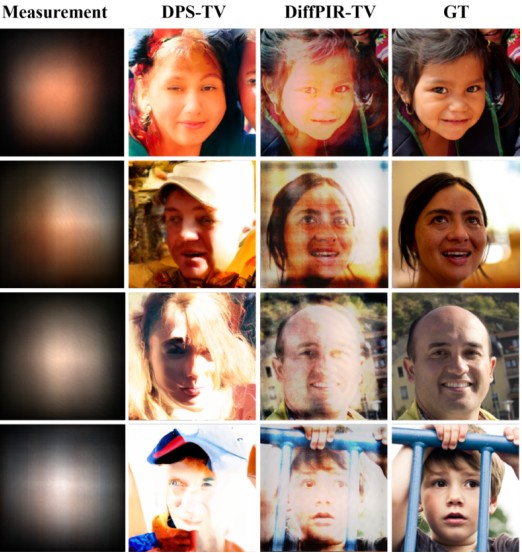

Figure S12: Additional qualitative results of the synthetic lensless imaging task using the Turing kernel on the FFHQ dataset under standard $\mathcal{L}_2$ data-fidelity term based diffusion algorithms.

### G.12 ADDITIONAL RESULTS OF TV REGULARIZATION ON EXISTING $L_2$ DATA-FIDELITY

To verify whether TV regularization is effective in representative algorithms that combine the $\mathcal{L}_2$ data-fidelity term in Eq. (9) and Eq. (10) with a diffusion prior—namely, DPS (Chung et al., 2023b) and DiffPIR (Zhu et al., 2023)—,we performed experiments on the FFHQ dataset with a sample size of 100 images (indices 0–99) for the lensless Turing kernel deblurring task. Specifically, we tested DPS augmented with a TV regularizer ($DPS_{TV}$) and DiffPIR augmented with a TV regularizer ($DiffPIR_{TV}$). As shown in Fig. S12 and Tab. S8, adding TV regularization to Eq. (9) and Eq. (10) results in a global smoothing effect, significantly degrading the fidelity of the sampling output. This approach is insufficient for handling the highly ill-posed problems addressed by our method.

Table S8: Performance results of DPS-TV and DiffPIR-TV for lensless turing kernel imaging task on the FFHQ dataset. Consequently, it is evident that the absolute performance values across all evaluation metrics are significantly low.

| Task | Dataset | Method | PSNR↑ / SSIM↑ / FID↓ / LPIPS↓ |
|---|---|---|---|
| Lensless Imaging (w/ Turing) | FFHQ | $DPS_{TV}$ | 9.81 / 0.366 / 149.41 / 0.560 |
| | | $DiffPIR_{TV}$ | 13.68 / 0.556 / 186.2 / 0.436 |

### G.13 HYPERPARAMETER ANALYSIS ON *real* LENSLESS DATASET

We conducted additional experiments on the real lensless imaging task to evaluate how varying step numbers in **Algorithm 1** affects the results. The experimental setup was identical to Sec. 5.3, and 100 images (indices 0–99) were tested. The scenarios are as follows:

**Case 1**: Increasing the TV-regularized optimization steps from 1,000 to 2,000.

**Case 2**: Setting skip step guidance $\mathcal{C} = 1$ for all sampling iterations.

**Case 3**: Increasing the re-initialization count from $G = 1$ to $G = 50$.

We varied these three settings, and their definitions are described in Sec. 4.2 and **Algorithm 1**. As shown in Fig. S13 and Tab. S9, from **Case 1** to **Case 3**, increasing the number of steps naturally extended the diffusion sampling time per image. However, the performance of the reconstructed images slightly deteriorated, likely due to saturation in the TV-regularized optimization process or overly constrained $\mathcal{L}_{\text{PiAC}}$ guidance. Although there is room for performance improvement, we emphasize that real-world datasets are intrinsically challenging due to model mismatch and hardware-induced noise. As our work is the first to address such highly ill-posed kernel degradation problems using zero-shot diffusion models, we believe this study represents an important first step. Specifically, in **Case 3**, increasing the re-initializing count $G$ reduced fidelity in real-world problems, unlike in synthetic conditions. This indicates that while adjusting $G$ may be beneficial in Sec. G.6 on the synthetic lensless imaging dataset, it poses challenges in more complex, real-world scenarios.

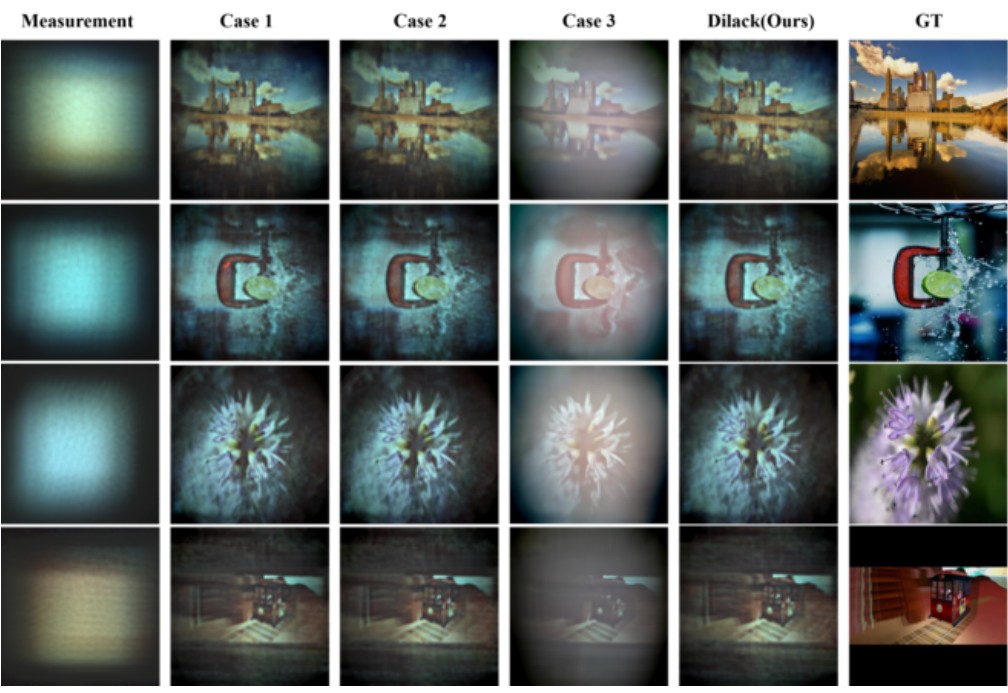

Figure S13: Additional qualitative results of *real* lensless imaging task on MirFlickr-lensless dataset under various hyperparameters.

Table S9: Performance comparison of various hyperparameters on the *real* lensless camera dataset.

| Task | Dataset | Method | Inference Time | PSNR↑ / SSIM↑ / FID↓ / LPIPS↓ |
|---|---|---|---|---|
| *Real* Lensless Imaging | **MirFlickr-lensless** | Case 1 | 06:39 | 12.80 / 0.318 / 294.76 / 0.596 |
| | | Case 2 | 06:30 | 12.81 / 0.318 / 295.86 / 0.597 |
| | | Case 3 | 07:14 | 10.87 / 0.313 / 305.49 / 0.608 |
| | | **Dilack(ours)** | 10:30 | **13.47 / 0.326 / 290.54 / 0.584** |

## G.14 IMPACT OF ADMM-TV OPTIMIZATION ITERATIONS

We employ ADMM-TV for pseudo-inverse guidance, with the optimization iteration serving as a critical parameter. To evaluate its impact, we conducted experiments using iterations of 1, 10, and 1,000 on 30 images (indices 0–29) from FFHQ and ImageNet in the lensless imaging task with a Turing PSF, as well as in the large motion deblurring task. As presented in Tab. S10, while 10 iterations achieve comparable performance, 1,000 iterations consistently deliver superior results across various tasks and datasets.

Table S10: Performance comparison of $ADMM_{TV}$ optimization iterations for Dilack across different tasks and datasets.

| Task | Dataset | $ADMM_{TV}$ iter. | num. of init. | PSNR↑ / SSIM↑ / FID↓ / LPIPS↓ |
|---|---|---|---|---|
| Lensless Imaging (w/ Turing) | FFHQ | 1,000 | 1 | **26.08 / 0.861 / 65.95 / 0.152** |
| | | 1 | 1,000 | 16.23 / 0.601 / 267.89 / 0.428 |
| | | 1 | 1 | 16.73 / 0.608 / 313.32 / 0.425 |
| | ImageNet | 1000 | 1 | **24.36 / 0.788 / 80.99 / 0.247** |
| | | 1 | 1000 | 15.68 / 0.523 / 235.04 / 0.465 |
| | | 1 | 1 | 15.41 / 0.518 / 258.07 / 0.478 |
| Large Motion Deblurring | FFHQ | 1000 | 1 | **22.97 / 0.732 / 197.59 / 0.365** |
| | | 1 | 1,000 | 14.88 / 0.489 / 259.68 / 0.581 |
| | | 1 | 1 | 14.72 / 0.474 / 330.15 / 0.596 |
| | ImageNet | 1000 | 1 | **20.61 / 0.596 / 200.81 / 0.477** |
| | | 1 | 1,000 | 14.22 / 0.386 / 293.76 / 0.621 |
| | | 1 | 1 | 13.34 / 0.368 / 337.58 / 0.645 |

## G.15 REGARDING THE POTENTIAL APPLICABILITY TO OTHER TASKS

For super-resolution (SR), the $A$ matrix consists of a combination of blur kernels and downsampling operations. As the degree of downsampling increases, the condition number becomes larger, presenting a challenging scenario where the strengths of our proposed method are expected to be effective. However, highly ill-posed SR has not been well investigated in the field yet, so we believe that we need to carefully validate one by one. Especially, the impact of downsampling operator is worth investigating. Thus, at this moment, we can say that our Dilack can work for SR as compared to other prior works, but there is still room for improvement due to the reasons mentioned. The qualitative results of toy experiments for SR on the FFHQ dataset can be found in Fig. S14, which are the results of our additional experiments with SR x4, x8.

For gaussian denoising, it may be difficult to expect the applicability of our Dilack since highly ill-posed cases with high noise levels have different ill-posedness from other tasks with kernels. Full investigation will be needed for these cases. Nonetheless, under Gaussian noise levels ($\sigma = 0.05$ to 0.1), our method performs comparable to DPS, though a more comprehensive investigation is required for these cases as a future work. The qualitative results of toy experiments for denoising on the FFHQ dataset can be found in Fig. S15, which are the results of our additional experiments.

## G.16 ADDITIONAL EXPERIMENTS UNDER DIFFERENT SEED VALUES

We evaluated DPS (Chung et al., 2023b), DiffPIR (Zhu et al., 2023), and Dilack on three sample images (indices 0–2) from the FFHQ and ImageNet datasets for lensless imaging with a Turing PSF and large motion deblurring.

For lensless imaging (Fig. S16 and S17), DPS generates diverse but inaccurate images due to its reliance on $\mathcal{L}_{LS}$, which struggles with this task. DiffPIR, based on $\mathcal{L}_{Pi}$, produces closer results but still shows notable errors. In contrast, our method generates consistently accurate images closely aligned with the ground truth (GT). For large motion deblurring (Fig. S20 and S19), DiffPIR performs

better but produces blurred outputs and inconsistent results across seeds. Our method, however, maintains consistency, with only minor noise artifacts.

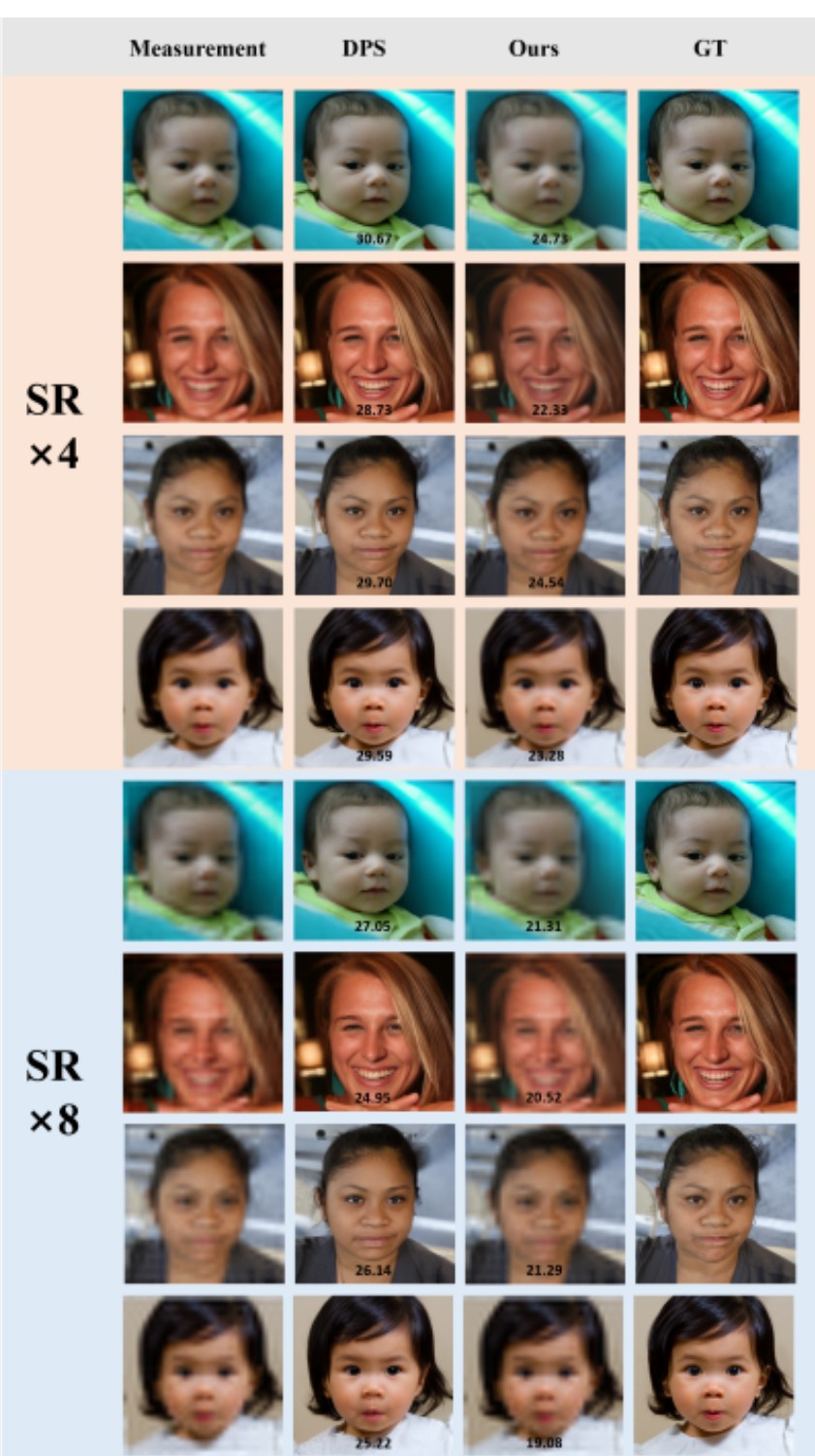

Figure S14: Qualitative comparison of the super-resolution task, as a preliminary evaluation of Dilack's potential extensibility to other applications.

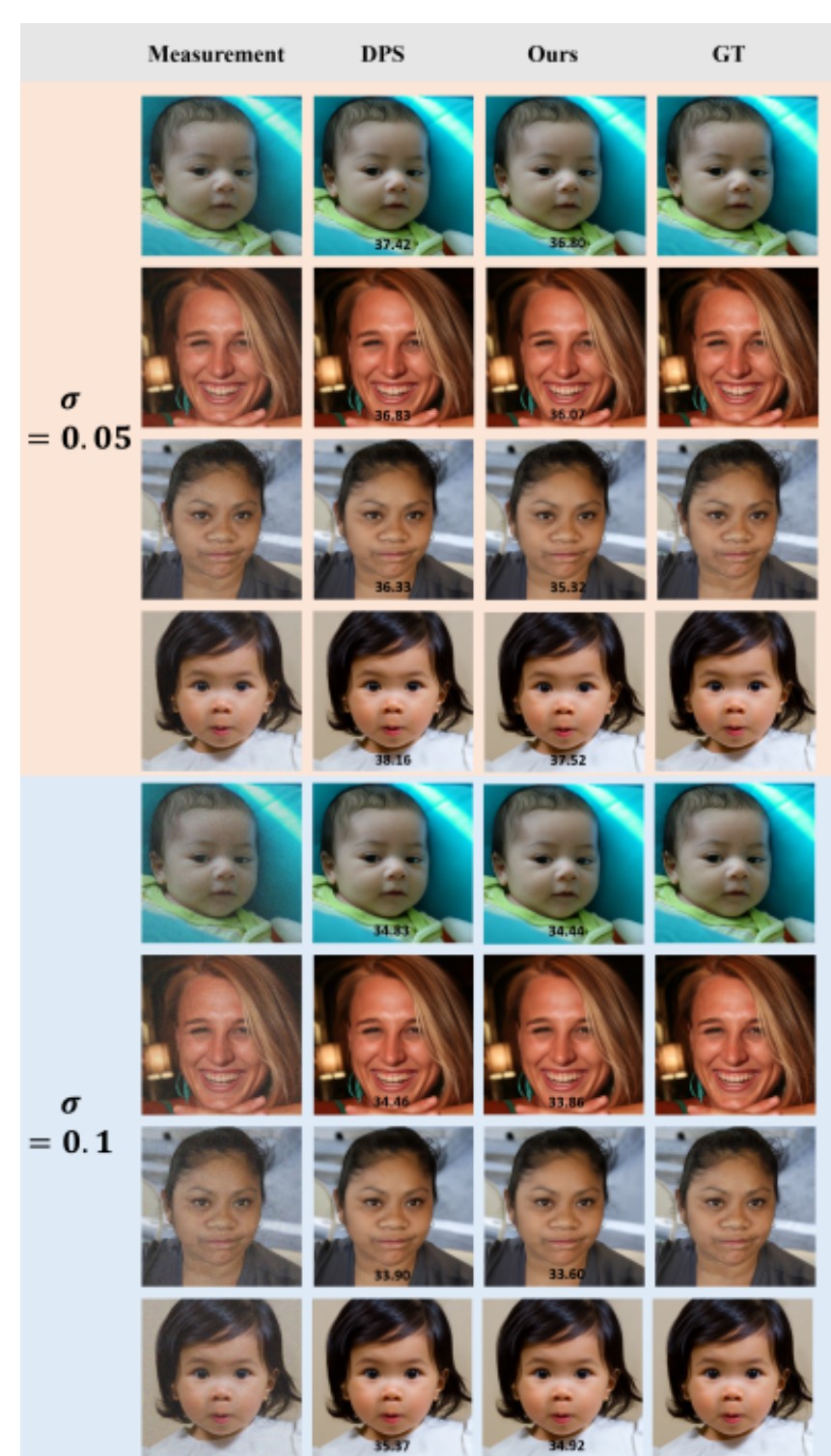

Figure S15: Qualitative comparison of the denoising task, as a preliminary evaluation of Dilack's potential extensibility to other applications.

All methods, including DPS, DiffPIR, DDPG, ReSample, and LDPS, were evaluated with fixed-seed sampling (*i.e.*, one sample per measurement) for PSNR. While a single sample may not fully capture performance for individual images, it provides reasonable aggregate results for the entire test dataset.

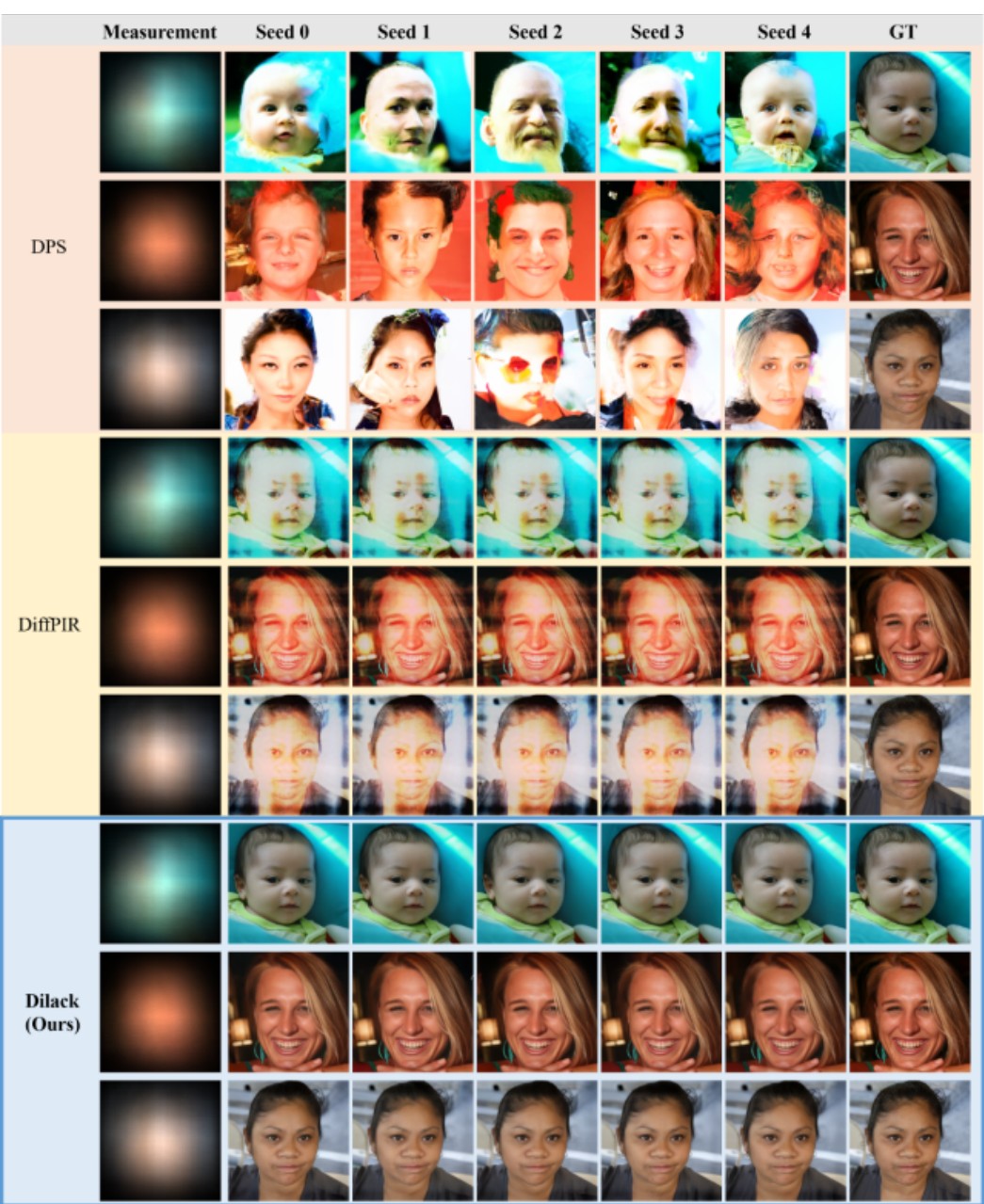

Figure S16: Additional qualitative results of the synthetic lensless imaging task using the Turing kernel on the FFHQ dataset under varying seed conditions.

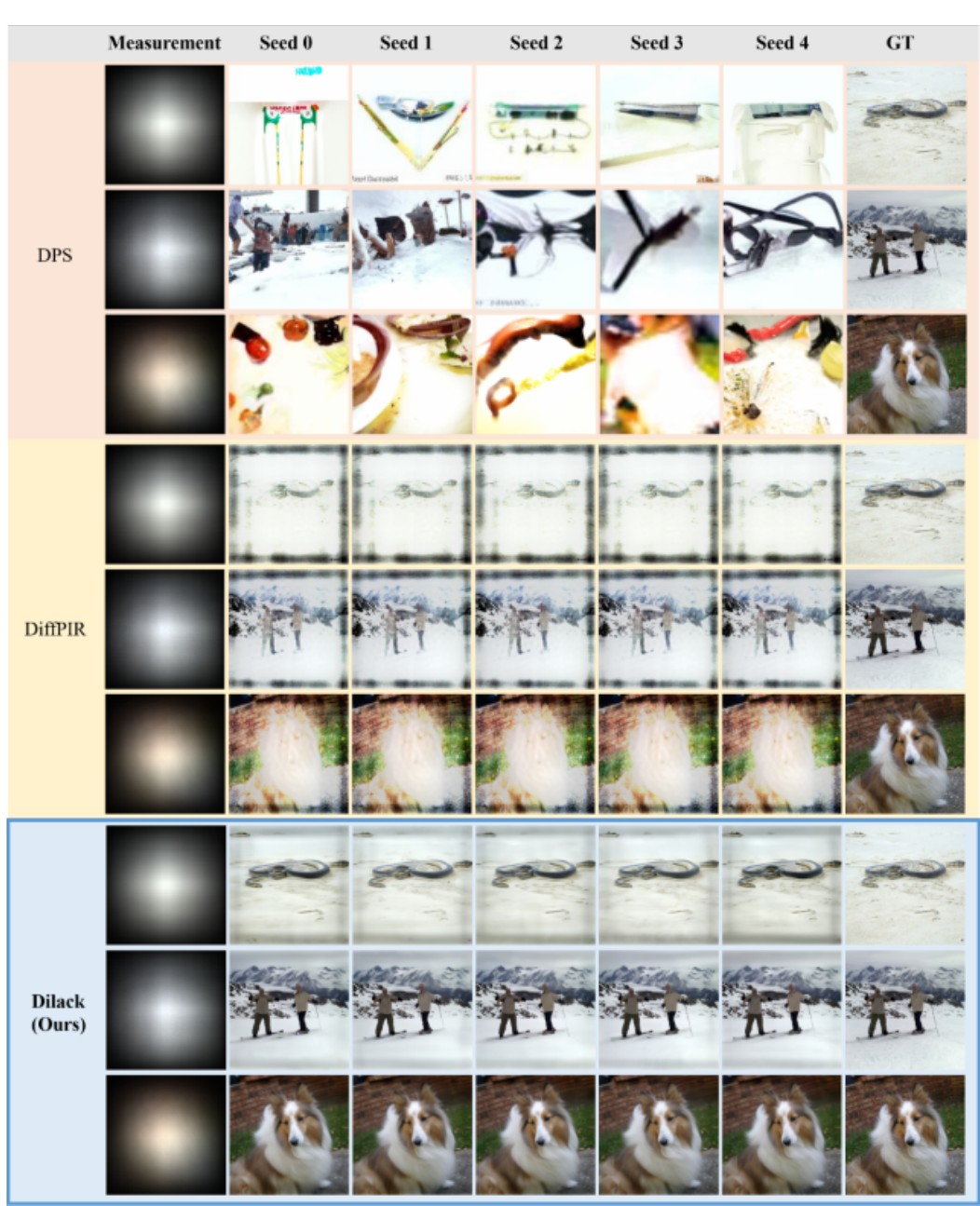

Figure S17: Additional qualitative results of the synthetic lensless imaging task using the Turing kernel on the ImageNet dataset under varying seed conditions.

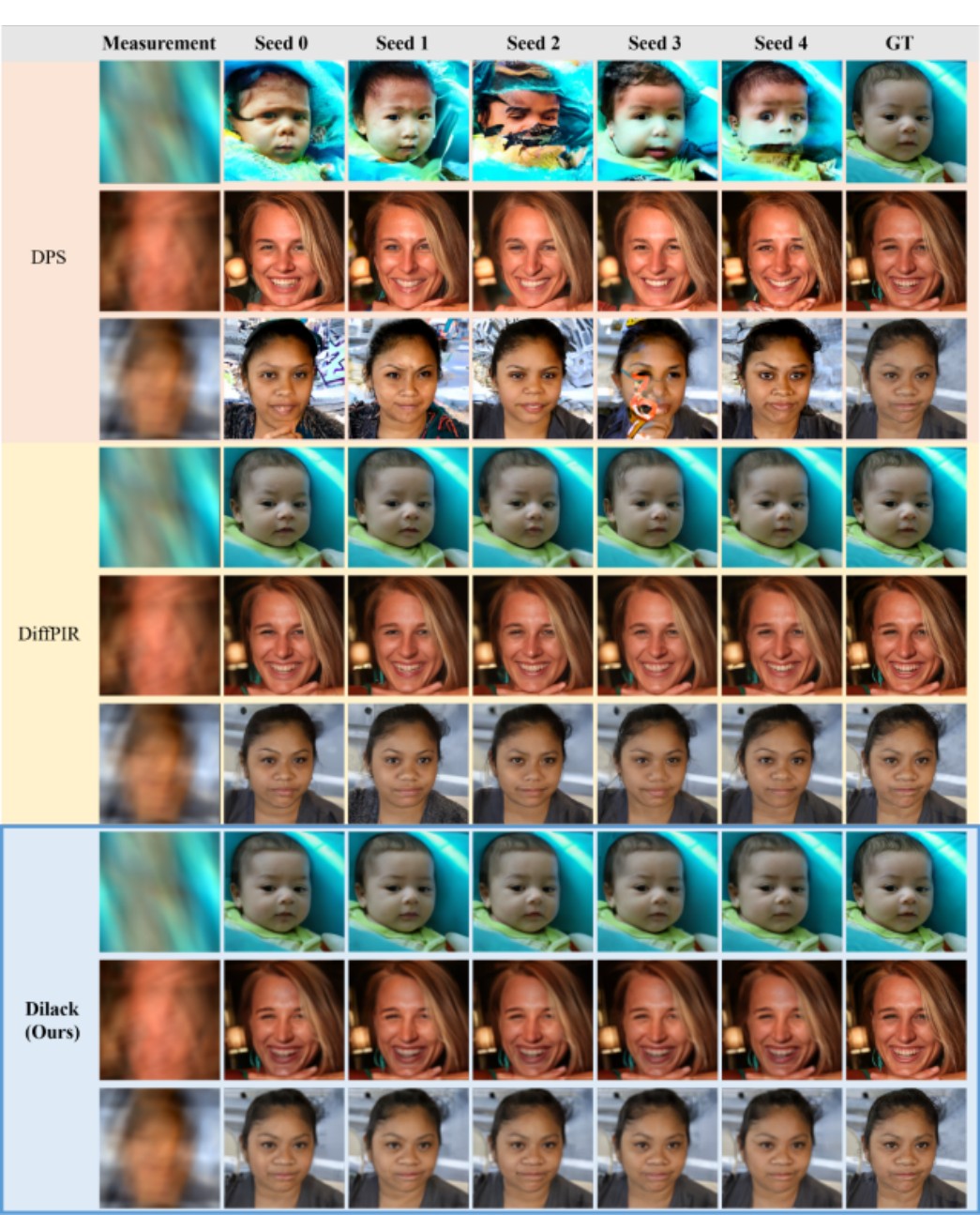

Figure S18: Additional qualitative results of large motion deblurring task on the FFHQ dataset under varying seed conditions. Note that, as discussed earlier in the main paper, large motion blur uses relatively simple kernel features, allowing existing methods using A† to perform well, but Dilack shows comparable results and our mask-guided approach outperforms pure $ADMM_{TV}$ in all aspects.

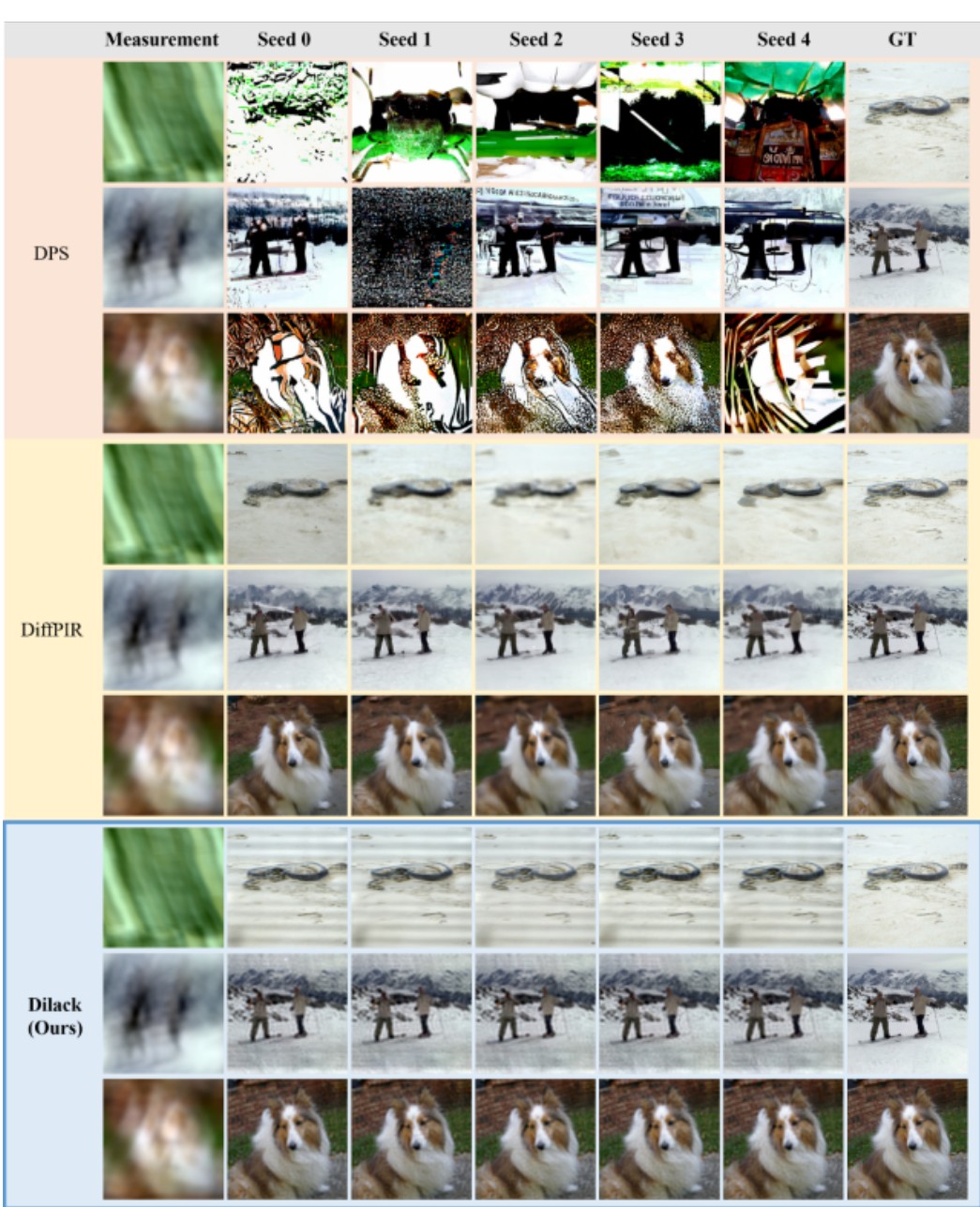

Figure S19: Additional qualitative results of large motion deblurring task on the ImageNet dataset under varying seed conditions. Note that, as discussed earlier in the main paper, large motion blur uses relatively simple kernel features, allowing existing methods using A† to perform well, but Dilack shows comparable results and our mask-guided approach outperforms pure ADMM$_{TV}$ in all aspects.

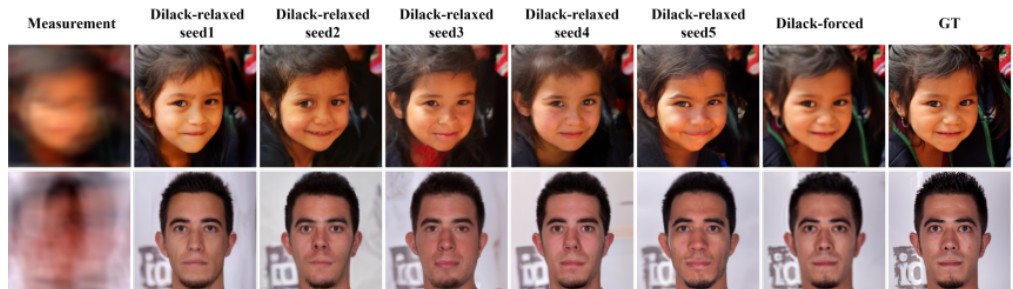

Figure S20: Diverse sampling outputs for the large motion deblurring task on the FFHQ dataset with relaxed fidelity constraints in our guidance settings. While Dilack prioritizes high fidelity over diversity, it remains fundamentally consistent with other zero-shot diffusion sampling methods.

### G.17 SAMPLING DIVERSITY UNDER RELAXED FIDELITY CONSTRAINTS

Our Dilack leverages the regularized data fidelity term, PiAC (Pseudo-inverse Anchor for Constraining), to achieve robust guidance in highly ill-posed problems. While this approach ensures high fidelity, it inherently results in lower output diversity. Nevertheless, our method remains fundamentally a "sampling" method, consistent with other zero-shot diffusion approaches.

To evaluate the diversity of outputs generated by our method, we adjusted the guidance scale $\rho$ to 0.5 and the masking ratio $\nu$ to 0.9, demonstrating the trade-off between fidelity and diversity. For this evaluation, we conducted experiments on the FFHQ dataset, testing with two randomly selected images as references while varying the sampling seed to analyze the diversity of generated outputs. As shown in Fig. S20, the outputs exhibit variations depending on the seed value, allowing for an evaluation of diversity.

## G.18 FURTHER QUALITATIVE RESULTS

In what follows, we present more qualitative results for our tasks: lensless imaging, large motion deblurring, and *real* lensless imaging. In severe ill-posed image inverse problems, and as the ill-posedness becomes more severe, Dilack demonstrates robustness unlike existing diffusion models, and provides more realistic results compared to classical methods. To the best of our knowledge, we are the first to approach mask-based camera reconstruction using a zero-shot diffusion model.

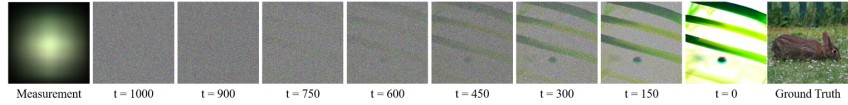

Figure S21: An example of DPS (Chung et al., 2023b)'s intermediate outputs per 150 iteration.

Figure S22: Additional qualitative results of synthetic lensless imaging task with Voronoi kernel on ImageNet dataset.

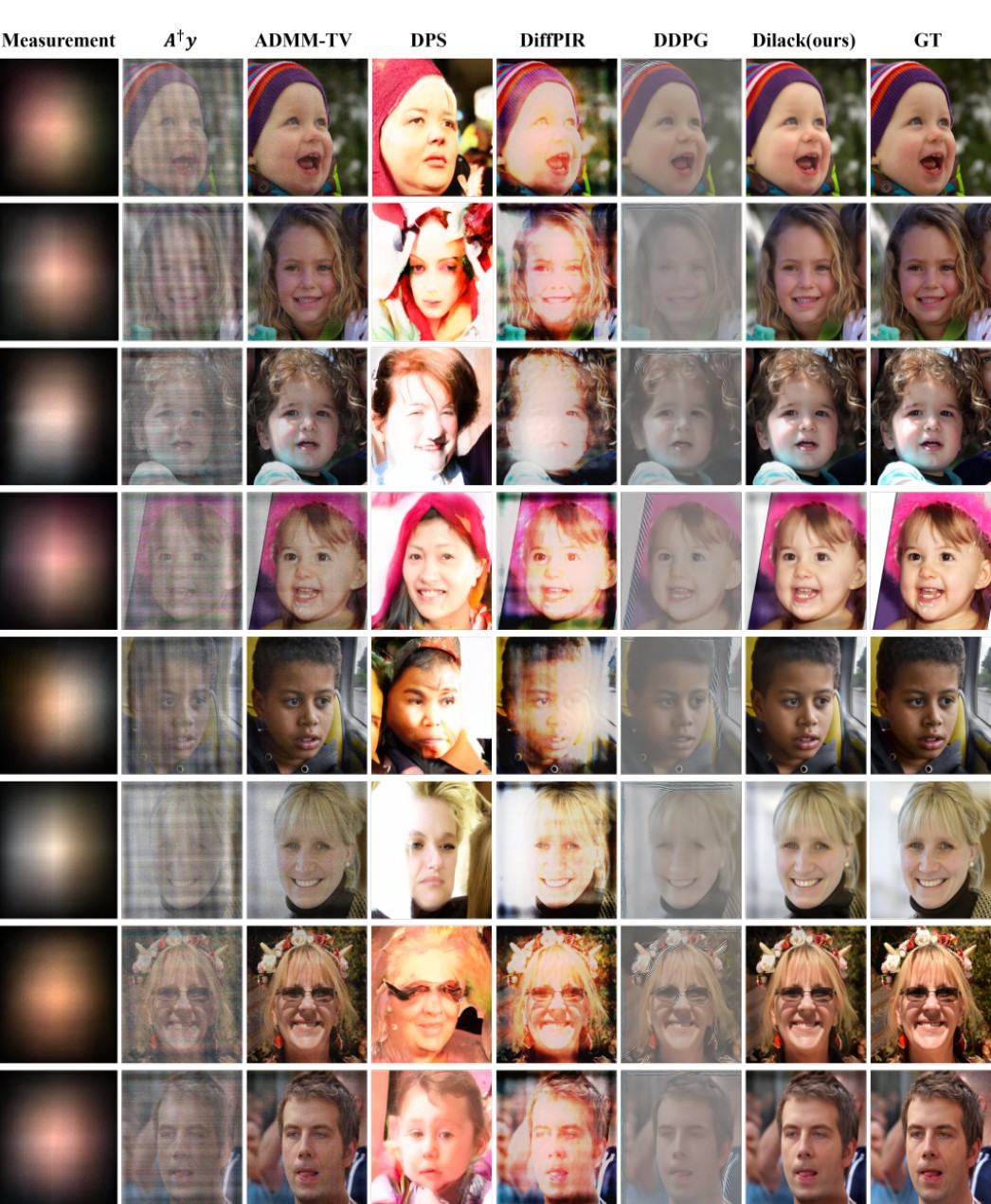

Figure S23: Additional qualitative results of synthetic lensless imaging task with Voronoi kernel on FFHQ dataset.

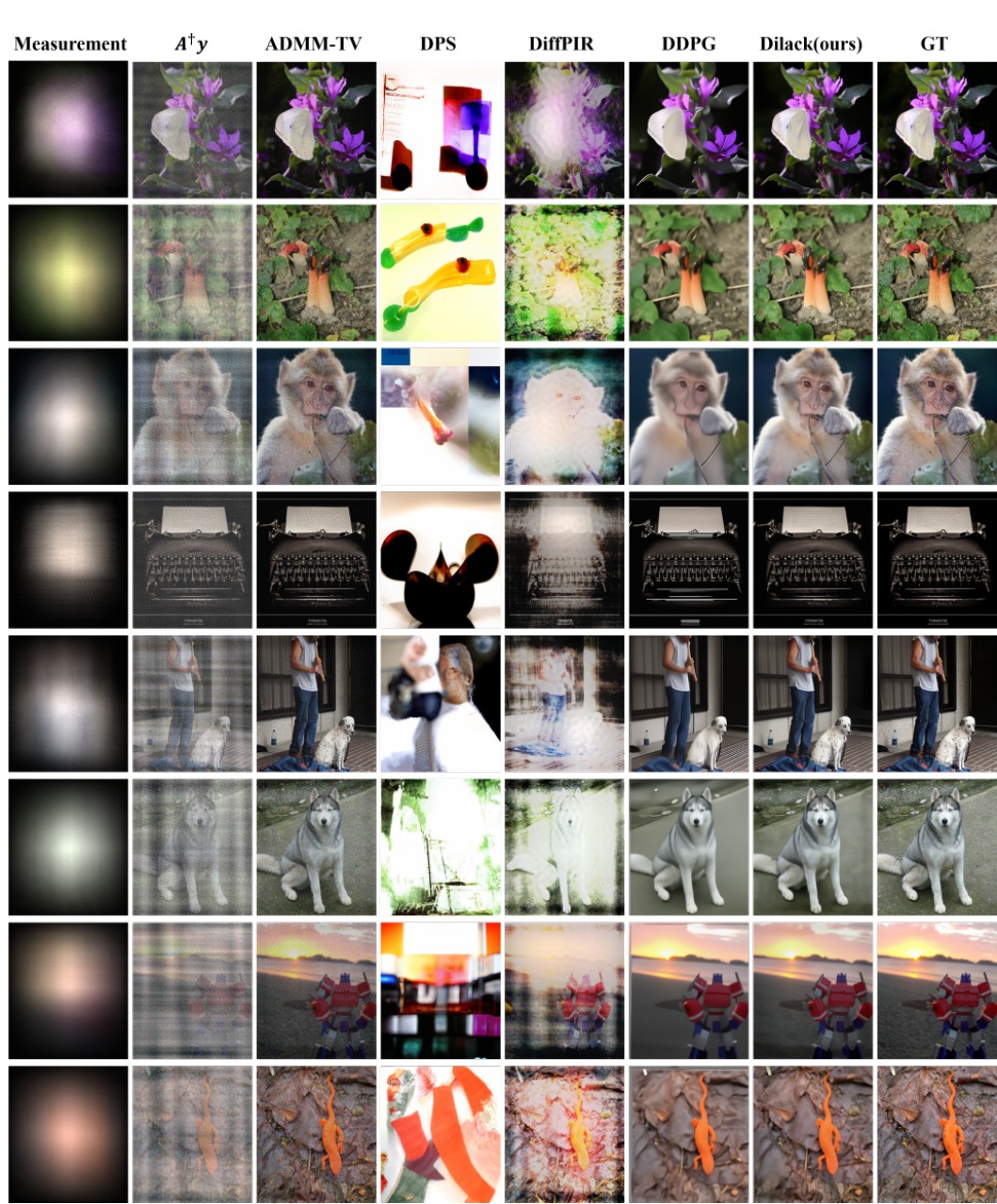

Figure S24: Additional qualitative results of synthetic lensless imaging task with Turing kernel on ImageNet dataset.

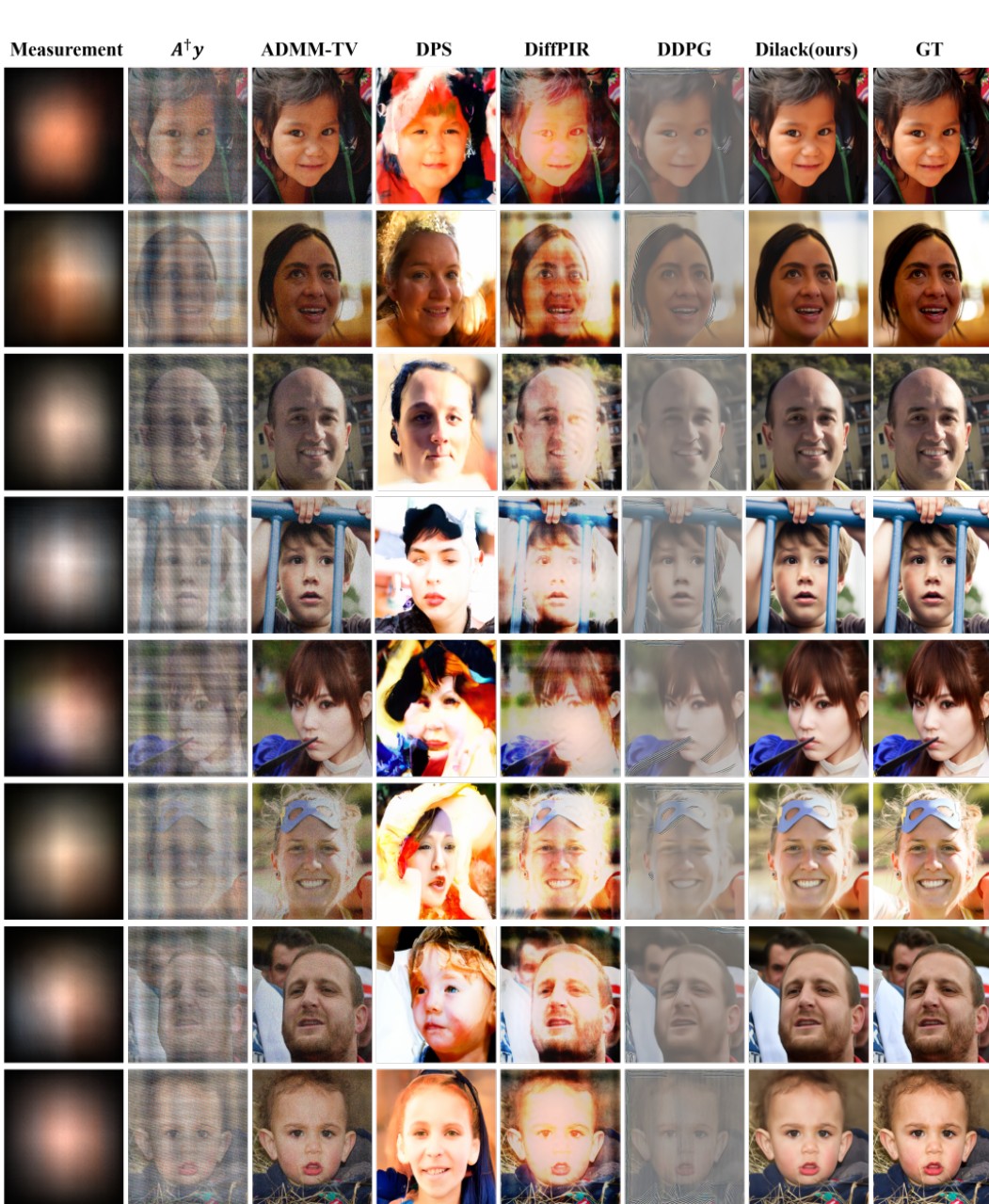

Figure S25: Additional qualitative results of synthetic lensless imaging task with Turing kernel on FFHQ dataset.

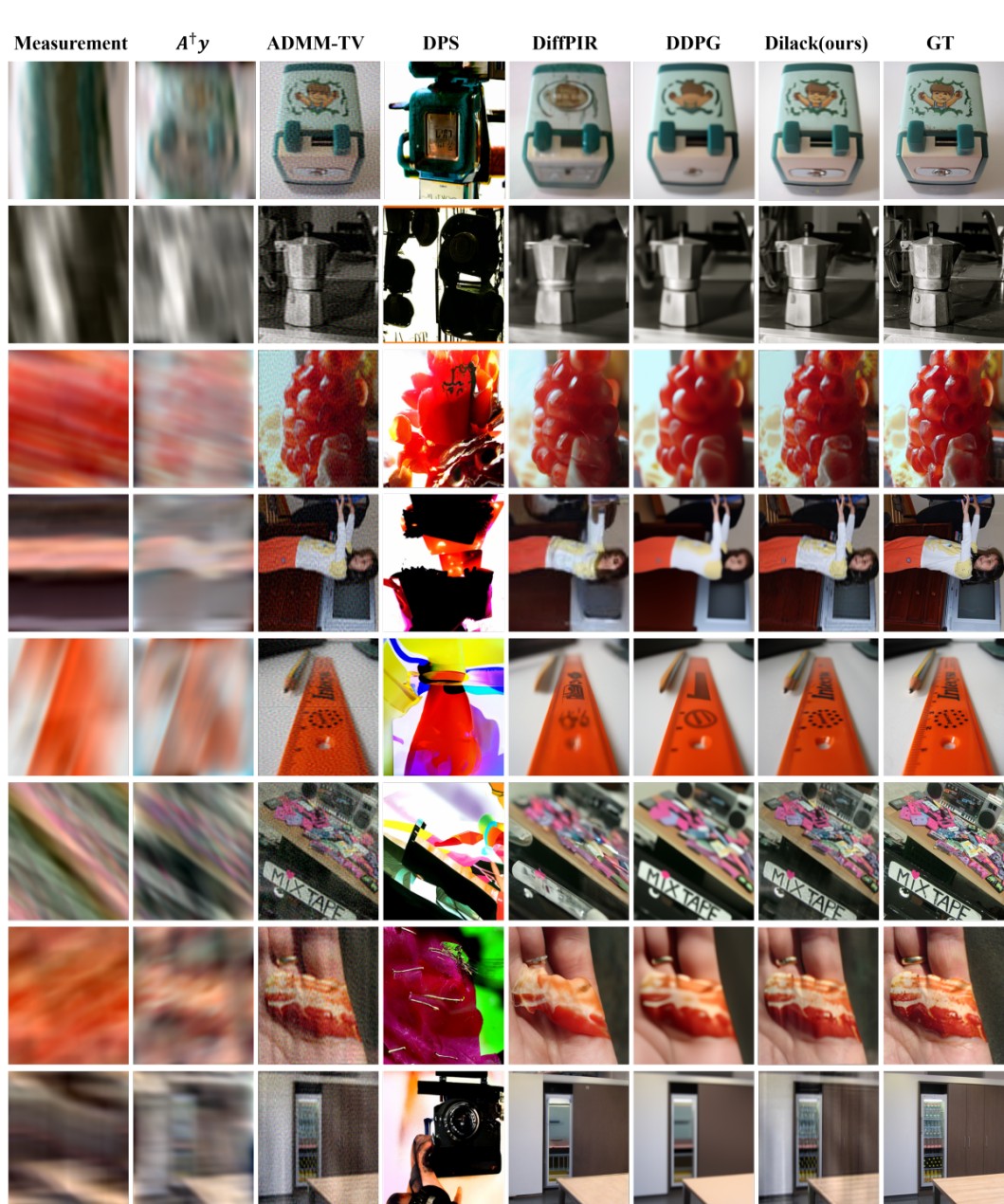

Figure S26: Additional qualitative results of large motion deblurring task on ImageNet dataset.

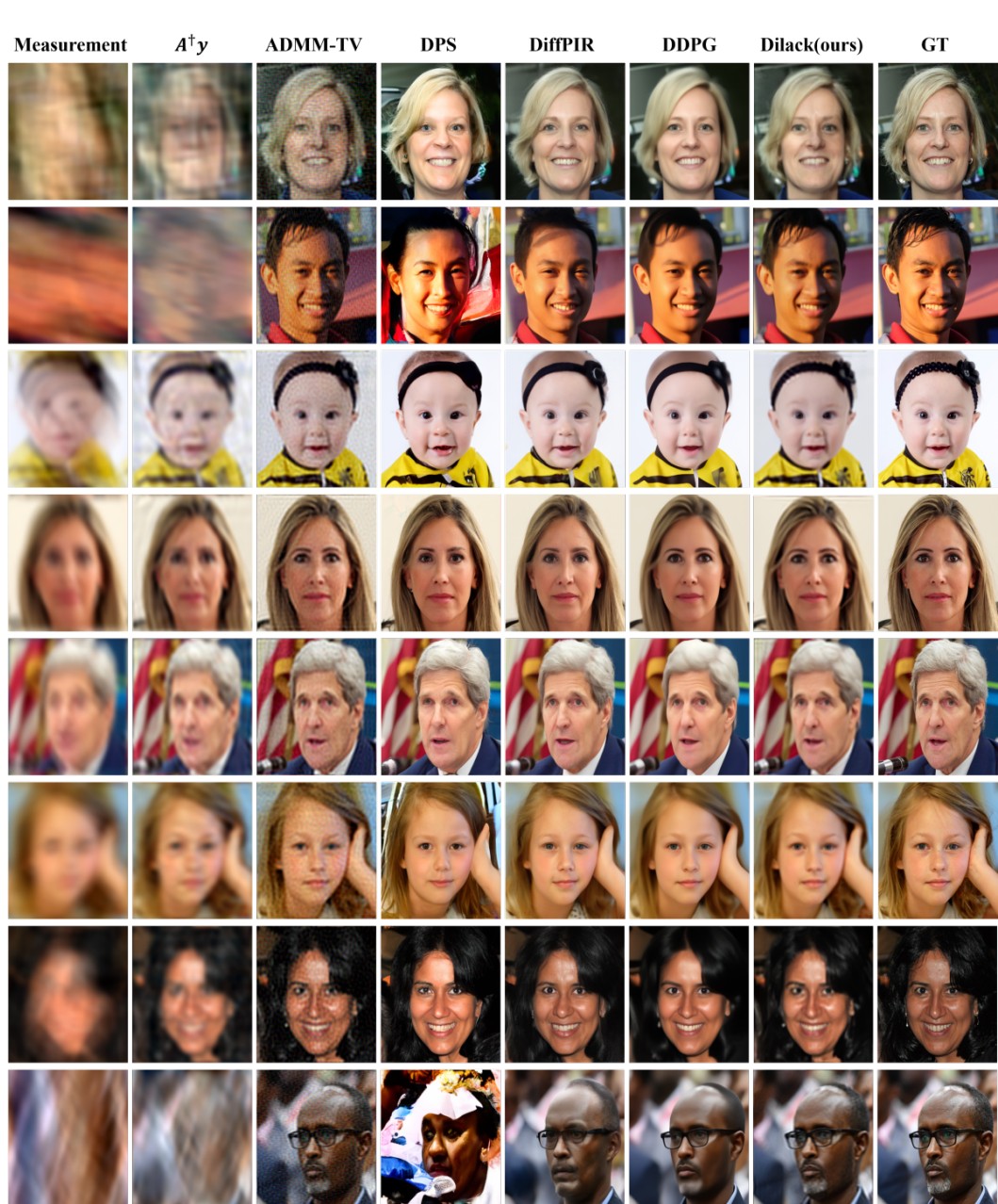

Figure S27: Additional qualitative results of large motion deblurring task on FFHQ dataset.

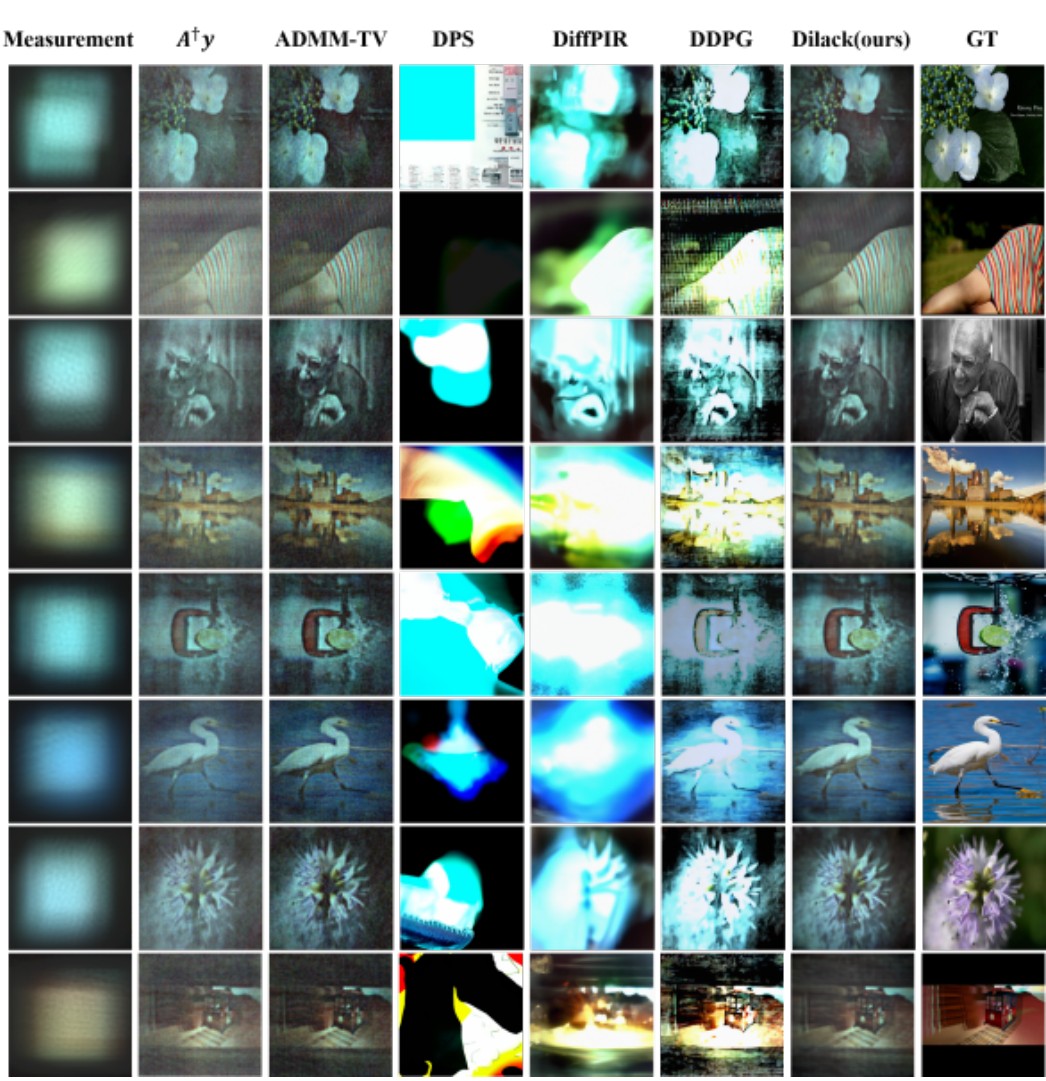

Figure S28: Additional qualitative results of *real* lensless imaging task on MirFlickr-lensless dataset.

