# OpenReview forum: "Masked, Regularized Fidelity With Diffusion Models For Highly Ill-posed Inverse Problems"
_ICLR.cc/2025/Conference — Submitted to ICLR 2025_

### Official Review · Reviewer_Tp52 · 2024-11-01

**Soundness:** 3
**Presentation:** 3
**Contribution:** 3
**Rating:** 6
**Confidence:** 4

**Summary:**

The paper proposes an improved diffusion posterior sampling algorithm for solving highly ill-posed inverse problems. By identifying inconsistent local regions and using it to guide some sparse iterations, the proposed method is capable of solving inverse problems in modern lensless imaging and large motion deblurring which most currently available diffusion-based posterior sampling algorithms fail to do.

**Strengths:**

1. The main contribution would be the extension of current diffusion-based posterior sampling algorithms on highly ill-posed inverse problems. The contribution is significant and groundbreaking.
2. The paper is well-written and has detailed theoretical deductions as well as comprehensive experiments.
3. The authors identify some critical challenges that current diffusion-based posterior sampling algorithms have.

**Weaknesses:**

1. In real-world applications (see Appendix G8), the restored images exhibit a smoother appearance compared to the ground truth. This suggests that there is still room for improvement in the number of steps used by the Dilack algorithm.
2. The computational complexity of the algorithm is quite high.

**Questions:**

1. How effective is the algorithm when applied to synthetic images and other tasks, such as super-resolution and denoising?
2. In the TV-regularized optimization component of the Dilack algorithm, is it feasible to perform just a single optimization step instead of seeking the minimum value? What would be the anticipated impact on overall performance?

---

> ### Author Response · Authors · 2024-11-22
> **Official Response by Authors**
>
> We would like to thank you for your constructive comments. See below for our point-by-point replies.
>
> > **[W1]**: "In real-world applications (see Appendix G8), the restored images exhibit a smoother appearance compared to the ground truth. This suggests that there is still room for improvement in the number of steps used by the Dilack algorithm."
>
> While the synthetic lensless imaging cases use the exact forward model, the real lensless imaging cases may often use the mismatched forward model, which makes the inverse problem much more challenging.
>
> > **[W2]**: "The computational complexity of the algorithm is quite high."
>
> **Appendix F.1** already discussed the computation cost, which is modestly increasing from 340 seconds to 390 seconds per image. However, note that our Dilack is practically the only solution that works for lensless imaging cases, so this modest increase can be justified.
>
> > **[Q1]**: "How effective is the algorithm when applied to synthetic images and other tasks, such as super-resolution and denoising?"
>
> For super-resolution (SR), $A$ matrix consists of a combination of a blur kernel and a downsampling operation. As the degree of downsampling increases, the condition number becomes larger, presenting a challenging scenario where the strengths of our proposed method are expected to be effective. However, highly ill-posed SR has not been well investigated in the field yet, so we believe that we need to carefully validate one by one. Especially, the impact of downsampling operator is worth investigating. Thus, at this moment, we can say that our Dilack can work for SR as compared to other prior works, but there is still room for improvement due to the reasons mentioned (see **Appendix G.15** and Fig. S14 (highlighted in *blue*), which are the results of our additional experiments with SR x4, x8).
>
> For denoising, it may be difficult to expect the applicability of our Dilack since highly ill-posed cases with high noise levels have different ill-posedness from other tasks with kernels. Full investigation will be needed for these cases. Nonetheless, under Gaussian noise levels ($\sigma$ = 0.05 to 0.1), our method performs comparable to DPS, though a more comprehensive investigation is required for these cases as a future work (see **Appendix G.15** and Fig. S15 (highlighted in *blue*), which are the results of our additional experiments).
>
> > **[Q2]**: "In the TV-regularized optimization component of the Dilack algorithm, is it feasible to perform just a single optimization step instead of seeking the minimum value? What would be the anticipated impact on overall performance?"
>
> Thank you for interesting suggestion. In our original Dilack, we ran 1000 ADMM iterations in line 8 of Alg 1 and this guidance anchor update occured G = 1 times. As suggested, we have conducted the cases with (1) 1 ADMM iteration (with the initial vector $x'_{t-1}$ from line 6) with 1000 guidance anchor updates (i.e., G=1000) and (2) 1 ADMM iteration with 1 guidance anchor update and the results are reported in the below table, demonstrating the superiority of our Dilack over other settings. It seems that initial guidance was very important especially for solving highly ill-posed inverse problems.
>
> | **Task**                | **Dataset** | **ADMM Iter.** | **G (# guid. anchor updates)** | **PSNR↑/SSIM↑/FID↓/LPIPS↓** |
> |-------------------------|-------------|-----------------|-----|----------------------------------------------|
> | **Lensless Imaging**    | **FFHQ**    | 1,000(ours) | 1(ours)           | **26.08** / **0.861** / **65.95**  / **0.152**      |
> |                         |             | 1 | 1,000                 | 16.23 / 0.601 / 267.89 / 0.428     |
> |                         |             | 1 | 1                  | 16.73     / 0.608     / 313.32   / 0.425      |
> |                         | **ImageNet**| 1,000(ours) | 1(ours)                | **24.36** / **0.788** / **80.99** / **0.247**      |
> |                         |             | 1 | 1,000                 | 15.68 / 0.523 / 235.04 / 0.465       |
> |                         |             | 1 | 1                  | 15.41     / 0.518     / 258.07     / 0.478      |
> | **Large Motion Deblurring** | **FFHQ** | 1,000(ours) | 1(ours)               | **22.97**     / **0.732**     / **197.59**     / **0.365**      |
> |                         |             | 1 | 1000                 | 14.88 / 0.489 / 259.68 / 0.581     |
> |                         |             | 1 | 1                  | 14.72     / 0.474     / 330.15     / 0.596      |
> |                         | **ImageNet**| 1,000(ours) | 1(ours)                | **20.61** / **0.596** / **200.81** / **0.477**      |
> |                         |             | 1 | 1,000                 | 14.22 / 0.386 / 293.76 / 0.621      |
> |                         |             | 1 | 1                  | 13.34     / 0.368     / 337.58     / 0.645      |

---

> > ### Comment · Reviewer_Tp52 · 2024-11-30
> >
> > I thank the authors for the additional experiments. However, I find the results in Fig. S14 not satisfactory, even for the common SR$\times 4$ task. Thus, I decide to maintain my current evaluation.

---

> > > ### Author Response · Authors · 2024-11-30
> > >
> > > Thank you so much for your valuable comments. Indeed, our results in Fig S14 do not look good. While it was expected to have some degradation for these extra cases of SR x4, x8 due to our approximation to improve the robustness of the data fidelity term for highly ill-posed inverse problems, these current results were not explained since our approximation is still reasonably accurate to match the original data fidelity term.
> > >
> > > We found that we accidentally put an additional blur operator for generating measurements, leading to severely degraded results (i.e., DPS solved easier inverse problem over ours in the experiment of Fig. S14). Our initial results showed that the average PSNRs / SSIMs of DPS and our Dilack for SR x4 with a mild noise were 28.7 dB / 0.797 and 28.5 dB / 0.803, respectively (using 4 random seeds), implies that our Dilack yielded comparable performance to DPS for SR problems. We will put full results for SR x4, x8 before the author response deadline and we hope that you reconsider your decision after looking into our corrected results for Fig. S14.

---

> ### Author Response · Authors · 2024-12-01
>
> As promised, we have updated our corrected results for Fig. S14 as described below (Figures will be updated in the final version). Here are the average PSNR and SSIM for 4 images (data indices 0–3) using five different random seeds for FFHQ super-resolution task (x4 and x8) with Gaussian noise ($\sigma$ = 0.05), following the same experimental settings as DPS. Our proposed Dilack yielded comparable performance to the prior arts (DPS).
>
> **Table. Super resolution x4 and x8 experiments**
>
> | Task | DPS | Ours |
> | -------- | -------- | -------- |
> | SR x4   | 28.34/0.794    | 28.47/0.806     |
> | SR x8   | 24.76/0.689    | 24.97/0.686     |

---

> > ### Author Response · Authors · 2024-12-02
> > **Additional Comment by Authors**
> >
> > We kindly remind you that only one day remains in the rebuttal period. Thank you for your feedback and further response. Please review the additional results and comments above and consider them when assigning your final score. Feel free to share any further concerns!

---

### Official Review · Reviewer_gLcf · 2024-11-03

**Soundness:** 2
**Presentation:** 2
**Contribution:** 2
**Rating:** 5
**Confidence:** 4

**Summary:**

In this paper, the authors introduce a new data-fidelity term for solving challenging inverse problems using diffusion models. Noting the limitations of existing approaches on highly ill-posed problems, they propose replacing the standard $L_2$  data-fidelity term with the $L_2$ distance between the solution obtained by regularizing the inverse problem with a TV prior and the posterior mean derived from the diffusion denoiser. They demonstrate the effectiveness of this approach through experiments on lensless imaging and large-motion deblurring tasks.

**Strengths:**

- The main idea, which is to replace the pseudo-inverse by a TV-regularized solution, is interesting. The approach of modifying the data-fidelity term is flexible and could potentially be adapted for other types of priors, opening doors for future work on similar challenging inverse problems.
- By using the posterior mean from a diffusion model, and the ADMM prior, the paper effectively combines generative modeling with traditional regularization, a creative fusion of modern and classical techniques.
- The paper provides comprehensive experiments on nontrivial tasks, such as lensless imaging and deblurring, that highlight the practical advantages of the proposed method.

**Weaknesses:**

- **Readability Issues**: The paper is sometimes challenging to read due to long, complex sentences, particularly in the abstract and Remark 1. Simplifying these sections could improve readability and flow.

- **Sampling vs. Maximization**: The method proposed here, like DPS and DiffPIR, is a **sampling** method aimed at sampling from the posterior distribution. However, this crucial point seems to be overlooked or misunderstood by the authors, as the paper appears to focus on solving equation (2) for posterior maximization rather than posterior sampling.

- **Explanation of $x$ Replacement in Equation (14)**: It is unclear why $x$ in line 298 is replaced by $x_{0|t}$ in equation (14). Why is the data-fidelity term $|| x_{TV} - x ||^2$ not used instead? The new data-fidelity term appears significantly different from $L_{PI}$, making it unclear whether a comparison between the two terms is justified. On what basis can Equation (15) be considered approximately true? Has it been verified, for example, for simpler inverse problems that $L_{PI}$ approximates $L_{PIAC}$?

- **Comparison with State-of-the-Art Methods**: It would be beneficial to compare the proposed approach with other state-of-the-art methods in image inverse problems, such as Plug-and-Play (PnP) methods like DPIR.

- **Comparison with DPS Method**: The comparison with DPS may be unfair, as DPS’s performance could likely be improved with step-size optimization. This omission may explain DPS’s lower performance in the experiments.

- **Mismatch Between Equation (17) and Algorithm 1**: Equation (17) does not match the steps in Algorithm 1 due to the addition of noise in the algorithm.

- **PSNR Comparison in Posterior Sampling Context**: Since the goal is to sample from the posterior distribution, comparing PSNR values may not be meaningful unless comparing the posterior means. Are PSNR values in the paper calculated based on a single sample from each method?

- **Algorithm Structure for Clarity**: Since \( G = 1 \) is used in practice, restructuring the algorithm into two main steps—1) ADMM-TV and 2) Diffusion—could improve clarity and make the methodology easier to understand.

**Questions:**

1. **Consistency in Comparisons**: Why do Tables 2 and 3 compare against different methods? Consistently using the same methods across both tables would allow for a more direct comparison of results.

2. **Code Release**: Is there a plan to release the code for this method?

3. **Clarification on Appendix Statement about ADMM and Regularization**: The sentence in the Appendix, “Incorporating Total Variation (TV) regularization into the Alternating Direction Method of Multipliers (ADMM) enhances the algorithm’s performance across various applications,” is somewhat misleading. It is not the use of TV specifically but rather the addition of a regularization term that enables the solution of the inverse problem. Additionally, any proximal splitting algorithm—not just ADMM—could be used for this purpose. Could you clarify this?

4. **Applicability of Data-Fidelity Term with Other Priors**: If the primary contribution is to modify the data-fidelity term, could this approach work with priors other than diffusion models? This generalization could broaden the method’s applicability.

5. **Comparison with Standard \( L_2 \) Data-Fidelity Term**: Would it be useful to compare your method to an approach that minimizes the standard $L_2$ data-fidelity term combined with a diffusion prior and a TV prior?

6. **Gradient Calculation Method**: For calculating the gradient of your data-fidelity term, do you use automatic differentiation? If so, does this require backpropagation through the diffusion model? If yes, is this process computationally intensive, and how does it impact runtime?

Raising my score to 5 after the discussion period.

---

> ### Author Response · Authors · 2024-11-22
> **Official Response by Authors, Part 1/4**
>
> We would like to thank you for your constructive comments. See below for our point-by-point replies.
>
> > **[W1]**: "**Readability Issues**: The paper is sometimes challenging to read due to long, complex sentences, particularly in the abstract and Remark 1. Simplifying these sections could improve readability and flow."
>
> As suggested, we revised the abstract and Remark 1, highlighted all modifications in *blue* text.
>
> > **[W2]**: "**Sampling vs. Maximization**: The method proposed here, like DPS and DiffPIR, is a sampling method aimed at sampling from the posterior distribution. However, this crucial point seems to be overlooked or misunderstood by the authors, as the paper appears to focus on solving equation (2) for posterior maximization rather than posterior sampling."
>
> We also exploited the posterior sampling framework like DPS and DiffPIR, but for the highly ill-posed problems like modern lensless imaging and large motion deblurring. However, these challenging inverse problems have quite large null space, so that naive sampling usually provide images far from GT. Our Dilack implicitly narrowed down this null space by using TV so that more faithful images to the measurement can be generated. Our Dilack has a similar aspect with [Chung et al., 2022b] (regularized fidelity in our Dilack vs. constrained fidelity in [Chung et al., 2022b]), but with a number of differences in our fidelity term design.
>
> Note that DPS and DiffPIR also generated the experimental results with fixed-seed sampling like ours, so the comparisons in this work is fair. However, to relieve your concern, we have conducted additional experiments on sampling diversity, which are detailed in **Appendix G.15** of the revised manuscript (highlighted in *blue*).
>
> > **[W3]**: "**Explanation of $x$ Replacement in Equation (14)**: It is unclear why $x$ in line 298 is replaced by $x_{0|t}$ in equation (14). Why is the data-fidelity term $||x_{TV} - x||^2$ not used instead? The new data-fidelity term appears significantly different from $L_{Pi}$, making it unclear whether a comparison between the two terms is justified. On what basis can Equation (15) be considered approximately true? Has it been verified, for example, for simpler inverse problems that $L_{Pi}$ approximates $L_{PiAC}$?"
>
> If the term in line 298 (of the initial manuscript) is used in the context of posterior sampling, $x$ becomes $x_t$. Then, it is straightforward (*e.g.*, consider $\tilde{x}^*$ as a measurement vector and the identity operator as a measurement operator) to apply Theorem 1 of DPS [Chung et al., 2022a], leading to the term $|| \tilde{x}^* - \hat{x}_{0|t}||^2$ as described. Note that the Theorem 1 of DPS does not have to have a mildly ill-posed inverse problems - even for highly ill-posed problems, the forward model does not yield drastic change in the measurement, so the theorem works.
>
> The relationship between $L_{Pi}$ and $L_{PiAC}$ looks interesting. There are two components to consider. One is $A^\dagger y$ where $L_{Pi}$ uses a minimum norm solution and $L_{PiAC}$ uses a TV solution, leading different maximum a posteriori (MAP) estimations. The other is $A^\dagger A$ where $L_{Pi}$ uses it as it is and $L_{PiAC}$ approximates it as an identity. The former only compares the values in the projected spaces while the latter compares the values in the whole spaces including the null space. Our Dilack sacrifies the aspect of measuring in the projected spaces for highly ill-posed inverse problems, but instead we approximate it by comparing the values in the null space that was filled by TV regularization.
>
> > **[W4]**: "**Comparison with SOTA Methods**: It would be beneficial to compare the proposed approach with other state-of-the-art methods in image inverse problems, such as Plug-and-Play (PnP) methods like DPIR [1]."
>
> As suggested, we have conducted additional experiments using DPIR pre-trained model, have evaluated it on lensless imaging (Turing PSF) trained with ImageNet and confirmed that our Dilack ourperformed DPIR by large margin. The quantitative results are summarized in the below table.
>
> | **Method**         | **PSNR↑**/**SSIM↑**/**FID↓**/**LPIPS↓** |
> |---------------------|---------------------|
> | **DPIR** [1]       | 9.28   /0.265   / 236.07     / 0.552     |
> | **Dilack (Ours)**  | 24.88    / 0.796    / 92.89     / 0.208     |
>
> [1] Zhang et al., "Plug-and-play image restoration with deep denoiser prior," IEEE T-PAMI 2021.

---

> ### Author Response · Authors · 2024-11-22
> **Official Response by Authors, Part 2/4**
>
> > **[W5]**: "**Comparison with DPS Method**: The comparison with DPS may be unfair, as DPS’s performance could likely be improved with step-size optimization. This omission may explain DPS’s lower performance in the experiments."
>
> To relieve your concern, we have conducted additional experiments using our two tasks on 100 sample images from the FFHQ dataset as shown in the below table, confirming that varying the step-size did not signifcantly improve the performance of DPS. Quantitative results and further details on this experiment have been included in **Appendix G.10** of the revised manuscript (highlighted in *blue*).
>
> - Ablation study on step-size effect on **DPS**
>
> | Task | Dataset | Step-size | PSNR↑/SSIM↑/FID↓/LPIPS↓      |
> |-------------------------------|---------|-----------|-----------------------------|
> | **Lensless Imaging (w/ Turing)** | **FFHQ** | 0.25  | 11.20/0.385/120.83/0.514      |
> |                               |         | 0.5       | 10.71/0.382/134.62/0.528     |
> |                               |         | 0.75      | 10.34/0.376/136.65/0.541     |
> |                               |         | 1.0       | 9.95/0.369/150.88/0.558      |
> | **Large Motion Deblurring**    | **FFHQ** | 0.25    | 17.70/0.502/94.48/0.364       |
> |                               |         | 0.5       | 17.99/0.514/91.80/0.348       |
> |                               |         | 0.75      | 17.87/0.512/96.66/0.347      |
> |                               |         | 1.0       | 17.85/0.511/99.10/0.349       |
>
> > **[W6]**: "**Mismatch Between Equation (17) and Algorithm 1**: Equation (17) does not match the steps in Algorithm 1 due to the addition of noise in the algorithm."
>
> Thank you for indicating this mismatch. We will revise it as below to address this concern.
>
> **Revised Equation (17):**
>
> \$\nabla\_{\mathbf{x}\_t} \log p\_t(\mathbf{x}\_t|\mathbf{y}) \simeq \mathbf{s}\_{\theta^*}(\mathbf{x}\_t, t) - \rho\, \mathcal{M}\_{\text{t}} \left[ \nabla\_{\mathbf{x}\_t} \|\| \tilde{\mathbf{x}}^{*} - \hat{\mathbf{x}}\_{0|t}\~\|\|_2^2 \right].\$
>
>
> See the response to [W3] for the justification on replacing $x_t$ with $x_{0|t}$.
>
> For your information, the exact equation for $\mathbf{x}_{t-1}$ with respect to $\mathbf{x}_t$, considering the noise addition pointed out by the reviewer, is given as follows:
>
> \$\mathbf{x}\_{t-1} \simeq \frac{1}{\sqrt{\alpha\_t}} \mathbf{x}\_t + \frac{1 - \alpha\_t}{\sqrt{\alpha\_t}} \mathbf{s}\_{\theta^*}(\mathbf{x}\_t, t) + \tilde{\sigma}\_t \mathbf{z} - \rho \mathcal{M}\_{\text{t}} \left[ \nabla\_{\mathbf{x}\_t} \left\| \tilde{\mathbf{x}}^{*} - \hat{\mathbf{x}}\_{0|t} \right\|_2^2 \right],\$
>
> where:
>
> - $\alpha_t$ is the **Noise Scaling Coefficient**,
> - $\tilde{\sigma}_t$ is the **Diffusion Adjustment Coefficient**,
> - $\mathbf{z}$ is Gaussian noise with mean 0 and variance equal to the identity matrix.
>
> > **[W7]**: "**PSNR Comparison in Posterior Sampling Context**: Since the goal is to sample from the posterior distribution, comparing PSNR values may not be meaningful unless comparing the posterior means. Are PSNR values in the paper calculated based on a single sample from each method?"
>
> We would like to point out that all prior works including DPS, DiffRIR, DDPG, ReSample and LDPS were consistently evaluated in PSNR with fixed-seed sampling (*i.e.*, a single sample per measurement) and we also stick to this setting. While a single sample may not provide a reliable value for one image, single samples for the entire test dataset seem to provide reasonable results by aggregating them.
>
> However, to relieve your concern, we have included additional results with varied seed values in **Appendix G.15** of the revised manuscript.
>
> > **[W8]**: "**Algorithm Structure for Clarity**: Since (G = 1) is used in practice, restructuring the algorithm into two main steps—1) ADMM-TV and 2) Diffusion—could improve clarity and make the methodology easier to understand."
>
> Thank you for the suggestion. We have revised our manuscript in a simpler way as suggested by benchmarking ReSample [Song et al., 2024] that enforces hard consistency in the beginning of diffusion process.

---

> ### Author Response · Authors · 2024-11-22
> **Official Response by Authors, Part 3/4**
>
> > **[Q1]**: "**Consistency in Comparisons**: Why do Tables 2 and 3 compare against different methods? Consistently using the same methods across both tables would allow for a more direct comparison of results."
>
> Note that the compared prior works were developed based on either pixel-space diffusion model or latent-space diffusion model. Thus, we grouped prior works into two and compared with one group or the other by fixing the diffusion model, reporting the results in Tables 2 and 3, respectively. This separation was our effort not to alter the original prior works and to demonstrate our Dilack working with both pixel-space and latent-space diffusion models without training or tuning.
>
> > **[Q2]**: "**Code Release**: Is there a plan to release the code for this method?"
>
> Yes, we plan to release. In fact, as noted in **Appendix F.1**, we have already submitted the Dilack code as part of the supplementary materials.
>
> > **[Q3]**: "**Clarification on Appendix Statement about ADMM and Regularization**: The sentence in the Appendix, “Incorporating Total Variation (TV) regularization into the Alternating Direction Method of Multipliers (ADMM) enhances the algorithm’s performance across various applications,” is somewhat misleading. It is not the use of TV specifically but rather the addition of a regularization term that enables the solution of the inverse problem. Additionally, any proximal splitting algorithm—not just ADMM—could be used for this purpose. Could you clarify this?"
>
> Thank you for raising this. It seems that the sentences in lines 899-905 may be misleading and are not essential, so we removed them in the revision. ADMM and TV are somewhat independent components, one is for optimization and the other is for a regularizer in the cost function. Thus, TV should be incorporated into the cost function for inverse problem and ADMM can solve that inverse problem.
>
> However, it is still true that the proposed combination of TV and ADMM, or $ADMM\_\{TV\}$, is quite effective for highly ill-posed inverse problems over other combinations. To demonstrate it, we have conducted additional experiments with different optimizations / regularizers in PiAC guidance for Dilack for the lensless (turing kernel) imaging on the FFHQ dataset (sampled 100 images) as summarized in the below table. This table demonstrated that our proposed Dilack with $ADMM\_\{TV\}$ consistently outperformed Dilack with other combinaions such as ADMM-$L_1$, ADMM-$L_2$, FISTA (Fast Iterative Soft-Thresholding Algorithm) with $L_1$ and HQS (Half-Quadratic Splitting) with TV.
>
> - Quantitative Comparison of Dilack and Its Variants with Different Regularizers & Optimization Algorithms
>
> | **Method** in PiAC Guidance + Dilack  | **PSNR↑/SSIM↑/FID↓/LPIPS↓** |
> |---------------------|----------|
> | **FISTA-$L_1$ + Dilack**   | 19.30/0.745/98.06/0.249 |
> | **HQS-TV + Dilack**   | 12.51/0.341/363.52/0.637 |
> | **ADMM-$L_1$ + Dilack**        | 16.11/0.672/126.69/0.301  |
> | **ADMM-$L_2$ + Dilack**        | 22.69/0.817/64.25/0.183  |
> | **ADMM-TV + Dilack (ours, original Dilack)**   | **26.23/0.863/54.84/0.149**      |
>
> > **[Q4]**: "**Applicability of Data-Fidelity Term with Other Priors**: If the primary contribution is to modify the data-fidelity term, could this approach work with priors other than diffusion models? This generalization could broaden the method’s applicability."
>
> The primary contribution of this work is the methods to leverage pre-trained diffusion models for solving highly ill-posed inverse problems from real lensless imaging or large motion blurring. While one of our core contributions is to propose a data-fidelity term, note that proposing good approximate measurement matching terms has been important in a number of recent works - see Table 1 / Figure 1 of a recent survey paper [2]. All fidelty designs were taylored to working with diffusion models. Thus, it is unclear if our proposed fidelity term is indeed generalizable.
>
> However, to alleviate your concern, we have applied our guidance setting using a CNN denoiser (following DPIR [1]), but it did not yield poor performance as expected. While this result was not in the revision, we can incorporate that into the next revision if needed.
>
> [1] Zhang et al., "Plug-and-play image restoration with deep denoiser prior," IEEE T-PAMI 2021.
> [2] Daras et al., A Survey on Diffusion Models for Inverse Problems, arXiv 2024.

---

> > ### Author Response · Authors · 2024-11-22
> > **Official Response by Authors, Part 4/4**
> >
> > > **[Q5]**: "**Comparison with Standard ( L_2 ) Data-Fidelity Term**: Would it be useful to compare your method to an approach that minimizes the standard $L_2$ data-fidelity term combined with a diffusion prior and a TV prior?"
> >
> > Note that DPS [Chung et al., 2022a] is the standard $L_2$ data-difelity term with a diffusion prior and our ADMM$_{TV}$ is the standard $L_2$ data-difelity term with a TV prior.
> >
> > To address this comment, we have conducted experiments combining the standard $L_2$ data fidelity term with a TV prior in both DPS and DiffPIR frameworks using a synthetic lensless imaging task with 100 images from ImageNet as shown below. The results confirmed that simply appending a TV regularizer to the $L_2$ term does not work to yield good performance. The quantiative and qualitative comparison results can be found in **Appendix G.12**.
> >
> > - Standard $L_2$ data fidelity term with a TV prior
> >
> > | **Method**         | **PSNR↑ / SSIM↑ / FID↓ / LPIPS↓** |
> > |---------------------|------------------------------|
> > | **DPS + TV** | 9.81 / 0.366 / 149.41 / 0.560 |
> > | **DiffPIR + TV** | 13.68 / 0.556 / 186.2 / 0.436 |
> > | **Dilack (ours)**         | **26.23** / **0.863** / **54.84** / **0.149** |
> >
> > > **[Q6]**: "**Gradient Calculation Method**: For calculating the gradient of your data-fidelity term, do you use automatic differentiation? If so, does this require backpropagation through the diffusion model? If yes, is this process computationally intensive, and how does it impact runtime?"
> >
> > Yes, our Dilack used automatic differentiation and thus the model requires backpropagation. This process is identical to DPS. However, as discussed in the **Limitation** subsection, the runtime increase is modest, increasing from 340 sec (DPS) to 390 sec per image, as shown in **Appendix F.1**.

---

> ### Comment · Reviewer_gLcf · 2024-11-25
> **Response to rebuttal**
>
> Thanks for the detailed answers.
>
> The authors answered most points. After careful reading, I still think that the following points where not properly answered.
>
> - W2 : the method is presented as a posterior maximization method with (2) when it is a posterior sampling method.
> - W3 : This is somewhat not consistent to introduce the proposed data-fildelity term from the DPS one. The proposed data-fidelity is more similar to the standard L2 data-fidelity (9). Indeed A^\dagger A is very different from Identity for highly ill-posed inverse problems. The discussion you provide to compare L_{Pi} and L_{Piac} should be incorporated in the manuscript.

---

> > ### Author Response · Authors · 2024-11-26
> > **Reponse to Reviewer's Further Questions**
> >
> > Thank you for your careful reading our response and additional questions!
> >
> > > **[W2+]**: The method is presented as a posterior maximization method with (2) when it is a posterior sampling method.
> >
> > We stated that Eq. (2) is "a typical formulation to solve the inverse problem based on Eq. (1)," so it does not necessarily have to make all the rest of the following works to be the same MAP methods. For example, DiffPIR (Zhu et al., 2023) also started with a similar formulation as our Eq. (2).
> >
> > Our method is fundamentally a posterior sampling method following others such as DPS except for a different guidance via another iterative algorithm with a number of different properties. Since our previous revision may not properly demonstrate this posterior samping aspect, we have conducted additional sampling experiments with a relaxed fidelity (e.g., controling a mask ratio and/or a regularization parameter in ADMM with TV). Specifically, we adjusted the guidance scale ($\rho$) to 0.5 and the masking ratio ($\nu$) to 0.9, relaxing our pseudo-inverse anchor to allow sampling diversity. These results, newly added in **Fig. S20** (p.g. 37) in **Appendix G.17**, demonstrate the diversity achievable under such conditions. Note that the diversity of posterior sampling methods including DPS, DiffPIR, DDPG and our proposed Dilack can be controlled by a regularization parameter for the trade-off in between robust performance and sample diversity and emphasizing the former may yield the results with less diversity so that they may not be from a posterior sampling. However, this trade-off does not necessarily have to change the nature of the proposed method - Dilack is a posterior sampling method with strong regularization parameters to emphasize more on the robust performance, but can also become a posterior sampling method with weak regularization parameters for diversity.
> >
> > > **[W3+]** : This is somewhat not consistent to introduce the proposed data-fildelity term from the DPS one. The proposed data-fidelity is more similar to the standard L2 data-fidelity (9). Indeed A^\dagger A is very different from Identity for highly ill-posed inverse problems. The discussion you provide to compare L_{Pi} and L_{Piac} should be incorporated in the manuscript.
> >
> > As you mentioned in "Indeed ... in the manuscript", we have revised our manuscript to include it. Note that we have updated the **4.1 PiAC** section in the revised manuscript with this clarification, highlighted in *blue*.
> >
> > Our previous explanation on how to drive the proposed data-fidelity term from the DPS was based on an approximate forward model with the identity forward model and the measurement constructed by ADMM-TV (however, note that this measurement could be relaxed by using a weak regularizer in TV). If this previous explanation still concerns you, then another explanation can be done based on a weighted least square data-fidelity that was used in the works for $L_{Pi}$. Our work can be seen as another approximation to the $L_{Pi}$ term with 1) the non-linear operator of ADMM-TV where the regularizer is controllable, 2) mask fidelity to ensure local interaction, and 3) the approximation of $A^\dagger A \approx I$, which is now relatively accurate due to the non-linear operator of ADMM-TV in $A^\dagger$. This new explanation, we hope, can provide another insight for our proposed $L_{PiAC}$.

---

> > > ### Comment · Reviewer_gLcf · 2024-12-02
> > > **Response to authors**
> > >
> > > Thanks for the comment. The updates done by the authors improve the quality of the manuscript. I will hence raise my score to 5. However, I do not raise my score above the acceptance threshold for the following reasons :
> > >
> > > W2: I am sorry but I need to again disagree with the authors on this point. The value of the regularization parameter does not make of your method "more" or "less" a posterior sampling method. The value of the regularization simply changes the posterior your are sampling, and makes it more or less narrowed around the maximum of the posterior. Moreover, re-reading again the introduction, it does not appear that the method is a posterior sampling method, but it looks like that it is still a posterior maximation method.
> > >
> > > W3: I am still concerned with the explanations of the proposed data-fidelity. I think it should be directly introduced, without relation with the one of DPS.

---

> > > > ### Author Response · Authors · 2024-12-02
> > > > **Reponse to Reviewer's Further Questions**
> > > >
> > > > > Thanks for the comment. The updates done by the authors improve the quality of the manuscript. I will hence raise my score to 5. However, I do not raise my score above the acceptance threshold for the following reasons :
> > > >
> > > > > W2: I am sorry but I need to again disagree with the authors on this point. The value of the regularization parameter does not make of your method "more" or "less" a posterior sampling method. The value of the regularization simply changes the posterior your are sampling, and makes it more or less narrowed around the maximum of the posterior. Moreover, re-reading again the introduction, it does not appear that the method is a posterior sampling method, but it looks like that it is still a posterior maximation method.
> > > >
> > > > > W3: I am still concerned with the explanations of the proposed data-fidelity. I think it should be directly introduced, without relation with the one of DPS.
> > > >
> > > > **Response:** Your comments have been helpful for us to improve our manuscript and we also appreciate the increased score. We would like to further clarify and address some incorrect descriptions. In fact, it seems that [W2] and [W3] are related, so we response to them at the same time.
> > > >
> > > > Indeed, our Eq. (2) made you confused since Eq. (2) is MAP, rather than a posterior sampling. Thus, L071-L077 should not be in the same paragraph as the prior works, but should be separated as posterior sampling methods that solve the inverse problem posed by Eq. (1), rather than solving Eq. (2). We will ensure that our proposed method (and other diffusion-based methods) is NOT solving Eq. (2) in the final version.
> > > >
> > > > It is interesting to note that the likelihood term (or data fidelity term) ususally persues the maximization (or minimization) even in posterior sampling-based methods. For example, the original DPS also uses "the surrogate function to **maximize** the likelihood, yielding **approximate** posterior sampling", as described in Page 4, the paragraph that includes Eq. (11) of (Chung et al., 2022a). Similarly, our Dilack has an iterative maximization step in the middle of the same poterior sampling instead of the analytic maximization step of the original DPS (see Alg. 1 of our manuscript vs. Alg. 1 of DPS in (Chung et al., 2022a)). Thus, our Dilack just like DPS yielded such diverse results for some of our experiments to demonstrate the capability of posterior sampling.
> > > >
> > > > Since our proposed method is tightly related with the original DPS (Chung et al., 2022a), it is not straightforward how to introduce our proposed method without the work of DPS. Now we clearly disconnect our method as well as other diffusion based methods from Eq. (2), we hope that this also clarifies your concern in [W3]. Note that our Dilack method and other prior diffusion-based works do share **the same score function for probabilistic diffusion**, which is the core for posterior sampling. Note that (Bora et al., 2017) was looking for the random seed z for the generative model or GAN) to maximize both the likelihood and the regularizer (i.e., solving Eq. (2)), but finding the random seed z when using the score function for solving Eq. (2) is challenging, which can not be solved by any method for now to the best of our knowledge.
> > > >
> > > > Warm regards,
> > > >
> > > > Paper 7416 authors

---

### Official Review · Reviewer_1sbM · 2024-11-04

**Soundness:** 2
**Presentation:** 2
**Contribution:** 2
**Rating:** 5
**Confidence:** 3

**Summary:**

The paper proposes a zero-shot method for highly ill-posed inverse problems (IP) based on a pre-trained diffusion model. The authors demonstrate that existing zero-shot IP methods tend to fail under conditions of severe degradation. This method introduces a pseudo-inverse anchor to constrain fidelity loss, utilizing an existing total variation (TV)-regularized solution. Additionally, they incorporate a locally masked loss, where the loss is dynamically activated or deactivated based on its similarity to the TV-regularized solution. The method achieves state-of-the-art (SOTA) results in highly ill-posed cases, such as lensless imaging and image deblurring.

**Strengths:**

1. The method is training-free which makes it computationally practical.

**Weaknesses:**

1. The other compared methods address the original (highly ill-posed) input, whereas the proposed method basically restores a more well-posed input (the TV-regularized solution). This raises the question of how fair the comparison is. It may make more sense to view the proposed method as a framework that can be integrated with existing methods rather than as a standalone inverse problem solution.

2. The setup presented in the paper is questionable; for example, how realistic is the kernel blur used in the study for real-world scenarios where the signal is almost completely lost?

3. The paper focuses on the non-blind case, assuming that the degradation forward model is known. It would be more interesting to investigate the more general, blind case instead, where the degradation model is unknown.

**Questions:**

1. In Figure 1, it appears that the diffusion model used is an unconditional model trained on ImageNet, which is known to be relatively weak due to the limited amount of training data compared to the large number of classes. One potential reason for the failure of existing IP methods may be the use of this weak pre-trained model. It would be interesting if the authors could reproduce the same analysis on the FFHQ dataset, where the model is much stronger, to determine whether the failures of existing methods are due to the weak pre-trained model or intrinsic issues.

---

> ### Author Response · Authors · 2024-11-22
> **Official Response by Authors**
>
> We would like to thank you for your constructive comments. See below for our point-by-point replies.
>
> > **[W1]**: "The other compared methods address the original (highly ill-posed) input, whereas the proposed method basically restores a more well-posed input (the TV-regularized solution). This raises the question of how fair the comparison is. It may make more sense to view the proposed method as a framework that can be integrated with existing methods rather than as a standalone inverse problem solution."
>
> If the "input" you mentioned is the initial image, all methods including ours use a zero mean Gaussian noise as an initial image (see line 1 of Algorithm 1 in the paper). If the "input" you mentioned is the measurement, then all methods including ours use a measurement vector y. Thus, in terms of "input", the comparisons were fair in our paper.
>
> The key difference between ours and other prior works is the design of guidance (*i.e.*, data fidelity term). This difference was motivated by studying highly ill-posed modern lensless imaging systems where prior diffusion-based works can not yield good images. Our PiAC guidance is the result of a number of advances in data fidelity term designs such as pseudo-inverse incorporation, TV regularization and mask operation so that the diffusion model can sample while ensuring (regularized) measurement consistency. So our proposed method is standalone.
>
> > **[W2]**: "The setup presented in the paper is questionable; for example, how realistic is the kernel blur used in the study for real-world scenarios where the signal is almost completely lost?"
>
> Kernel blurs for lensless imaging are in fact "real" kernels used in our lensless imaging systems (hardware) we have built. Kernel blurs for large motions, which are less ill-posed than the lensless imaging cases, may not occur as often as small motion blurs or lensless imaging, but they are realistic for imaging with zoom lens (small motion in camera will result in large motion blur).
>
> > **[W3]**: "The paper focuses on the non-blind case, assuming that the degradation forward model is known. It would be more interesting to investigate the more general, blind case instead, where the degradation model is unknown."
>
> This work is a demonstration that solving these highly ill-posed inverse problems is extremely challenging even though they are non-blind. Note that no prior arts were able to solve these challenges. Thus, it will be even more challenging to solve blind problems, so we believe that one may be able to try to solve them based on our work in the future. We believe that this work was an important stepping stone towards the blind, highly ill-posed inverse solvers. We will discuss potential directions for this important future works in the revision such as the potential work for handling model mismatch in lensless imaging.
>
> > **[Q1]**: "In Figure 1, it appears that the diffusion model used is an unconditional model trained on ImageNet, which is known to be relatively weak due to the limited amount of training data compared to the large number of classes. One potential reason for the failure of existing IP methods may be the use of this weak pre-trained model. It would be interesting if the authors could reproduce the same analysis on the FFHQ dataset, where the model is much stronger, to determine whether the failures of existing methods are due to the weak pre-trained model or intrinsic issues."
>
> We did not clearly indicate the pre-trained diffusion model used in Fig. 1. To clarify, the results in Fig. 1 are based on sampling outputs from existing methods (DPS, DiffPIR) and Dilack (ours) using an FFHQ pre-trained diffusion model, not ImageNet. Even with this stronger pre-trained model, existing zero-shot diffusion-based approaches still struggle significantly with tasks like lensless imaging and large motion blur, underscoring the intrinsic challenges of these modern practical imaging systems.

---

> > ### Author Response · Authors · 2024-12-02
> > **Additional Comment by Authors**
> >
> > We kindly remind you that only one day remains in the rebuttal period. After reviewing our response, please feel free to share any additional comments or concerns you may have.

---

### Official Review · Reviewer_M6aG · 2024-11-04

**Soundness:** 3
**Presentation:** 2
**Contribution:** 3
**Rating:** 5
**Confidence:** 4

**Summary:**

The paper presented a diffusion model designed to tackle highly ill-posed inverse problems involving large and complex kernels. It introduced extra data fidelity terms, including pseudo-inverse anchor for constraining (PiAC) fidelity loss, and employed a masked fidelity approach that dynamically emphasizes local consistency.

**Strengths:**

The methodology includes the development of the pseudo-inverse anchor which reflects a certain level of theoretical insight.

The authors claim improvements in restoration quality for specific applications like lensless imaging and large motion deblurring.

**Weaknesses:**

The central concept of the proposed approach is not clearly, and the overall description of the methodology lacks clarity.

The proposed algorithm incorporates many hyperparameters, which may complicate the implementation and optimization processes. This abundance of hyperparameters can also raise concerns regarding the stability and generalizability of the algorithm, as tuning them effectively may be challenging for practitioners.

**Questions:**

The experimental results presented in the manuscript are unconvincing. For example, in the table above Figure 4 regarding Large Motion Deblurring, the proposed method only outperforms other methods in 1 out of 8 cases. The authors should also discuss the computational cost associated with their method, as this is crucial for evaluating its practicality.

The parameters L_{PS} and L_{Pi} are defined in formulas (9) and (10), respectively, but are used in Figure 1 without reference to their definitions. It is essential for the authors to mention these definitions in the caption of Figure 1 to ensure clarity for readers.

The authors increase the kernel size from 64^2 to 256^2 without providing any intermediary states or specifying the size of the images being processed. Additionally, the condition numbers related to these kernels are not clearly demonstrated. Clarification on these points is necessary for reproducibility and understanding.

While the authors use a total variation (TV)-regularized solution tilde(x^*) for the pseudo-inverse anchor, they should discuss how about replacing TV with other deblurring techniques, such as BM3D. This comparison could provide valuable context regarding the effectiveness of their approach.

The authors incorporate TV into their algorithm but do not specify the boundary conditions used for the TV regularization. Moreover, guidance on how to select the parameter lambda_t before the TV term is missing. The authors also mention that the patch size and top percentage threshold for the MROI are hyperparameters that need optimal settings. They should address the practicality of including numerous hyperparameters and their potential impact on the algorithm's stability.

The inclusion of skip step guidance in Algorithm 1 lacks sufficient justification and introduction, making it difficult to assess its effectiveness. The authors should provide a rationale for this component to strengthen their argument.

The authors state, "Each element of MROI is set to 1 if the patch-wise difference between tilde{x}^* and hat{x}_{0|t} falls within the top percentage threshold of all sums of differences observed; otherwise, it is set to 0." This explanation is confusing and requires further clarification. The authors should provide a clear methodology for setting the MROI to enhance understanding.

---

> ### Author Response · Authors · 2024-11-22
> **Official Response by Authors, Part 1/3**
>
> We would like to thank you for your constructive comments. See below for our point-by-point replies.
>
> > **[W1]**: "The central concept of the proposed approach is not clearly, and the overall description of the methodology lacks clarity."
>
> We have revised the submission with a particular focus on improving clarity in Abstract, Introduction, Analysis, and Methodology sections. See the revised manuscript, with all modifications highlighted in ***blue*** text.
>
> > **[W2]**: "The proposed algorithm incorporates many hyperparameters, which may complicate the implementation and optimization processes. This abundance of hyperparameters can also raise concerns regarding the stability and generalizability of the algorithm, as tuning them effectively may be challenging for practitioners."
>
> Newly introduced hyperparameters are related to ADMM solver, which has been well-studied in the field, or ROI mask, which is relatively easy to optimize. Thus, determining them relied on prior works or simple tuning. Specifically, in lensless imaging, ADMM automatically adjusts hyperparameters through residual balancing [1,2], while fixing the $\lambda$ and $\rho$ during hardware calibration. For ROI masks, only two parameters — patch size and mask threshold — were introduced, which turned out to be not relatively sensitive to different tasks / data. Moreover, unlike comparison methods (*e.g.*, DPS, DiffPIR) that require task-specific tuning (*e.g.*, step-sizes), our method consistently uses a fixed guidance $\alpha$ value.
>
> [1] Wohlberg et al., "ADMM penalty parameter selection by residual balancing," arXiv, 2017.
> [2] Galatsanos et al., "Methods for choosing the regularization parameter and estimating the noise variance in image restoration and their relation," IEEE T-IP, 1992.
>
> > **[Q1-first part]**: "The experimental results presented in the manuscript are unconvincing. For example, in the table above Fig. 4 regarding Large Motion Deblurring, the proposed method only outperforms other methods in 1 out of 8 cases."
>
> Firstly, note that our proposed Dilack significantly outperformed all compared methods for two lensless imaging cases as shown in Table 1, which are already convincing experimental results. Secondly, as discussed in Section 3 (Challenge 1: Large motion deblurring), it was expected that DiffIR performs as well as the proposed method at least for large motion deblurring cases that have less complicated kernels than lensless imaging and the results were reported in Table 1, demonstrating that our propose Dilack yielded comparable performance to DiffIR and still outperformed most prior works. DDPG is a recent SOTA method that yielded great PSNR performance for large motion deblurring. However, the output image in DDPG often yielded artifacts near edges or details like characters on ImageNet as illustrated in Fig. 4 so that our Dilack outperformed DDPG in SSIM as well as FID, LPIPS. For FFHQ, faces have less details (e.g., no text, less sharp edges) over ImageNet, so this disadvantage of DDPG has been minimized, yielding SOTA performance in all metrix except for FID.
>
> > **[Q1-second part]**: "The authors should also discuss the computational cost associated with their method, as this is crucial for evaluating its practicality."
>
> **Appendix F.1** already discussed the computation cost, which is modestly increasing from 340 seconds to 390 seconds per image. Note that our Dilack is practically the only solution that works for lensless imaging cases, so this modest increase can be justified.
>
> > **[Q2]**: "The parameters $L_{LS}$ and $L_{Pi}$ are defined in formulas (9) and (10), respectively, but are used in Figure 1 without reference to their definitions. It is essential for the authors to mention these definitions in the caption of Figure 1 to ensure clarity for readers."
>
> We have added citations to Fig. 1(a) and included references to the corresponding equations for $L_{LS}$ (Eq. 9) and $L_{Pi}$ (Eq. 10) in the caption as suggested.

---

> > ### Author Response · Authors · 2024-11-22
> > **Official Response by Authors, Part 2/3**
> >
> > > **[Q3]**: "The authors increase the kernel size from 64$^2$ to 256$^2$ without providing any intermediary states or specifying the size of the images being processed. Additionally, the condition numbers related to these kernels are not clearly demonstrated. Clarification on these points is necessary for reproducibility and understanding."
> >
> > The image size was 256$^2$. Two kernel sizes were chosed following the prior works for lensless imaging (256$^2$) [Antipa et al., 2018; Asif et al., 2016; Boominathan et al., 2022; Poudel & Nakarmi, 2024; Lee et al. 2023; Poudel & Nakarmi, 2024; Li et al. 2023a; Rego et al. 2021; Zeng & Lam, 2021] and for motion deblurring (64$^2$) in DPS and DiffPIR. Then, we decreased the lensless kernel size to 64$^2$ as well as increased the motion blur kernel size to 256$^2$. No intermediate sizes were further studied since these two settings were able to provide enough results to confirm the behaviors of diverse diffusion-based methods with complicated kernels. As shown in Fig.2, for the (a) motion blur kernel, the condition number increases from 1.21 to 4.80 as the first row kernel size increases from 64 to 256. Similarly, for the (b) lensless camera PSF, it increases from 7.43 to 15.11. we have updated the **Analysis 1** and **Fig.2** in **Sec.3** of the revised manuscript (highlighted in *blue*).
> >
> > > **[Q4]**: "While the authors use a total variation (TV)-regularized solution ($\tilde{x}^*$) for the pseudo-inverse anchor, they should discuss how about replacing TV with other deblurring techniques, such as BM3D [3]. This comparison could provide valuable context regarding the effectiveness of their approach."
> >
> > > [3] Dabov et al., "Image denoising by sparse 3-D transform-domain collaborative filtering," IEEE T-IP 2007
> >
> > As suggested, we have conducted an experiment to replace TV with BM3D while preserving all other components of the framework. We tested on 3 images from FFHQ in the lensless imaging task with a turing PSF and the large motion deblurring task. Unfortutenaly, BM3D failed to provide an effective guidance (see **Appendix G.8** of the revision). Unlike a convex framework with TV, a non-convex framework with BM3D has local minima and thus it may not yield good guidance for highly ill-posed inverse problems.
> >
> > > **[Q5]**: "The authors incorporate TV into their algorithm but do not specify the boundary conditions used for the TV regularization. Moreover, guidance on how to select the parameter lambda_t before the TV term is missing. The authors also mention that the patch size and top percentage threshold for the $M_{ROI}$ are hyperparameters that need optimal settings. They should address the practicality of including numerous hyperparameters and their potential impact on the algorithm's stability."
> >
> > Neumann boundary condition was used. Note that the details regarding the initialization and re-initialization strategy for $\lambda_t$ during sampling are already provided in **Appendix G.6**. We refer you to our response to [W2] above for hyperparameter selection - many of them were determined using prior works or were not sensitive to different settings / data.
> >
> > > **[Q6]**: "The inclusion of skip step guidance in **Algorithm 1** lacks sufficient justification and introduction, making it difficult to assess its effectiveness. The authors should provide a rationale for this component to strengthen their argument."
> >
> > Note that ablation studies on skip step guidance were presented in Table S3 in **Appendix G.5**, leading to our current setting. Skip step guidance is a technique designed to balance the preservation of degraded input information with the generative quality of the output in image restoration tasks as also detailed in [Ding et al., 2023, Song et al., 2024] to ensure that the generated image loosely adheres to the features of the degraded input by applying guidance periodically rather than continuously. This skip step strategy can also be interpreted as alternating minimization between data fidelity minimization and diffusion process.

---

> > > ### Author Response · Authors · 2024-11-22
> > > **Official Response by Authors, Part 3/3**
> > >
> > > > **[Q7]**: "The authors $\tilde{x}^*$ state that "*Each element of $M_{ROI}$ is set to 1 if the patch-wise difference between $\tilde{x}^\star$ and $\hat{x}_{0|t}$ falls within the top percentage threshold of all sums of differences observed; otherwise, it is set to 0.*" This explanation is confusing and requires further clarification. The authors should provide a clear methodology for setting the $M_{ROI}$ to enhance understanding."
> > >
> > > In response to your feedback, we have revised **Section 4.2** (Masked Fidelity), with the updates highlighted in *blue*. In summary, the ROI mask is a 2D binary mask dynamically updated at each iteration $t$, based on the patch-wise $L1$-norm between the TV-regularized solution $\tilde{\mathbf{x}}^*$ and the current sample $\hat{\mathbf{x}}_{0|t}$. See the below for more details.
> > > - **Mask value = 1:** In these regions, $\mathcal{L}_{\text{PiAC}}$ enforces spatial consistency, preventing the generation of random, eccentric images from the pre-trained diffusion model.
> > > - **Mask value = 0:** Here, the diffusion prior $\mathbf{s}_{\theta^*}(\mathbf{x}_t, t)$ operates without $\mathcal{L}\_{\text{PiAC}}$, focusing solely on modulating beneficial local elements for image reconstruction (IR).

---

> > > > ### Author Response · Authors · 2024-12-02
> > > > **Additional Comment by Authors**
> > > >
> > > > We kindly remind you that only one day remains in the rebuttal period. After reviewing our response, please feel free to share any additional comments or concerns you may have.

---

### Meta-Review · Area_Chair_hC7K · 2024-12-17

**Metareview:**

The paper proposes a zero-shot diffusion model for highly ill-posed inverse problems with large and complex kernels. The reviewers find that the central concept of the proposed approach is not clear, and the paper is sometimes challenging to read. Besides, some experiment results are not satisfactory.
Based on the reviewers‘ rating, I believe that this paper is not yet ready for publication at ICLR.

**Additional Comments On Reviewer Discussion:**

Reviewer gLcf has riased the score to 5,  which is still below the acceptance threshold.

---

### Decision · Program_Chairs · 2025-01-22

Reject